# Moshpit SGD: Communication-Efficient Decentralized Training on Heterogeneous Unreliable Devices

**Max Ryabinin**[*]
Yandex, Russia
HSE University, Russia

**Eduard Gorbunov**[*]
MIPT, Russia
HSE University, Russia
Yandex, Russia

**Vsevolod Plokhotnyuk**
Yandex, Russia
HSE University, Russia

**Gennady Pekhimenko**
University of Toronto, Canada
Vector Institute, Canada

## Abstract

Training deep neural networks on large datasets can often be accelerated by using multiple compute nodes. This approach, known as distributed training, can utilize hundreds of computers via specialized message-passing protocols such as Ring All-Reduce. However, running these protocols at scale requires reliable high-speed networking that is only available in dedicated clusters. In contrast, many real-world applications, such as federated learning and cloud-based distributed training, operate on unreliable devices with unstable network bandwidth. As a result, these applications are restricted to using parameter servers or gossip-based averaging protocols. In this work, we lift that restriction by proposing Moshpit All-Reduce — an iterative averaging protocol that exponentially converges to the global average. We demonstrate the efficiency of our protocol for distributed optimization with strong theoretical guarantees. The experiments show 1.3x speedup for ResNet-50 training on ImageNet compared to competitive gossip-based strategies and 1.5x speedup when training ALBERT-large on preemptible compute nodes.

## 1 Introduction

Many recent influential discoveries in deep learning were enabled by the trend of scaling model and dataset size. Over the last decade, computer vision has grown from training models with 60 million parameters [1] on 1.3 million images [2] to 15 times more parameters [3] and 200 times more training data [4]. In natural language processing, the state-of-the-art language models [5] with 175 billion parameters are trained on over 570GB of texts, and even this does not saturate the model quality [6]. Training these large models can take years even with a top-of-the-line GPU server [7]. As a result, researchers and practitioners often have to run distributed training with multiple machines [8].

The dominant approach to distributed deep learning is data-parallel training [9], where each worker processes a fraction of the training batch and then exchanges its gradients with peers. If done naïvely, the gradient exchange step can overload the network as the number of workers increases. To combat this issue, modern distributed training algorithms take advantage of communication-efficient protocols, such as all-reduce [10]. These protocols allow workers to collectively compute the global average gradient with a constant communication overhead, regardless of the total number of peers.

---

[*]Equal contribution. Correspondence to `mryabinin0@gmail.com`.

35th Conference on Neural Information Processing Systems (NeurIPS 2021).

However, this efficiency makes the protocols more fragile: if any single participant fails or takes too long to process its batch, all other nodes are stalled. Therefore, scaling all-reduce protocols beyond a couple of servers requires specialized infrastructure with dedicated ultra-high bandwidth networking [8]. This kind of infrastructure is notoriously expensive compared to regular GPU servers or preemptible cloud VMs (see Appendix A for details).

Hence, it is tempting to consider distributed training on cheap unreliable instances as a cost-efficient alternative. A similar scenario arises in federated learning [11], where a single model is trained on heterogeneous devices due to privacy concerns. In both scenarios, workers use a shared network, where both latency and bandwidth can vary drastically due to interference from other users [12]. Furthermore, compute nodes are also subject to failure (or preemption) caused by factors beyond the protocol's control.

Running large-scale distributed training in these circumstances requires fault- and latency-tolerant algorithms [14, 15]. Most of these algorithms replace all-reduce averaging with **gossip**: each participant periodically downloads the latest parameters from their neighbors in a sparsely connected communication graph and averages the results. The updates gradually propagate through the graph over multiple rounds of averaging. However, the communication required to perform gossip grows linearly with the number of neighbors. Hence, when scaling to hundreds of peers, decentralized SGD has to keep the communication graph sparse, slowing down the convergence.

In this work, we propose an alternative approach. Instead of relying on a predefined communication graph, participants dynamically organize themselves into groups using a fully decentralized matchmaking algorithm called **Moshpit All-Reduce**. This strategy allows us to use communication-efficient all-reduce protocols that significantly reduce the network load compared to gossip-based averaging, while still being able to operate in unreliable hardware and network conditions.

Our contributions can be summarized as follows:

- We propose **Moshpit All-Reduce** — a novel decentralized averaging protocol for large-scale training with unreliable communication-constrained devices. According to our analysis, this method has exponential convergence rate independent of network topology and size.
- Armed with this averaging protocol, we develop **Moshpit SGD** for distributed optimization. We derive convergence rates for this algorithm and establish its equivalence to Centralized (Local) SGD in terms of iteration complexity under realistic assumptions.
- Our experiments demonstrate that Moshpit All-Reduce is significantly more efficient under network latency in realistic conditions. In particular, we train ResNet-50 on ImageNet to 75% accuracy 1.3 times faster than existing decentralized training algorithms and pretrain ALBERT-large 1.5 times faster on preemptible cloud VMs.[2]

## 2 Related Work

### 2.1 Data parallel training

The most popular way to accelerate neural network training with multiple devices is data-parallel training [9, 16, 17]. On each optimization step, this strategy splits the training batch among participants. Each participant then runs forward and backward passes to obtain gradients of the objective function on their part of the training batch. After that, we can aggregate the gradients from workers and perform an optimization step. There are two main strategies for this aggregation.

Historically, the first solution to gradient aggregation was to use Parameter Server (PS) [18]: a separate process or a dedicated server that keeps track of model parameters and optimizer statistics. After each round, the PS accumulates the gradients from each worker and updates the model parameters using SGD or any other optimizer, such as Adam [19]. Finally, the server distributes the updated model parameters to workers.

This strategy is robust and easy to implement, but it requires the server to regularly download full model gradients from every single worker. As a result, the parameter server can quickly become a bottleneck for large-scale training [20]. Since the original PS, researchers have proposed several modifications that reduce the communication load: accumulating multiple batches [22], compression [23, 24], server sharding [25, 26]. A more detailed overview is given in Appendix B.

---

[2]Implementation and code of experiments are at `github.com/yandex-research/moshpit-sgd`.

In turn, many practical distributed training systems have instead switched to averaging with All-Reduce [16, 27, 28, 17]. This name refers to a collection of protocols originally developed for HPC applications. Workers can follow these protocols to collectively compute the average[3] gradient more efficiently than with a central server.

## 2.2 Communication-efficient All-Reduce

There are several all-reduce protocols optimized for different network topologies. The simplest one is known as Butterfly All-Reduce [10]. Each of $N$ participants splits its local vector into $N$ chunks. Then, $i$-th worker aggregates $i$-th chunk of data from all peers and sends back the averaged chunk.

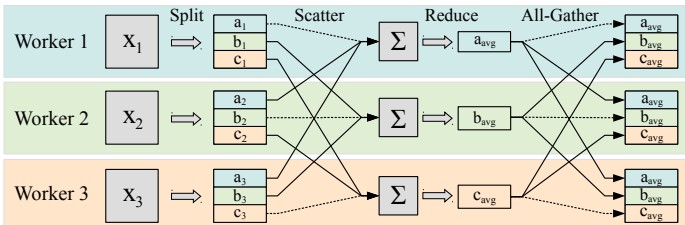

Figure 1: A schematic illustration of Butterfly All-Reduce.

As long as the vector size $s$ is greater than $N$, this protocol uses $\mathcal{O}\left(s \times \frac{N-1}{N}\right)$ total bandwidth on each worker. However, it requires all-to-all communication, which is not always practical for the HPC infrastructure due to network contention [10]. As a result, real-world systems typically use Ring or Tree All-Reduce, where each worker only communicates with a small subset of its peers.

These protocols enable highly efficient and scalable averaging with $\mathcal{O}(1)$ or $\mathcal{O}(\log N)$ total communication per worker, but they also share a common drawback: they cannot tolerate node failures or network instability. If any single participant fails to execute its part or takes long to respond, this paralyzes all other workers.

## 2.3 Distributed training in unstable conditions

Some distributed training applications must deal with unstable network bandwidth and/or unreliable workers. This issue is most prevalent in federated learning [11, 29, 30]. When dealing with privacy-sensitive data distributed across multiple actors, such as hospital servers [31, 32] or mobile phones [33, 34], one must train the model using whichever hardware and network available to those actors.

Another important motivational factor is cost: HPC-grade infrastructure can be prohibitively expensive, pushing researchers and practitioners towards commodity servers or preemptible cloud VMs that are significantly cheaper (see Appendix A). Another solution is to use volunteer computing [35, 36] with abundant, but even less reliable, compute resources.

Training under these conditions requires specialized strategies. At a small scale, one can deploy one or a few reliable parameter servers to aggregate the updates from workers. This strategy can tolerate individual node failures [37], but scales poorly due to the reasons discussed in Section 2.1.

## 2.4 Decentralized training

If there are too many participants for PS, it can be advantageous to use decentralized SGD via **gossip-based** averaging [38, 39, 14]. In this scenario, participants form a sparse graph: each worker periodically downloads parameters from its neighbors and mixes them with local parameters.

In essence, gossip-based averaging removes the communication bottlenecks of PS at the cost of using different local parameters on each peer. That said, gossip-based optimization algorithms can match, and sometimes even outperform, their centralized counterparts in terms of training speed [40, 41, 42, 14, 43]. However, the convergence properties of gossip averaging and gossip-based optimization methods significantly depend on the communication graph through the spectral properties of the mixing matrix [44, 42] or the Laplacian matrix of the network [45, 46].

---

[3]All-Reduce works with any commutative associative operation, such as min, max, or product.

Consequently, as the number of peers increases, gossip-based averaging has to either increase the number of neighbors (hence more communication) or accept slower convergence speed. Because of this, gossip is less communication-efficient than all-reduce algorithms reviewed in Section 2.2. However, gossip-based algorithms are more robust to changes, which makes them applicable to time-varying networks [47, 48, 49, 50] and federated learning [51, 52, 53].

# 3 Moshpit SGD

Large-scale training with unreliable participants requires a protocol that is both communication-efficient and fault-tolerant. Unfortunately, existing methods have only provide one of these properties. To better address our conditions, we propose Moshpit All-Reduce — a fully decentralized averaging protocol that combines the efficiency of all-reduce and the fault tolerance of gossip-based averaging.

The rest of this section is organized as follows:

- Section 3.1 describes the protocol and proves its correctness and communication efficiency;
- Section 3.2 provides the analysis of the protocol and proves exponential convergence rate for averaging and the rate matching the one of centralized Local-SGD for optimization;
- Section 3.3 contains implementation details for training with heterogeneous compute nodes.

## 3.1 Moshpit All-Reduce

The core idea of Moshpit All-Reduce is that workers perform averaging in small independent groups. That way, a single failed participant would only affect his current group. In turn, the composition of each group should be chosen dynamically to converge in the least number of steps. Ideally, if there are 9 peers with local parameters $\theta$, we can average them in 2 rounds, as demonstrated in Figure 2:

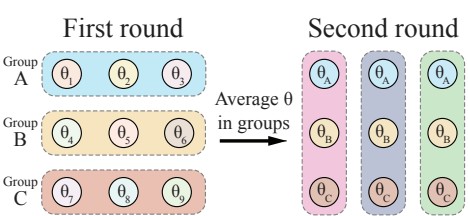

Figure 2: Example averaging order for 9 peers in 2 rounds. On each round, peers are split into 3 groups that run All-Reduce in parallel.

**Algorithm 1** Moshpit All-Reduce (for $i$-th peer)

**Input:** parameters $\{\theta_j\}_{j=1}^{N}$, number of peers $N$, $d$, $M$, number of iterations $T$, peer index $i$
$\theta_i^0 := \theta_i$
$C_i^0 := \texttt{get\_initial\_index(i)}$
**for** $t \in 1 \dots T$ **do**
    $\texttt{DHT}[C_i^{t-1}, t].\texttt{add}(\texttt{address}_i)$
    $\texttt{Matchmaking()}$ // wait for peers to assemble
    $\texttt{peers}_t := \texttt{DHT.get}([C_i^{t-1}, t])$
    $\theta_i^t, c_i^t := \texttt{AllReduce}(\theta_i^{t-1}, \texttt{peers}_t)$
    $C_i^t := (C_i^{t-1}[\texttt{1:}], c_i^t)$ // same as eq. (1)
**end for**
**Return** $\theta_i^T$

To achieve this in a decentralized system, we use Distributed Hash Tables (DHT) — a decentralized key-value storage; Appendix B contains its more detailed description. On each averaging round:

- Each worker computes its group key $C_i$;
- Workers add their network addresses to the DHT key corresponding to $C_i$;
- Each worker can now fetch a full list of peers that have the same $C_i$ and run All-Reduce with those peers.

Unfortunately, the averaging structure from Figure 2 is impossible to maintain when participants are constantly joining, leaving, and failing. However, we can achieve equivalent results without global structure using a simple rule: *if two peers were in the same group in round $t$, they must choose different groups in round $t+1$.*

A natural way to enforce this rule is to take advantage of the chunk indices from Butterfly All-Reduce (see Figure 1). Recall that each worker accumulates a *unique* chunk of parameters defined by an index $c_i$. By setting $C_i := c_i$, we can guarantee that any workers that were in the same group at a round $t$ will have different group indices in round $t+1$.

This averaging scheme can be generalized to more than two dimensions in order to fit a larger number of peers or reduce the group size. For a $d$-dimensional hypercube, nodes should find groups of peers that they have not communicated with during $d-1$ previous rounds. To that end, we define $C_i$ as tuples containing chunk indices from $d-1$ previous rounds ($t$ denotes the communication round):

$$C_i^t := (c_i^{t-d+1}, c_i^{t-d+2}, \ldots, c_i^t). \tag{1}$$

The above intuition can be formalized with Algorithm 1. Here, $N$ peers form a virtual $d$-dimensional grid with $M$ peers per row and average their parameters $\theta_i$ over $T$ rounds. DHT$[\cdot]$ is a shortcut for using the DHT to add or retrieve values for a given key. The Matchmaking step corresponds to the decentralized matchmaking procedure that organizes active workers with the same index into groups, described in detail in Appendix E. In turn, AllReduce denotes running all-reduce to compute the average $\theta$ in a given group. The get_initial_index function takes the peer index $i$ and returns $d-1$ integers in range $[0, M)$ such that the size of initial groups does not exceed $M$. This way, the groups formed on subsequent rounds will also have at most $M$ participants. One possible strategy is:

$$\texttt{get\_initial\_index}(i) = \left(\lfloor i/M^{d-1} \rfloor \bmod M\right)_{j \in \{1, \ldots, d\}} \tag{2}$$

If $N = M^d$ and there are no node/network failures, Algorithm 1 is equivalent to Torus All-Reduce [54], achieving the exact average after $d$ rounds of communication (see Appendix C.1). However, our typical use case is far from this perfect scenario; for example, some groups can have less than $M$ members. Furthermore, a peer might fail during all-reduce, causing its groupmates to skip a round of averaging. Still, Moshpit All-Reduce is applicable even in these conditions:

**Theorem 3.1** (Correctness). *If all workers have a non-zero probability of successfully running a communication round and the order of* peers$_t$ *is random, then all local vectors $\theta_i^t$ converge to the global average with probability 1:*

$$\forall i, \left\| \theta_i^t - \frac{1}{N} \sum_i \theta_i^0 \right\|_2^2 \xrightarrow[t \to \infty]{} 0. \tag{3}$$

*Proof (sketch, complete in Appendix C.2).* Running all-reduce with a subset of peers preserves the invariant $\frac{1}{N} \sum_i \theta_i^t = \frac{1}{N} \sum_i \theta_i^{t-1}$ and reduces the deviation of $\theta_i^t$ from the overall average. $\qquad \square$

**Complexity.** The matchmaking protocol is implemented over Kademlia DHT [55], meaning that each read and write operation needs at most $\mathcal{O}(\log N)$ requests and $\mathcal{O}(M)$ bandwidth to load peers$_t$.

After the matchmaking is over, each group runs a single all-reduce round to compute the average. In principle, Moshpit Averaging can use any general-purpose all-reduce protocol. We opted for a butterfly-like version (Figure 1), as it is simpler than Ring All-Reduce while still being communication-efficient. The communication complexity of this algorithm is $\mathcal{O}\left(\max(s, M) \times \frac{M-1}{M}\right)$, where $s$ is the size of vector $\theta$. Thus, the total time complexity of Algorithm 1 becomes:

$$\mathcal{O}\left(T \times \left[\log_2 N + M + \max(s, M) \times \frac{M-1}{M}\right]\right). \tag{4}$$

This compares favorably to gossip, where network load grows linearly with the number of neighbors.

## 3.2 Convergence analysis

### 3.2.1 Mixing properties of Moshpit Averaging

As stated in the previous section, Moshpit All-Reduce computes the exact average when $N = M^d$, which cannot be guaranteed in practice. Therefore, additional analysis is needed to establish how quickly Moshpit Averaging approximates the actual average of $N$ vectors stored on peers.

In the following theorem, we provide such analysis for a simplified version of Moshpit Averaging. One can find the full proof in Appendix C.3.

**Theorem 3.2.** *Consider a modification of Moshpit All-Reduce that works as follows: at each iteration $k \geq 1$, 1) peers are randomly split in $r$ disjoint groups of sizes $M_1^k, \ldots, M_r^k$ in such a way that $\sum_{i=1}^r M_i^k = N$ and $M_i^k \geq 1$ for all $i = 1, \ldots, r$ and 2) peers from each group compute their group average via All-Reduce. Let $\theta_1, \ldots, \theta_N$ be the input vectors of this procedure and $\theta_1^T, \ldots, \theta_N^T$ be the outputs after $T$ iterations. Also, let $\overline{\theta} = \frac{1}{N} \sum_{i=1}^N \theta_i$ Then,*

$$\mathbb{E}\left[\frac{1}{N} \sum_{i=1}^N \|\theta_i^T - \overline{\theta}\|^2\right] = \left(\frac{r-1}{N} + \frac{r}{N^2}\right)^T \frac{1}{N} \sum_{i=1}^N \|\theta_i - \overline{\theta}\|^2. \tag{5}$$

---

**Algorithm 2** Moshpit SGD

---

1: **Input:** starting point $\theta^0$, learning rate $\gamma > 0$, communication period $\tau \geq 1$
2: **for** $k = 0, 1, \ldots$ **do**
3:    **for** each peer $i \in P_{k+1}$ in parallel **do**
4:       Compute the stochastic gradient $g_i^k$ at the current point $\theta_i^k$
5:       **if** $k + 1 \mod \tau = 0$ **then**
6:          $\theta_i^{k+1} = \text{Moshpit All-Reduce}_{j \in P_{k+1}}(\theta_j^k - \gamma g_j^k)$ for $i$-th peer (Algorithm 1)
7:       **else**
8:          $\theta_i^{k+1} = \theta_i^k - \gamma g_i^k$
9:       **end if**
10:    **end for**
11: **end for**

---

In particular, this result implies that even if workers are randomly split into pairs at each iteration, the simplified version of Moshpit Averaging makes the average distortion (the left-hand side of Equation 5) less than $\varepsilon$ in expectation after $\mathcal{O}\left(\log(1/\varepsilon)\right)$ iterations. That is, this algorithm finds $\varepsilon$-accurate average on each node with the rate that *does not* depend on the spectral properties of the communication graph like gossip and its variants (see Section 2.4 and Appendix B.1). Since Moshpit Averaging prevents two peers from participating in the same groups during successive iterations, the actual algorithm should find $\varepsilon$-accurate averages on participating peers even faster than Equation 5 predicts. Moreover, in Appendix C.3 we explain how this result can be generalized to the case when $\{M_i^k\}_{i=1}^N$ and $r$ depends on $k$ or even is random. In Appendix C.4, we also provide the guarantees measuring how fast Algorithm 1 reduces the variance when averaging random vectors.

### 3.2.2 Moshpit SGD

We consider a classical distributed optimization problem

$$\min_{\theta \in \mathbb{R}^n} \left\{ f(\theta) = \frac{1}{N} \sum_{i=1}^N f_i(\theta) \right\}, \tag{6}$$

where $N$ is the number of workers and worker $i$ has access only to the function $f_i$.
We propose a new algorithm called Moshpit SGD to solve this problem (see Algorithm 2). In this algorithm, workers perform independent local SGD steps and periodically synchronize their parameters $\theta_i^k$ with other peers using Moshpit All-Reduce. Moreover, we define the indices of participating nodes at iteration $k$ as $P_{k+1}$ ($P_0 = \{1, \ldots, N\}$) allowing peers to vanish.

First of all, we list the key assumptions that we use in the convergence analysis of Moshpit SGD.

**Assumption 3.1** (Bounded variance). *We assume that for all $k \geq 0$ and $i = 1, \ldots, N$ stochastic gradients $g_i^k$ satisfy $\mathbb{E}\left[g_i^k \mid \theta_i^k\right] = \nabla f_i(\theta_i^k)$ and*

$$\mathbb{E}\left[\|g_i^k - \nabla f_i(\theta_i^k)\|^2 \mid \theta_i^k\right] \quad \leq \quad \sigma^2. \tag{7}$$

This assumption is classical in the stochastic optimization literature [56, 57]. We notice that our analysis can be generalized to the settings when the stochastic gradients satisfy less restrictive assumptions such as expected smoothness [58] or have more sophisticated structure similar to [59] using the theoretical framework from [60].

The following assumption controls the averaging properties and the effect of the peers' vanishing.

**Assumption 3.2** (Averaging quality & peers' vanishing). *We assume that the vanishing of peers does not change the global average of the iterates of Moshpit SGD too much, i.e., $P_{k+1} \subseteq P_k$ and $|P_k| \geq N_{\min}$ for all $k \geq 0$, $|P_{a\tau}| \leq 2|P_{a(\tau+1)}|$ for all non-negative integers $a \geq 0$, and there exist such $\widetilde{\theta} \in \mathbb{R}^n$ and a sequence of non-negative numbers $\{\Delta_{pv}^k\}_{k \geq 0}$ that $\forall k \geq 0$*

$$\mathbb{E}\left[\langle \theta^{k+1} - \widehat{\theta}^{k+1}, \theta^{k+1} + \widehat{\theta}^{k+1} - 2\widetilde{\theta}\rangle\right] \leq \Delta_{pv}^k, \ f \ convex; \tag{8}$$

$$\mathbb{E}\left[\langle \nabla f(\theta^k), \theta^{k+1} - \widehat{\theta}^{k+1}\rangle + L\|\widehat{\theta}^{k+1} - \theta^{k+1}\|^2\right] \leq \Delta_{pv}^k, \ f \ non\text{-}convex, \ L\text{-}smooth, \ (Def.\ D.1) \tag{9}$$

*where $N_k = |P_k|$, $\theta^{k+1} = \frac{1}{N_{k+1}} \sum_{i \in P_{k+1}} \theta_i^{k+1}$, and $\widehat{\theta}^{k+1} = \frac{1}{N_k} \sum_{i \in P_k}(\theta_i^k - \gamma g_i^k)$ for $k \geq 0$.*

*Moreover, we assume that for some $\delta_{aq} \geq 0$ and for all non-negative integers $a \geq 0$,*

$$\mathbb{E}\left[\frac{1}{N_{a\tau}}\sum_{i \in P_{a\tau}}\|\theta_i^{a\tau} - \theta^{a\tau}\|^2\right] \leq \gamma^2 \delta_{aq}^2. \tag{10}$$

If $P_k = P_{k+1} = \{1, \ldots, N\}$ for all $k \geq 0$, i.e., peers do not vanish, then $\theta^k = \widehat{\theta}^k$ and properties (8, 9) hold with $\Delta_{pv}^k \equiv 0$ for all $k \geq 0$. Moreover, according to the mixing properties of Moshpit Averaging established in Theorem 3.2, inequality 10 holds after $\mathcal{O}\left(\log\left(1/\gamma^2\delta_{aq}^2\right)\right)$ iterations of Algorithm 1. Therefore, the assumption above is natural and well-motivated.

Under these assumptions, we derive the convergence rates both for convex and non-convex problems. The full statements and complete proofs are deferred to Appendix D.

**Theorem 3.3** (Convex case). *Let $f_1 = \ldots = f_N = f$, function $f$ be $\mu$-strongly convex (Def. D.2) and $L$-smooth (see Def. D.1), and Assumptions 3.1 and 3.2 hold with $\Delta_{pv}^k = \delta_{pv,1}\gamma\mu\mathbb{E}[\|\theta^k - \theta^*\|^2] + \gamma^2\delta_{pv,2}^2$ and $\widetilde{\theta} = \theta^*$, where $\theta^* \in \arg\min_{\theta \in \mathbb{R}^n} f(\theta)$ and $\delta_{pv,1} \in [0,1)$, $\delta_{pv,2} \geq 0$. Then there exists a choice of $\gamma$ such that $\mathbb{E}\left[f(\overline{\theta}^K) - f(\theta^*)\right] \leq \varepsilon$ after $K$ iterations of Moshpit SGD, where $K$ equals*

$$\widetilde{\mathcal{O}}\left(\frac{L}{(1-\delta_{pv,1})\mu} + \frac{\delta_{pv,2}^2 + \sigma^2/N_{\min}}{(1-\delta_{pv,1})\mu\varepsilon} + \sqrt{\frac{L((\tau-1)\sigma^2 + \delta_{aq}^2)}{(1-\delta_{pv,1})^2\mu^2\varepsilon}}\right), \ \mu > 0; \tag{11}$$

$$\mathcal{O}\left(\frac{LR_0^2}{\varepsilon} + \frac{R_0^2(\delta_{pv,2}^2 + \sigma^2/N_{\min})}{\varepsilon^2} + \frac{R_0^2\sqrt{L((\tau-1)\sigma^2 + \delta_{aq}^2)}}{\varepsilon^{3/2}}\right), \ \mu = 0, \tag{12}$$

*where $\overline{\theta}^K = \frac{1}{W_K}\sum_{k=0}^{K}\frac{1}{N_k}\sum_{i \in P_k} w_k\theta_i^k$, $w_k = (1-\gamma\mu)^{-(k+1)}$, $W_K = \sum_{k=0}^{K} w_k$, $R_0 = \|\theta^0 - \theta^*\|$ and $\widetilde{\mathcal{O}}(\cdot)$ hides constant and $\log(1/\varepsilon)$ factors.*

That is, if $\delta_{pv,1} \leq 1/2$, $N_{\min} = \Omega(N)$, $\delta_{pv,2}^2 = \mathcal{O}(\sigma^2/N_{\min})$, and $\delta_{aq}^2 = \mathcal{O}((\tau-1)\sigma^2)$, then Moshpit SGD has the same iteration complexity as Local-SGD in the homogeneous case [61, 62]. However, the averaging steps of Moshpit SGD are much faster than those of the parameter-server architecture when the number of peers is large. Also, unlike the state-of-the-art convergence guarantees for Decentralized Local-SGD [63], our bounds do not depend on the spectral properties of the communication graph (see Appendix B.1 for the details).

**Theorem 3.4** (Non-convex case). *Let $f_1 = \ldots = f_N = f$, function $f$ be $L$-smooth and bounded from below by $f_*$, and Assumptions 3.1 and 3.2 hold with $\Delta_{pv}^k = \delta_{pv,1}\gamma\mathbb{E}[\|\nabla f(\theta^k)\|^2] + L\gamma^2\delta_{pv,2}^2$, $\delta_{pv,1} \in [0, 1/2)$, $\delta_{pv,2} \geq 0$. Then there exists such choice of $\gamma$ that $\mathbb{E}\left[\|\nabla f(\theta_{rand}^K)\|^2\right] \leq \varepsilon^2$ after $K$ iterations of Moshpit SGD, where $K$ equals*

$$\mathcal{O}\left(\frac{L\Delta_0}{(1-2\delta_{pv,1})^2\varepsilon^2}\left[1 + \tau\sqrt{1-2\delta_{pv,1}} + \frac{\delta_{pv,2}^2 + \sigma^2/N_{\min}}{\varepsilon^2} + \frac{\sqrt{(1-2\delta_{pv,1})(\delta_{aq}^2 + (\tau-1)\sigma^2)}}{\varepsilon}\right]\right),$$

$\Delta_0 = f(\theta^0) - f(\theta^*)$ and $\theta_{rand}^K$ is chosen uniformly from $\{\theta^0, \theta^1, \ldots, \theta^{K-1}\}$ defined in As. 3.2.

Again, if $\delta_{pv,1} \leq 1/3$, $N_{\min} = \Omega(N)$, $\delta_{pv,2}^2 = \mathcal{O}(\sigma^2/N_{\min})$, and $\delta_{aq}^2 = \mathcal{O}((\tau-1)\sigma^2)$, then the above theorem recovers the state-of-the-art results in the non-convex case for Local-SGD [64, 63].

### 3.3 Implementation details

Training on heterogeneous unreliable hardware also poses a number of engineering challenges. The most obvious one is that the system must be able to recover from node failures. To address this challenge, we use a fully decentralized infrastructure where all information is replicated in a Distributed Hash Table; see Appendix B.5 for details. When a new worker joins midway through training, it can download the latest model parameters and metadata from any other peer (see Appendix F). Another challenge arises when devices in a group have uneven network bandwidth. In that case, we dynamically adjust the communication load of each peer to avoid being bottlenecked. More information on this procedure can be found in Appendix G.

# 4 Experiments

In this section, we conduct empirical evaluation of the proposed averaging protocol and its corresponding optimization algorithm. First, we check the theoretical properties of Moshpit All-Reduce in a controlled setup (Section 4.1). Then, we compare Moshpit SGD with other distributed methods on practical tasks of image classification and masked language model pretraining (Sections 4.2 and 4.3).

## 4.1 Decentralized averaging

In this series of experiments, we aim to empirically verify the convergence and fault tolerance properties proven in Section 3.2. To measure this in a controlled setting, we create peers with parameters that are scalar values drawn from the standard Gaussian distribution. We study the convergence of different distributed methods with respect to the number of workers $N$ and their individual failure rate for a single iteration of averaging $p$ (failed peers return in the next round).

We compare Moshpit Averaging with the following algorithms from prior work: All-Reduce (with restarts in case of node failures), Gossip, PushSum (equivalent to the method described in [15]). Also, we provide the results of averaging in random groups as a simpler version of our approach. However, the implementation of group averaging maintains approximately the same group size across all iterations: this property might be hard to achieve in a decentralized setting, and as a result, the estimate of this method's performance should be considered highly optimistic.

We report the average squared difference between the worker parameters and the actual average of all values; the results are averaged across 100 restarts from different random initializations. We compare the convergence for 512–1024 peers and consider failure probabilities ranging from 0 to 0.01. For Moshpit Averaging and random group averaging, we use groups of size 32, which corresponds to $M = 32$ and $d = 2$ for Algorithm 1.

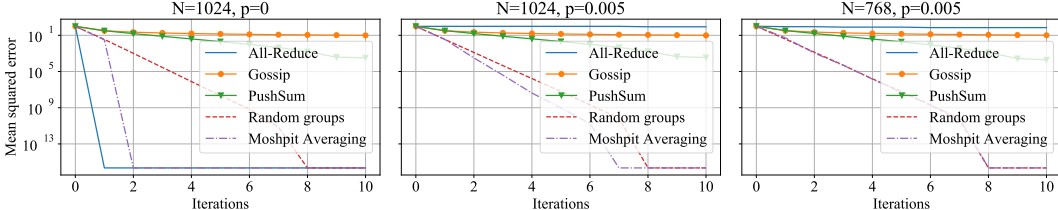

Figure 3: Convergence of averaging algorithms in different configurations.

Figure 3 displays the results of experiments for several combinations of $N$ and $p$; the complete results with additional grid configurations are available in Appendix I. We make several key observations:

1. When the failure rate of each peer is zero, standard All-Reduce predictably computes the average faster than all other methods. However, as soon as $p$ reaches a value of at least 0.005, the number of retries needed for the success becomes prohibitively high.

2. Previous decentralized averaging methods, such as Gossip or PushSum, require significantly more iterations for convergence to the global average than Moshpit All-Reduce, likely due to the structure of their communication graphs.

3. As discussed in Section 3.1, when the total number of peers is equal to the grid capacity and there are no failures, Moshpit All-Reduce matches the result of regular All-Reduce with the number of steps equal to the number of grid dimensions (2 in this case).

4. Averaging in random groups can perform comparably to Moshpit Averaging when the number of peers is less than half of the grid capacity. The reason for this behavior is that when the workers do not fully occupy the grid, the group sizes are no longer guaranteed to be equal across groups and across iterations. In the worst case, there can be groups of only one peer for certain grid coordinates, which may significantly affect the convergence. However, as the grid utilization grows, Moshpit Averaging starts to outperform random group averaging. Moreover, even if we use 512 peers, arranging them in a proper 8x8x8 grid leads to faster convergence.

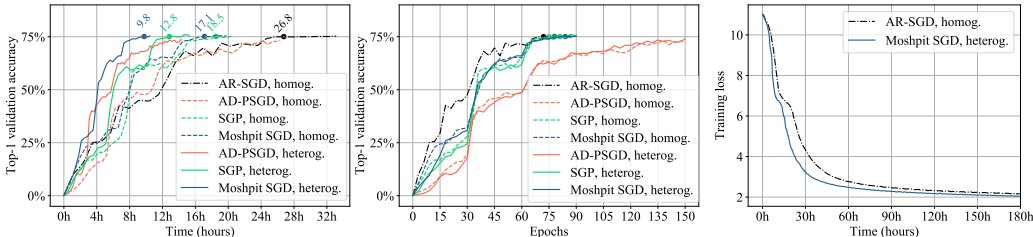

Figure 4: **(Left, Middle)** ResNet-50 top-1 validation accuracy for ImageNet as a function of training time (left) and epochs (middle). **(Right)** Full training objective (MLM + SOP) of ALBERT-large on BookCorpus as a function of training time.

## 4.2 ImageNet training

Here, we evaluate the performance of Moshpit SGD in distributed training. More specifically, we train ResNet-50 [65] on the ILSVRC [2] dataset, following the training protocol of [16]. Trainers use SGD with Nesterov momentum with a batch size of 256 and 32-bit precision regardless of the GPU type[4]. We evaluate the following training strategies:

- **All-Reduce SGD (AR-SGD)** — traditional distributed training with all-reduce gradient averaging;
- **Asynchronous Decentralized Parallel SGD (AD-PSGD)** — parallel SGD that runs gossip communication in a cycle: each worker averages parameters with 2 neighbors [66]. Communication rounds are overlapped with computation;
- **Stochastic Gradient Push (SGP)** — a more advanced algorithm with an exponential communication graph and push-based communication [15];
- **Moshpit SGD** — similar to **SGP**, but with 1 round of Moshpit Averaging instead of PushSum.

We report top-1 validation accuracy as a function of training time in two experimental setups:

- **Homogeneous**: 16 servers with a single Tesla V100-PCIe GPU, 6 CPU cores, and 64GB RAM.
- **Heterogeneous**: a total of 81 GPUs (V100, 1080Ti, and P40) across 64 servers and workstations.[5]

All servers and workstations communicate over the network with 1Gb/s Ethernet (non-dedicated symmetric bandwidth). The machines are located in two data centers and one office within 300 km of one another. The communication latency is 1–6ms depending on the location. To simulate shared usage, at the beginning of each communication round we inject additional latency sampled from the exponential distribution [67] with the mean of 100ms.

For Moshpit SGD, we use a two-dimensional "grid" with 4 and 8 groups for homogeneous and heterogeneous setups respectively. For AD-PSGD, we attempt to compensate for slow convergence by training for 60 more epochs without changing the learning rate schedule. Finally, we only report AR-SGD in the first setup, as it is unsuitable for heterogeneous hardware.

The results in Figure 4 (Left) demonstrate that the two most efficient strategies for our setting are Moshpit SGD and SGP. In the **homogeneous** setup, Moshpit is only slightly more efficient than SGP, likely due to higher efficiency of all-reduce. This advantage increases to over 30% for the **heterogeneous** setup with 64 servers. In turn, AR-SGD demonstrates the best performance per iteration, but its training time is by far the longest due to network latency ($1.5\times$ of Moshpit SGD). Finally, AD-PSGD predictably shows the best throughput (time per epoch), but achieves lower accuracy even after training for 150 epochs. We report results for smaller setups in Appendix J.

## 4.3 Masked Language Model training

Finally, we evaluate Moshpit All-Reduce training performance in the wild with preemptible cloud instances. For this experiment, we perform one of the most resource-demanding tasks in modern deep learning — unsupervised pretraining of Transformers [68, 69, 70, 5]. We opt for the ALBERT model [71] to make better use of communication-constrained devices. This model has fewer trainable parameters due to layer-wise weight sharing.

---

[4]For GPUs that cannot fit this into memory, we accumulate gradients over 2 batches of 128 examples.

[5]We provide a detailed configuration in Appendix H.

Specifically, we train ALBERT-large (18M parameters) on the BookCorpus [72] dataset, following the training setup from the original paper. We minimize the masked language modeling loss (MLM) along with the sentence order prediction loss (SOP) using the LAMB optimizer [17] with a global batch size of 4096 and sequence length 512. We measure convergence in terms of full training loss [73, 74]. Similarly to Section 4.2, we use two training setups:

- **Homogeneous:** a single cloud instance with 8 Tesla V100-PCIe GPUs and 56 vCPUs;
- **Heterogeneous:** a total of 66 preemptible GPUs, 32 of which are cloud T4, and the remaining 34 are various devices rented on a public marketplace.

Despite the fact that the latter setup has almost $3\times$ more raw compute[6], its hourly rent costs less than the homogeneous setup due to relying on preemptible instances[7]. This instance type is much cheaper than regular cloud instances, but it can be interrupted at any time. As a side-effect, the participants in **heterogeneous** setup are also spread across 3 continents with uneven network bandwidth, ranging from 100Mb/s to 1500Mb/s per worker. These limitations make it impractical to deploy conventional all-reduce protocols. By contrast, the fully decentralized nature of Moshpit SGD allows it to operate on unreliable nodes.

In this setup, the participants accumulate gradients over multiple local batches and use DHT to track the global batch size. Once the swarm collectively accumulates gradients over 4096 training samples, it runs 2 rounds of Moshpit All-Reduce with $M=8$ and $d=2$. Unfortunately, training with simple parameter averaging does not converge, likely due to diverging LAMB statistics. To mitigate this issue, workers recover "pseudo-gradients" [76, 77] after averaging to update the optimizer statistics.

Figure 4 (right) demonstrates that Moshpit SGD with a fully preemptible fleet of machines trains 1.5 times faster than the traditional data-parallel setup. The final loss achieved by two training strategies is the same within the margin of error. A closer investigation reveals that this speedup is entirely explained by the reduced iteration time. An interesting observation is that the iteration time of Moshpit SGD varies between 10–22 seconds, while AR-SGD consistently spends 25s per step. This can be explained by natural variation in the preemptible fleet size: there were 30–66 active participants depending on the resource availability.

## 5 Conclusion and future work

In this work, we propose Moshpit All-Reduce, a decentralized averaging protocol intended for distributed optimization in unstable and network-constrained environments. It has favorable theoretical properties when compared to gossip-based approaches and achieves considerable speedups in distributed training for image classification and masked language modeling.

Our approach was primarily designed for cloud-based training and federated learning, as well as for distributed training on unreliable instances; future work might explore additional settings, such as collaborative training of neural networks. Another potential research direction is to study the interactions of Moshpit All-Reduce with other methods that improve communication efficiency of distributed optimization, such as gradient compression. Finally, the idea of arranging All-Reduce nodes into groups can be improved to address specific issues that may arise in practice, such as the varying number of workers and their geographical distribution.

## Acknowledgements

We would like to thank Anastasia Koloskova, Liudmila Prokhorenkova and Anton Osokin for helpful feedback and discussions. We are also grateful to the anonymous reviewers for their suggestions on improving the paper. Finally, we would like to thank Dmitry Afanasiev, Vladimir Aliev, Anand Jayarajan and Michael Solotky for their suggestions on the technical aspects of our study. This project was supported in part by the Canada Foundation for Innovation JELF grant, NSERC Discovery grant, AWS Machine Learning Research Award, and Facebook Faculty Research Award. The paper was also partially supported by by a grant for research centers in the field of artificial intelligence, provided by the Analytical Center for the Government of the Russian Federation in accordance with the subsidy agreement (agreement identifier 000000D730321P5Q0002) and the agreement with the Moscow Institute of Physics and Technology dated November 1, 2021 No. 70-2021-00138. The computational resources for the experiments were provided by the Amazon Research Awards program and Yandex.

---

[6]Based on official performance benchmarks [75].

[7]Please refer to Appendix H for full experimental setups.

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
