# Supplementary Material

## A   GPU instance costs

This section provides a brief cost analysis of typical deep learning compute resources both in the cloud and on-premises. For brevity, we limit this analysis to the popular GPUs available at the time of submission. Note that the exact costs will depend on a variety of factors such as the cloud provider, the region, electricity costs, and market fluctuations. Therefore, we warn the reader to consider this analysis only as a rough estimate.

Specifically, we estimate the compute costs for the occasional usage scenario: running a single set of experiments over several weeks or conducting infrequent experiments. This scenario covers most research scientists and small organizations. The most straightforward way to provision a GPU server in such a scenario is to rent it from a cloud provider (e.g., GCP or AWS) or a public marketplace (e.g., Vast.ai or Golem).

While the exact server specifications vary from one provider to another, there are two broad categories of GPU machines: regular and preemptible. Regular instance types typically offer 1–8 GPUs per node with tight uptime guarantees (typically $99.99\%$) and a high-bandwidth network (tens of Gb/s). In turn, preemptible instances provide the same resource type at a significant discount with the condition that the machine can be terminated at any time after short notice.

To account for individual variations, we report the average rent price over three popular cloud providers. We consider three popular instance types: two high-end instances with 8 Tesla V100 or A100 GPUs and a low-end instance with a single Tesla T4 GPU. We also describe several low-end servers and workstations available on a public marketplace. Unlike cloud VMs, these instances are hosted on non-curated hardware with less uptime guarantees (typically $95\% - 99.9\%$), slower network and significant variation in performance. However, marketplace instances are the cheapest in terms of cost per TFLOPS. To quantify this, we report the average over three most affordable instances that fit the chosen minimum requirements.

As a point of comparison, we also measure each system's training performance for BERT-Large [68] fine-tuning on SQuAD v1.1 [78] in PyTorch with mixed precision. We follow the official benchmarking protocol by [75] and reuse the official performance results for V100, A100, and T4 instances. The only exception is GTX 1080Ti, where we use full 32-bit precision because that device does not support efficient half-precision operations.

Table 1: Cloud and marketplace GPU instance pricing for short-term usage.

| Minimum system specifications | | | | Average cost, $/hour | | BERT-Large |
|---|---|---|---|---|---|---|
| GPU | CPU cores | CPU type | RAM, GB | Regular | Preemptible | training samples/s |
| Cloud instances | | | | | | |
| 8× V100 | 64 | Intel Xeon Broadwell | 480 | 23.47 | 7.13 | 354 |
| 8× A100 | 96 | AMD Epyc ROME | 960 | 30.65 | 10.18 | 755 |
| 1× T4 | 4 | Intel Xeon Cascade Lake | 16 | 0.46 | 0.18 | 18 |
| Marketplace instances | | | | | | |
| 6× 3090 | 32 | AMD Epyc Rome | 480 | 5.04 | 4.17 | 154 |
| 4× 2080Ti | 16 | Intel Xeon Haswell | 240 | 0.96 | 0.84 | 83.4 |
| 1× RTX 1080Ti | 8 | Intel Xeon Haswell | 16 | 0.22 | 0.16 | 12 |

Table 1 shows two main tendencies. First, preemptible *cloud* instances are, on average, three times cheaper than their non-preemptible counterparts[8]. Second, the high-end HPC-grade servers that offer the highest raw performance are less cost-effective than lower-tier servers and marketplace instances. In theory, one could match the raw floating-point performance of a 8×V100 instance at a fraction of its cost using multiple lower-tier workstations, such as 4× RTX 2080Ti, with a smaller total cost.

---

[8]The cost can be up to $11\times$ cheaper for some instance types, e.g. Azure V100 instances in the central US region at the time of writing.

However, in practice, running distributed training with these workstations is challenging due to their unreliability and slow network connection.

Note that this analysis does not represent the cloud costs for sustained GPU usage. If an organization plans to constantly use GPU resources over a period of multiple years, they can reduce the costs by deploying their own compute infrastructure or relying on the sustained usage discounts reaching up to 60–70%. Thus, the long-term compute costs are much harder to analyze and depend on a number of additional factors, such as local electricity prices for on-premise infrastructure. However, this scenario offers similar trade-offs: HPC-grade infrastructure offers greater interconnectivity, but requires expensive network interface cards, high-end switches and a more complex setup process.

## B  Additional Related Work

In this section, we review some of the papers relevant to our work, but omitted from the main part due to space constraints.

### B.1  Decentralized training

In this subsection, we give additional details about the dependence of gossip-based optimization methods on the spectral properties on the communication graph through the spectral properties of the mixing matrix [44, 42] or the Laplacian matrix [45, 46] of the network. That is, gossip finds approximate average on nodes with accuracy $\varepsilon$ after $\mathcal{O}\left((1 - \lambda_2(\mathbf{M}))^{-1} \log(\varepsilon^{-1})\right)$ iterations, where $\mathbf{M}$ is the mixing matrix and $\lambda_2(\mathbf{M})$ is the second largest eigenvalue of $\mathbf{M}$ when sorted by absolute value. The quantity $\eta = 1 - \lambda_2(\mathbf{M})$ is called the spectral gap of the mixing matrix $\mathbf{M}$, and $\eta^{-1}$ is typically a polynomial of the total number of nodes $N$ when the maximal degree of the node is $\mathcal{O}(1)$. For example, for uniformly averaging $\mathbf{M}$ one can show that $\eta^{-1} = \mathcal{O}(N^2)$ for the ring topology (node degree 2), $\eta^{-1} = \mathcal{O}(N)$ for the two-dimensional torus topology (node degree 2), and $\eta^{-1} = \mathcal{O}(1)$ for the fully connected graph (node degree $N - 1$); one can find more examples in [79]. Similarly, the communication complexity of decentralized optimization methods often has multiplicative dependence on either $\mathcal{O}(\eta^{-1})$ (see [80] and references therein) or $\mathcal{O}(\eta^{-1/2})$ [42, 46, 81, 82], which is not improvable for gossip-based methods [83, 40].

Contrary to this, Moshpit All-Reduce does not depend on a fixed communication graph and the properties of its mixing matrix. However, it depends on the number of averaging groups and the total number of peers (see Theorem 3.2), which can be viewed as properties of a time-varying random communication graph. Fortunately, this dependence is often much better than in gossip: as we mentioned in the main part of the paper, even if workers are randomly split into pairs at each iteration, the simplified version of Moshpit All-Reduce makes the average distortion (the left-hand side of Equation 5) at least 2 times smaller after each round on average.

### B.2  Compressed communication

Another popular approach to address the communication bottleneck is communication compression [84, 85, 86, 87, 88]: before sending any information (e.g., iterates, gradients, Hessians or more sophisticated data) over the network, peers compress this information by applying a possibly random transformation. As the result, peers send fewer bits for each communication round, but the total number of communication rounds needed to achieve the predefined accuracy of the solution increases. However, compression can be useful in situations when the reduction in communication costs of one round is more important than the increase in the number of these rounds [89].

There are two distinct groups of works on distributed training with compressed communication: ones that focus on unbiased compression operators (e.g., Rand-K, $\ell_p$-quantization) and ones studying algorithms with biased compressors (e.g., Top-K); see a detailed summary of popular compression operators in [90]). Quantized SGD (QSGD) [85] and TernGrad [91] were among the first compression methods with convergence guarantees. Next, the convergence analysis of these methods was generalized and tightened in the (strongly) convex case in [92]. Moreover, the authors of [92] proposed a modification of QSGD called DIANA: this algorithm is based on the quantization of gradients' differences, which helps it achieve linear convergence in the strongly convex case when peers compute full gradients. Next, DIANA was generalized to arbitrary unbiased compression in [93], where authors also developed and analyzed the variance-reduced version of DIANA. After that, several

further modifications, such as Accelerated DIANA [94] and DIANA with bidirectional compression [95, 96], were proposed. Finally, we refer the reader to [97, 98, 99, 100] for state-of-the-art results for distributed methods with unbiased compression in the non-convex case.

However, naïve application of biased compression operators can lead to significantly worse performance in practice. For instance, as it was shown recently in [90], parallel SGD with Top-1 compression can diverge exponentially fast. Therefore, biased compressors are used jointly with so-called error-compensation [84]. The first analysis of Error-Compensated SGD (EC-SGD) was proposed in [101, 102] which then was generalized and tightened in [90]. Next, several further improvements, such as an accelerated version of EC-SGD [103] and linearly converging EC-SGD [95], were recently proposed. However, current theory does not show any superiority of distributed methods with biased compressors to the ones with unbiased compression operators. In addition, one can combine decentralized communication with compression. Such combinations with unbiased compression operators were studied in [104, 105] and with biased operators in [24, 106]. In this paper, we do not study the interaction of different compression methods and Moshpit Averaging, leaving this promising direction to future work.

### B.3 Multiple local steps

Alternatively, to reduce the impact of the communication bottleneck, it is possible to perform several local optimization steps on each peer between the communication rounds. This approach is based on the idea that the increased computational load of peers will decrease the number of communication rounds required to obtain the optimal parameters; it is frequently used in federated learning [107, 108]. In particular, one of the most popular methods with multiple local steps is called Local-SGD or Federated Averaging [107, 109]. The first results on its convergence were given in [109, 110], and later they were tightened and generalized both for homogeneous [61, 62] and heterogeneous cases [61, 111]. Recently, further modifications of Local-SGD were proposed and analyzed: these modifications include acceleration [112], variance reduction [60], communication compression [113, 98, 99], decentralization [64, 63], adaptive and proximal methods [76, 114], and resistance to client drift [59]. Moshpit SGD can perform multiple local gradient steps before synchronization by design, as shown in Algorithm 2.

### B.4 Asynchronous methods

In the previous subsections, we mostly discussed synchronous distributed methods, since they are more widespread and better studied than asynchronous ones. Mainly, this is because asynchronous methods are more difficult to implement, debug and analyze under general assumptions. However, such methods can be more efficient in terms of using computational resources, which leads to faster wall-clock convergence [115]. In recent years, several asynchronous stochastic methods [116, 117, 118], methods with no shared memory [119, 120], and methods with delayed updates [121, 122, 123, 95] were proposed and analyzed: one can find more details in a recent survey [115]. Moshpit SGD belongs to this family of asynchronous approaches as well, because the averaging steps happen in smaller groups and can be interleaved with local parameter updates.

### B.5 Distributed Hash Tables

In this work, we set out to improve distributed averaging with a dynamic matchmaking protocol. Without a central server, this protocol relies on decentralized data structures to organize peers. The main data structure we use is the Distributed Hash Table, or DHT. On a high level, DHT is a distributed fault-tolerant "dictionary" that can be accessed by every participant. Each key-value pair is stored on a subset of peers determined by the $\mathrm{hash}$ function of the key.

Each participant has a unique identifier (ID) sampled uniformly from the $\mathrm{hash}$ function output range. When storing a $(key,\ value)$ pair, one must find $k$ peers whose IDs are nearest to $\mathrm{hash}(key)$ according to a chosen metric. After that, the participant requests each of those peers to store $(key,\ value)$. When retrieving a value for a key, one should compute $\mathrm{hash}(key)$, search for peers with IDs nearest to that $\mathrm{hash}$ value and request the value from those peers.

Specific DHT versions, such as Chord [124] or Kademlia [55], employ different hash types and algorithms for finding nearest peers. For instance, Kademlia DHT sorts peers based on the XOR distance function: $d(x, y) = \mathrm{int}(x \oplus y)$.

In DHT, each participant is directly aware of only a small subset of peers. When storing or retrieving a key, the participant requests additional peers from its neighbors in a semi-greedy search, minimizing the XOR distance until it finds $k$ nearest peers. In Kademlia, nodes form a special navigable graph structure that lets them find nearest peers in at most $\mathcal{O}(k + \log N)$ requests to other peers, where $N$ is the total number of participants. Due to their scalability and fault-tolerance, DHTs found numerous applications including BitTorrent, Ethereum, I2P and decentralized deep learning [36].

# C   Proofs of Mixing Properties of Moshpit All-Reduce

**Notation.** Throughout the following sections, we use the standard notation from the literature on stochastic optimization. That is, for any $n$-dimensional vectors $x = (x_1, \ldots, x_n)^\top, y = (y_1, \ldots, y_n)^\top \in \mathbb{R}^n$ we use $\langle x, y \rangle$ to denote the standard inner product: $\langle x, y \rangle = x_1 y_1 + \ldots + x_n y_n$. Next, we use $\|x\|$ to denote the $\ell_2$=norm of $x$ ($\|x\| = \sqrt{\langle x, x \rangle}$), $\mathbb{E}[\xi]$ to denote an expectation of a random variable $\xi$, $\mathbb{E}[\xi \mid \eta]$ is used for the conditional expectation of $\xi$ given $\eta$, and $\mathbb{P}\{E\}$ denotes the probability of an event $E$.

## C.1   Computing exact average in a full grid

As discussed in Section 3.1, Moshpit All-Reduce obtains the exact average of parameter vectors from $N$ peers arranged in a grid with $d$ coordinates and $M$ positions per coordinate when $N \equiv M^d$. That is, when the grid is full and each step averages $M$ parameter values along a single grid coordinate without repetitions, the algorithm needs only $d$ steps to compute the actual average across all nodes. In this section, we give a proof of this fact.

First, let us formally define the setting and the averaging steps of Moshpit All-Reduce in this specific case. Let $\theta_{i_1 i_2 \ldots i_d}$ be the parameter vector of the worker with coordinates $i_1, i_2, \ldots, i_d$; each coordinate $i_k$ takes values from 1 to $M$, because the hypercube of peers is completely full (thus, due to the pigeonhole principle, there are no unoccupied coordinates). Next, arrange the coordinates of these vector according to the order of averaging iterations: namely, at iteration 1

$$\overline{\theta}^1_{i_1 i_2 \ldots i_d} = \frac{1}{M} \sum_{j_1=1}^{M} \theta_{j_1 i_2 \ldots i_d}, \quad i_1 \in \{1, \ldots, M\}, \tag{13}$$

which means that for the first iteration, we take the average across the first axis $\overline{\theta}^1$ and replicate it across all $M$ resulting vectors regardless of their index $i_1$. The next averaging steps can be expressed similarly with a simple recurrence relation:

$$\overline{\theta}^t_{i_1 i_2 \ldots i_d} = \frac{1}{M} \sum_{j_t=1}^{M} \overline{\theta}^{t-1}_{i_1 \ldots i_{t-1} j_t i_{t+1} \ldots i_d}. \tag{14}$$

Given this formal definition, we can now state and prove the exact averaging result:

**Theorem C.1** (Exact average in a full $d$-dimensional hypercube after $d$ steps). *Assume that $M^d$ peers are arranged in a $d$-dimensional hypercube with $M$ positions in each dimension. Also, assume that each peer fully participates in every averaging step and $M$-sized groups for each averaging iteration are determined based on the hypercube coordinates. Then, if Moshpit All-Reduce is ran in the above setup for $d$ iterations without repeating groups (i.e. averaging across each dimension exactly once), its result for each participant is the average value of $\theta$ across all $M^d$ peers.*

*Proof.* We can directly obtain the expression for the average by expanding the recurrence and rearranging the sums:

$$
\begin{aligned}
\overline{\theta}^d_{i_1 i_2 \ldots i_d} &= \frac{1}{M} \sum_{j_d=1}^{M} \overline{\theta}^{d-1}_{i_1 \ldots i_{d-1} j_d} = \frac{1}{M} \sum_{j_d=1}^{M} \left( \frac{1}{M} \sum_{j_{d-1}=1}^{M} \overline{\theta}_{i_1 i_2 \ldots j_{d-1} j_d} \right) = \ldots \\
&= \frac{1}{M} \underbrace{\left( \sum_{j_d=1}^{M} \left( \frac{1}{M} \sum_{j_{d-1}=1}^{M} \cdots \sum_{j_2=1}^{M} \left( \frac{1}{M} \sum_{j_1=1}^{M} \theta_{j_1 \ldots j_d} \right) \right) \right)}_{d \text{ summations}} \\
&= \frac{1}{M^d} \sum_{j_d=1}^{M} \sum_{j_{d-1}=1}^{M} \cdots \sum_{j_2=1}^{M} \sum_{j_1=1}^{M} \theta_{j_1 \ldots j_d} = \frac{1}{M^d} \sum_{j_1, \ldots, j_d=1}^{M} \theta_{j_1 \ldots j_d}.
\end{aligned}
$$

But this is exactly the global average of all $\theta$, since there are $M^d$ participants and each vector is represented in the sum because of summation over all possible indices. $\square$

Notice that for a given grid of peers, if some of its indices do not have corresponding parameter vectors, Equation 14 may result in different average vectors on different workers due to different numbers of peers along a coordinate for different indices. For example, running two iterations of Moshpit Averaging with $d = 2$, $M = 2$ and three parameter vectors $\theta_{11}$, $\theta_{21}$, $\theta_{22}$ results in $\frac{\theta_{11}+\theta_{21}}{2}$ on the first worker and $\frac{\theta_{11}+\theta_{21}}{4} + \theta_{22}$ on other workers, with neither equal to the global average. However, the variance of the averaged vectors does decrease, which is formally proven in Section C.3.

## C.2 Proof of Theorem 3.1

Below we provide the complete proof of Theorem 3.1. For the readers' convenience, we restate the theorem.

**Theorem C.2** (Theorem 3.1). *If all workers have non-zero probability of successfully running a communication round in Moshpit Averaging and the order of* `peers`$_t$ *is random, then all local vectors* $\theta^t_i$ *converge to the global average with probability* 1:

$$
\forall i = 1, \ldots, N \quad \left\| \theta^t_i - \frac{1}{N} \sum_{i=1}^{N} \theta^0_i \right\|^2 \xrightarrow[t \to \infty]{} 0. \tag{15}
$$

*Proof of Theorem 3.1.* First of all, we notice that (15) is equivalent to

$$
\forall i = 1, \ldots, N, \ \forall j = 1, \ldots, n \quad \left( \theta^t_i(j) - \frac{1}{N} \sum_{i=1}^{N} \theta^0_i(j) \right)^2 \xrightarrow[t \to \infty]{} 0, \tag{16}
$$

where $\theta^t_i(j)$ denotes $j$-th component of $\theta^t_i$. Consider an arbitrary component $j \in \{1, \ldots, n\}$ and the sequence of intervals $\{I_{j,t}\}_{t \geq 0}$ where $I_{j,t} = \text{conv}\{\theta^t_1(j), \theta^t_2(j), \ldots, \theta^t_N(j)\}$. Then, $\{I_{j,t}\}_{t \geq 0}$ is a sequence of nested intervals ($I_{j,t+1} \subseteq I_{j,t} \forall t \geq 0$), since averaging in groups does not expand the convex hull of $\{\theta^t_1, \theta^t_2, \ldots, \theta^t_N\}$. For convenience, we specify the bounds of the intervals: $I_{j,t} = [a_{j,t}, b_{j,t}]$. Using the Cantor's intersection theorem, we conclude that

$$
\bigcap_{t=0}^{\infty} I_{j,t} = I_j = [a_j, b_j],
$$

where $\overline{\theta}(j) = \frac{1}{N} \sum_{i=1}^{n} \theta^0_i(j) \in [a_j, b_j]$. If $[a_j, b_j] = \{\overline{\theta}(j)\}$ with probability 1, then (16) holds with probability 1 as well. Suppose the opposite: there exist such $j \in \{1, \ldots, n\}$, $[a, b]$ and $\delta, \Delta > 0$ that $\overline{\theta}(j) \in [a, b]$, $b - a = \Delta$ and

$$
\mathbb{P} \underbrace{\left\{ [a, b] \subseteq \bigcap_{t=0}^{\infty} I_{j,t} \right\}}_{E} = \delta > 0 \quad \text{and} \quad \forall \varepsilon > 0 \ \mathbb{P} \underbrace{\left\{ [a - \varepsilon, b + \varepsilon] \subseteq \bigcap_{t=0}^{\infty} I_{j,t} \right\}}_{E_\varepsilon} < \delta.
$$

This implies that for all $\varepsilon > 0$ there exists such $T_\varepsilon > 0$ that

$$\mathbb{P}\left\{\underbrace{\forall t \geq T_\varepsilon \ a_{j,t} \in [a - \varepsilon, a], b_{j,t} \in [b, b + \varepsilon]}_{E'_\varepsilon}\right\} = \delta_\varepsilon > 0.$$

Consider $\varepsilon = \frac{\Delta}{(2N+100)^{2N}}$ and assume that the event $E'_\varepsilon$ holds. Next, we introduce new notation: $J^t_{\text{left}} = \{i \in \{1, \ldots, n\} \mid \theta^t_i(j) \in [a - \varepsilon, a]\}$ and $J^t_{\text{right}} = \{i \in \{1, \ldots, n\} \mid \theta^t_i(j) \in [b, b + \varepsilon]\}$. Since $E'_\varepsilon$ holds the sets $J^t_{\text{left}}$ and $J^t_{\text{right}}$ are non-empty for all $t \geq T_\varepsilon$ with probability $\delta_\varepsilon > 0$:

$$\mathbb{P}\left\{\forall t \geq T_\varepsilon \ J^t_{\text{left}} \neq \varnothing \text{ and } J^t_{\text{right}} \neq \varnothing\right\} = \delta_\varepsilon > 0. \tag{17}$$

We notice that every pair of workers $i_1, i_2$ has a non-zero probability of taking part in the averaging inside the common group at each iteration since all workers have a non-zero probability of successfully running a communication round and the order of $\texttt{peers}_t$ is random. This implies that every pair of workers $i_1, i_2$ with probability 1 take part in the averaging inside the common group infinitely many times when $t$ goes to the infinity.

Next, we choose some $t_0 \geq T_\varepsilon$. Let $J^{t_0}_{\text{left}} = \{i_{l,1}, \ldots, i_{l,q_l}\}$ and $J^{t_0}_{\text{right}} = \{i_{r,1}, \ldots, i_{r,q_r}\}$. Consider the event $E'_{\varepsilon,0} \subseteq E'_\varepsilon$ such that in $E'_{\varepsilon,0}$ peer $i_{l,1}$ computes an average in the group containing any peer from $J^{t_0}_{\text{right}}$ at some iteration $t_1 > t_0$. Our observations above imply that $\mathbb{P}\{E'_{\varepsilon,0}\} = \mathbb{P}\{E'_\varepsilon\} = \delta_\varepsilon > 0$. Then, $\theta^{t_1}_{i_{l,1}}(j) \geq \frac{N-1}{N}(a-\varepsilon) + \frac{1}{N}b = a - \varepsilon + \frac{1}{N}(\Delta + \varepsilon) = a - \frac{\Delta}{(2N+100)^{2N}} + \frac{1}{N}\left(\Delta + \frac{\Delta}{(2N+100)^{2N}}\right) > a + \frac{\Delta}{2N}$, i.e., $\theta^{t_1}_{i_{l,1}}(j) \in (a, b]$ meaning that $i_{l,1} \notin J^{t_1}_{\text{left}}$. The last part of the proof shows that for any $t \geq t_1$, the peer $i_{l,1}$ will never be the part of $J^t_{\text{left}}$ and after a finite number of iterations $J^t_{\text{left}} = \varnothing$ with probability $\delta_\varepsilon > 0$ when $E'_{\varepsilon,0}$ holds, implying the contradiction with (17).

To show that, we consider the following set of peers: $\widehat{J}^{t_1}_{\text{left}} = \{i \in \{1, \ldots, n\} \mid \exists t \geq t_1 : \theta^t_i(j) \in [a - \varepsilon, a + \frac{\Delta}{2N})\}$. Next, we consider the event $E'_{\varepsilon,1} \subseteq E'_{\varepsilon,0}$ such that in $E'_{\varepsilon,1}$ peer $i_{l,1}$ computes an average in the group containing some peer $i_{l,avg,1}$ from $\widehat{J}^{t_1}_{\text{left}}$ at some iteration $t_2 > t_1$ (and $t_2$ is the first such moment after $t_1$). Again, our observations imply $\mathbb{P}\{E'_{\varepsilon,1}\} = \mathbb{P}\{E'_{\varepsilon,0}\} = \delta_\varepsilon > 0$. Then, $\theta^{t_2}_{i_{l,1}}(j) = \theta^{t_2}_{i_{l,avg,1}}(j) > \frac{N-1}{N}(a - \varepsilon) + \frac{1}{N}\left(a + \frac{\Delta}{2N}\right) = a + \frac{\Delta}{2N^2} - \frac{(N-1)\Delta}{N(2N+100)^{2N}} > a + \frac{\Delta}{4N^2}$. After that, we consider the event $E'_{\varepsilon,2} \subseteq E'_{\varepsilon,1}$ such that in $E'_{\varepsilon,2}$ peer $i_{l,1}$ or $i_{l,avg,1}$ computes an average in the group containing a peer $i_{l,avg,2} \neq i_{l,avg,1}$ from $\widehat{J}^{t_1}_{\text{left}}$ at an iteration $t_3 > t_2$ (and $t_3$ is the first such moment after $t_2$). Then, $\theta^{t_3}_{i_{l,1}}(j), \theta^{t_3}_{i_{l,avg,1}}(j)$ and $\theta^{t_3}_{i_{l,avg,2}}(j)$ are greater than $\frac{N-1}{N}(a - \varepsilon) + \frac{1}{N}\left(a + \frac{\Delta}{4N^2}\right) = a + \frac{\Delta}{4N^3} - \frac{(N-1)\Delta}{N(2N+100)^{2N}} > a + \frac{\Delta}{8N^3}$.

Therefore, after at least $N - 1$ of such averaging iterations, with probability $\delta_\varepsilon$ all $\theta^t_i(j)$ will be greater than $a + \frac{\Delta}{(2N)^N} > a$ while $E'_\varepsilon$ holds. This contradicts (17). Therefore,

$$\bigcap_{t=0}^{\infty} I_{j,t} = \{\overline{\theta}(j)\}$$

with probability 1, which concludes the proof. $\qquad\square$

### C.3  Proof of Theorem 3.2

In this section, we provide the complete proof of Theorem 3.2. For convenience, we restate the theorem below.

**Theorem C.3** (Theorem 3.2, averaging convergence rate)**.** *Consider the modification of Moshpit All-Reduce that works as follows: at each iteration $k \geq 1$ 1) peers are randomly split into $r$ disjoint groups of sizes $M^k_1, \ldots, M^k_r$ in such a way that $\sum_{i=1}^r M^k_i = N$ and $M^k_i \geq 1 \ \forall i = 1, \ldots, r$ and 2) peers from each group compute their group average via All-Reduce. Let $\theta_1, \ldots, \theta_N$ be the input vectors of this procedure and $\theta^T_1, \ldots, \theta^T_N$ be the outputs after $T$ iterations. Then,*

$$\mathbb{E}\left[\frac{1}{N}\sum_{i=1}^N \|\theta^T_i - \overline{\theta}\|^2\right] = \left(\frac{r-1}{N} + \frac{r}{N^2}\right)^T \cdot \frac{1}{N}\sum_{i=1}^N \|\theta_i - \overline{\theta}\|^2, \tag{18}$$

*where $\overline{\theta} = \frac{1}{N}\sum_{i=1}^N \theta_i$.*

*Proof.* First of all, let us clarify the procedure of random splitting of peers in $r$ groups. We assume that at iteration $k$ of the modified algorithm we generate a random permutation $\pi^k = (\pi_1^k, \ldots, \pi_N^k)$ of $1, \ldots, N$. Next, $J_1^k = \{\pi_1^k, \ldots, \pi_{M_1^k}^k\}$ form the indices of the first group of workers, $J_2^k = \{\pi_{M_1^k+1}^k, \ldots, \pi_{M_2^k}^k\}$ are the indices of the second group, and $J_r^k = \{\pi_{M_1^k+M_2^k+\ldots+M_{r-1}^k+1}^k, \ldots, \pi_N^k\}$ are the indices of group $r$. In other words, we generate a random permutation and take contiguous subgroups of indices corresponding to predefined group sizes $M_i^k$, starting from the first group.

By definition, we have $\bigsqcup_{i=1}^r J_i^k = \{1, 2, \ldots, N\}$, where $\sqcup$ defines the disjoint union operator. Moreover, notice that group sizes $M_1^k, \ldots, M_r^k$ can depend on $k$ and even be random: for our analysis, it is sufficient that the randomness defining the permutation is independent from $M_1^k, \ldots, M_r^k$. Next, vectors $\theta_1^k, \ldots, \theta_N^k$ are obtained by the following formula:

$$\forall j = 1, \ldots, N, \quad \theta_j^k = \frac{1}{M_i^k} \sum_{t \in J_i^k} \theta_t^{k-1}, \quad \text{where } J_i^k \text{ is the group for which } j \in J_i^k.$$

Using this, we show that the average of vectors $\{\theta_i^k\}_{i=1}^n$ remains the same throughout the iterations of Moshpit All-Reduce:

$$\frac{1}{N} \sum_{j=1}^N \theta_j^k = \frac{1}{N} \sum_{i=1}^r M_i^k \cdot \frac{1}{M_i^k} \sum_{t \in J_i^k} \theta_t^{k-1} = \frac{1}{N} \sum_{i=1}^r \sum_{t \in J_i^k} \theta_t^{k-1} = \frac{1}{N} \sum_{j=1}^N \theta_j^{k-1}.$$

Therefore, the quantity $\frac{1}{N} \sum_{j=1}^N \|\theta_j^k - \overline{\theta}\|^2$ (average distortion) measures the quality of averaging. For this quantity, we can derive the following expression:

$$
\begin{aligned}
\frac{1}{N} \sum_{j=1}^N \|\theta_j^k - \overline{\theta}\|^2 &= \frac{1}{N} \sum_{i=1}^r M_i^k \left\| \frac{1}{M_i^k} \sum_{t \in J_i^k} \theta_t^{k-1} - \overline{\theta} \right\|^2 \\
&= \frac{1}{N} \sum_{i=1}^r \frac{1}{M_i^k} \left( \sum_{t \in J_i^k} \|\theta_t^{k-1} - \overline{\theta}\|^2 + 2 \sum_{t,l \in J_i^k, t<l} \langle \theta_t^{k-1} - \overline{\theta}, \theta_l^{k-1} - \overline{\theta} \rangle \right).
\end{aligned}
$$

Taking the expectation $\mathbb{E}_{\pi^k}[\cdot]$ with respect to the randomness coming from the choice of $\pi^k$ we get

$$
\begin{aligned}
&\mathbb{E}_{\pi^k} \left[ \frac{1}{N} \sum_{j=1}^N \|\theta_j^k - \overline{\theta}\|^2 \right] \\
&= \frac{1}{N} \sum_{i=1}^r \frac{1}{M_i^k} \left( \mathbb{E}_{\pi^k} \left[ \sum_{t \in J_i^k} \|\theta_t^{k-1} - \overline{\theta}\|^2 \right] + 2 \mathbb{E}_{\pi^k} \left[ \sum_{t,l \in J_i^k, t<l} \langle \theta_t^{k-1} - \overline{\theta}, \theta_l^{k-1} - \overline{\theta} \rangle \right] \right).
\end{aligned}
$$

Since $\forall j, j_1, j_2 \in \{1, \ldots, N\}, j_1 \neq j_2$ and for all $i = 1, \ldots, r$

$$\mathbb{P}\{j \in J_i^k\} = \frac{M_i^k}{N}, \quad \mathbb{P}\{j_1, j_2 \in J_i^k\} = \frac{M_i^k(M_i^k - 1)}{N^2},$$

we have

$$
\begin{aligned}
\mathbb{E}_{\pi^k}\left[\frac{1}{N}\sum_{j=1}^{N}\|\theta_j^k-\overline{\theta}\|^2\right] &= \frac{1}{N}\sum_{i=1}^{r}\frac{1}{N}\sum_{j=1}^{N}\|\theta_j^{k-1}-\overline{\theta}\|^2 \\
&\quad +\frac{1}{N}\sum_{i=1}^{r}2\frac{M_i^k-1}{N^2}\sum_{1\le j_1<j_2\le N}\langle\theta_{j_1}^{k-1}-\overline{\theta},\theta_{j_2}^{k-1}-\overline{\theta}\rangle \\
&= \frac{r}{N^2}\sum_{j=1}^{N}\|\theta_j^{k-1}-\overline{\theta}\|^2+2\frac{N-r}{N^3}\sum_{1\le j_1<j_2\le N}\langle\theta_{j_1}^{k-1}-\overline{\theta},\theta_{j_2}^{k-1}-\overline{\theta}\rangle \\
&= \left(\frac{r}{N^2}-\frac{N-r}{N^3}\right)\sum_{j=1}^{N}\|\theta_j^{k-1}-\overline{\theta}\|^2+\frac{N-r}{N^3}\sum_{j=1}^{N}\|\theta_j^{k-1}-\overline{\theta}\|^2 \\
&\quad +2\frac{N-r}{N^3}\sum_{1\le j_1<j_2\le N}\langle\theta_{j_1}^{k-1}-\overline{\theta},\theta_{j_2}^{k-1}-\overline{\theta}\rangle \\
&= \frac{N(r-1)+r}{N^3}\sum_{j=1}^{N}\|\theta_j^{k-1}-\overline{\theta}\|^2+\frac{N-r}{N^3}\underbrace{\left\|\sum_{j=1}^{N}(\theta_j^{k-1}-\overline{\theta})\right\|^2}_{\|N\overline{\theta}-N\overline{\theta}\|^2=0} \\
&= \left(\frac{r-1}{N}+\frac{r}{N^2}\right)\cdot\frac{1}{N}\sum_{j=1}^{N}\|\theta_j^{k-1}-\overline{\theta}\|^2.
\end{aligned}
$$

Finally, we take the full expectation from the both sides of the above equation and apply the tower property $\mathbb{E}\left[\mathbb{E}_{\pi^k}\left[\cdot\right]\right]=\mathbb{E}\left[\cdot\right]$:

$$
\mathbb{E}\left[\frac{1}{N}\sum_{j=1}^{N}\|\theta_j^k-\overline{\theta}\|^2\right]=\left(\frac{r-1}{N}+\frac{r}{N^2}\right)\mathbb{E}\left[\frac{1}{N}\sum_{j=1}^{N}\|\theta_j^{k-1}-\overline{\theta}\|^2\right].
$$

Unrolling the recurrence for $k=T$, we establish (18). $\qquad\square$

**Remark C.1.** *The result implies that increasing the group size $\alpha>1$ times implies almost $\alpha$ times faster convergence to the average.*

**Remark C.2.** *Our analysis can be easily generalized to the case when number of groups $r$ can depend on $k$ and be a random variable independent from the choice of permutations and the number of groups at previous steps. In this case, (18) transforms into*

$$
\mathbb{E}\left[\frac{1}{N}\sum_{i=1}^{N}\|\theta_i^T-\overline{\theta}\|^2\right]=\frac{1}{N}\sum_{i=1}^{N}\|\theta_i-\overline{\theta}\|^2\cdot\prod_{k=1}^{T}\left(\frac{\mathbb{E}[r_k]-1}{N}+\frac{\mathbb{E}[r_k]}{N^2}\right), \tag{19}
$$

*where $r_k$ is the number of groups at iteration $k$.*

### C.4 Additional Guarantees For Moshpit Averaging

In this section, we derive the result measuring the rate of variance reduction when averaging random vectors with Algorithm 1. We start with the following technical lemma:

**Lemma C.1.** *Let $\xi\sim Binom(M,p)$ have a binomial distribution with parameters $M$ (number of trials) and $p$ (probability of success for each trial). Then*

$$
m_1(M,p):=\mathbb{E}\left[\min\left\{\frac{1}{\xi},1\right\}\right]=(1-p)^M+\sum_{i=1}^{M}\frac{(1-p)^{M-i}-(1-p)^M}{i}, \tag{20}
$$

$$
m_2(M,p):=\mathbb{E}\left[\min\left\{\frac{1}{\xi^2},1\right\}\right]=(1-p)^M+\sum_{i=1}^{M}\frac{(1-p)^{M-i}-(1-p)^M}{i}\sum_{j=i}^{M}\frac{1}{j}. \tag{21}
$$

*Proof.* We start with the proof of (20). By definition of the expectation, we have

$$\mathbb{E}\left[\min\left\{\frac{1}{\xi},1\right\}\right] = (1-p)^M + \sum_{i=1}^{M}\frac{1}{i}p^i(1-p)^{M-i}\binom{M}{i}.$$

For simplicity of further derivations, we introduce the following notation: $m_1(M,p) = \mathbb{E}\left[\min\left\{\frac{1}{\xi},1\right\}\right]$ and $m_2(M,p) = \mathbb{E}\left[\min\left\{\frac{1}{\xi^2},1\right\}\right]$. Taking the derivative of $m_1(M,p)$ by $p$, we obtain

$$
\begin{aligned}
m_1'(M,p) &= -M(1-p)^{M-1} + \sum_{i=1}^{M}p^{i-1}(1-p)^{M-i}\binom{M}{i} \\
&\quad - \sum_{i=1}^{M}\frac{M-i}{i}p^i(1-p)^{M-i-1}\binom{M}{i} \\
&= -M(1-p)^{M-1} + \frac{1}{p}\left(-(1-p)^M + \sum_{i=0}^{M}p^i(1-p)^{M-i}\binom{M}{i}\right) \\
&\quad - \frac{M}{1-p}\sum_{i=1}^{M}\frac{1}{i}p^i(1-p)^{M-i}\binom{M}{i} \\
&\quad + \frac{1}{1-p}\left(-(1-p)^M + \sum_{i=0}^{M}p^i(1-p)^{M-i}\binom{M}{i}\right) \\
&= -M(1-p)^{M-1} + \frac{1}{p}\left(1-(1-p)^M\right) - \frac{M}{1-p}\left(m_1(M,p)-(1-p)^M\right) \\
&\quad + \frac{1}{1-p}\left(1-(1-p)^M\right) \\
&= \frac{1}{p(1-p)} - \frac{(1-p)^{M-1}}{p} - \frac{M}{1-p}m_1(M,p).
\end{aligned}
$$

Rearranging the terms, we get the following linear first-order ODE

$$m_1'(M,p) + \frac{M}{1-p}m_1(M,p) = \frac{1}{p(1-p)} - \frac{(1-p)^{M-1}}{p}. \tag{22}$$

To solve it, we consider the following homogeneous ODE:

$$m_1'(M,p) + \frac{M}{1-p}m_1(M,p) = 0.$$

The solution of this ODE is $m_1(M,p) = C(1-p)^M$, where $C \in \mathbb{R}$ is an arbitrary real constant. Next, we go back to the initial ODE (22) and try to find a solution of the form $m_1(M,p) = C(p)(1-p)^M$, where $C(p) : \mathbb{R} \to \mathbb{R}$ is a differentiable function:

$$
\begin{aligned}
\left(C(p)(1-p)^M\right)' + \frac{M}{1-p}C(p)(1-p)^M &= \frac{1}{p(1-p)} - \frac{(1-p)^{M-1}}{p} \\
&\Downarrow \\
C'(p)(1-p)^M &= \frac{1}{p(1-p)} - \frac{(1-p)^{M-1}}{p} \\
&\Downarrow \\
C'(p) &= \frac{1}{p(1-p)^{M+1}} - \frac{1}{p(1-p)}.
\end{aligned}
$$

Since

$$\frac{1}{x(1-x)^{k+1}} = \frac{1}{x(1-x)^k} + \frac{1}{(1-x)^{k+1}} \tag{23}$$

for all $x \notin \{0, 1\}$ and all non-negative integers $k$, we have

$$C'(p) = \frac{1}{p} + \frac{1}{1-p} + \frac{1}{(1-p)^2} + \ldots + \frac{1}{(1-p)^{M+1}} - \frac{1}{p} - \frac{1}{1-p}$$

$$\Downarrow$$

$$C'(p) = \sum_{i=1}^{M} (1-p)^{-i-1},$$

hence

$$C(p) = \hat{C} + \sum_{i=1}^{M} \frac{1}{i}(1-p)^{-i},$$

where $\hat{C}$ is a real constant. Putting all together, we obtain

$$m_1(M, p) = C(p)(1-p)^M = \hat{C}(1-p)^M + \sum_{i=1}^{M} \frac{1}{i}(1-p)^{M-i}.$$

Taking $m_1(M, 0) = 1$ into account, we conclude that $\hat{C} = 1 - \sum_{i=1}^{M} \frac{1}{i}$ and obtain (20).

Using a similar technique, we derive (21). By definition of the expectation, we have

$$m_2(M, p) = (1-p)^M + \sum_{i=1}^{M} \frac{1}{i^2} p^i (1-p)^{M-i} \binom{M}{i}.$$

Taking the derivative of $m_2(M, p)$ by $p$, we obtain

$$m_2'(M, p) = -M(1-p)^{M-1} + \sum_{i=1}^{M} \frac{1}{i} p^{i-1} (1-p)^{M-i} \binom{M}{i}$$

$$- \sum_{i=1}^{M} \frac{M-i}{i^2} p^i (1-p)^{M-i-1} \binom{M}{i}$$

$$= -M(1-p)^{M-1} + \frac{1}{p} \sum_{i=1}^{M} \frac{1}{i} p^i (1-p)^{M-i} \binom{M}{i}$$

$$- \frac{M}{1-p} \sum_{i=1}^{M} \frac{1}{i^2} p^i (1-p)^{M-i} \binom{M}{i} + \frac{1}{1-p} \sum_{i=1}^{M} \frac{1}{i} p^i (1-p)^{M-i} \binom{M}{i}$$

$$= -M(1-p)^{M-1} + \frac{1}{p} \left( m_1(M, p) - (1-p)^M \right)$$

$$+ \frac{1}{1-p} \left( -M m_2(M, p) + M(1-p)^M + m_1(M, p) - (1-p)^M \right)$$

$$= \frac{m_1(M, p)}{p(1-p)} - \frac{(1-p)^{M-1}}{p} - \frac{M}{1-p} m_2(M, p).$$

Rearranging the terms, we get the following linear first-order ODE

$$m_2'(M, p) + \frac{M}{1-p} m_2(M, p) = \frac{m_1(M, p)}{p(1-p)} - \frac{(1-p)^{M-1}}{p}. \tag{24}$$

To solve this ODE, we consider the homogeneous ODE:

$$m_2'(M, p) + \frac{M}{1-p} m_2(M, p) = 0.$$

The solution of this ODE is $m_2(M, p) = C(1-p)^M$, where $C \in \mathbb{R}$ is an arbitrary real constant. Next, we go back to the initial ODE (24) and try to find a solution of the form $m_2(M, p) = C(p)(1-p)^M$,

where $C(p): \mathbb{R} \to \mathbb{R}$ is a differentiable function:

$$\left(C(p)(1-p)^M\right)' + \frac{M}{1-p}C(p)(1-p)^M = \frac{m_1(M,p)}{p(1-p)} - \frac{(1-p)^{M-1}}{p}$$

$$\Downarrow$$

$$C'(p)(1-p)^M = \frac{m_1(M,p)}{p(1-p)} - \frac{(1-p)^{M-1}}{p}$$

$$\Downarrow$$

$$C'(p) = \frac{m_1(M,p)}{p(1-p)^{M+1}} - \frac{1}{p(1-p)}.$$

Using (23) and (20), we derive

$$
\begin{aligned}
C'(p) \overset{(20)}{=} & -\frac{\sum_{i=1}^{M}\frac{1}{i}}{p(1-p)} + \frac{\sum_{i=1}^{M}\frac{1}{i}(1-p)^{M-i}}{p(1-p)^{M+1}} \\
= & -\sum_{i=1}^{M}\frac{1}{ip(1-p)} + \sum_{i=1}^{M}\frac{1}{ip(1-p)^{i+1}} \\
\overset{(23)}{=} & -\sum_{i=1}^{M}\frac{1}{i}\left(\frac{1}{p} + \frac{1}{1-p}\right) \\
& + \sum_{i=1}^{M}\frac{1}{i}\left(\frac{1}{p} + \frac{1}{1-p} + \frac{1}{(1-p)^2} + \ldots + \frac{1}{(1-p)^{i+1}}\right) \\
= & \sum_{i=1}^{M}\frac{1}{i}\left(\frac{1}{(1-p)^2} + \ldots + \frac{1}{(1-p)^{i+1}}\right) = \sum_{i=1}^{M}\frac{1}{(1-p)^{i+1}}\sum_{j=i}^{M}\frac{1}{j},
\end{aligned}
$$

hence

$$C(p) = \hat{C} + \sum_{i=1}^{M}\frac{1}{i}(1-p)^{-i}\sum_{j=i}^{M}\frac{1}{j},$$

where $\hat{C}$ is a real constant. Putting all together, we obtain

$$m_2(M,p) = C(p)(1-p)^M = \hat{C}(1-p)^M + \sum_{i=1}^{M}\frac{1}{i}(1-p)^{M-i}\sum_{j=i}^{M}\frac{1}{j}.$$

Taking $m_2(M,0) = 1$ into account, we conclude that $\hat{C} = 1 - \sum_{i=1}^{M}\frac{1}{i}\sum_{j=i}^{M}\frac{1}{j}$ and obtain (21). $\quad\square$

Using this lemma, we derive the following result:

**Theorem C.4.** *Assume that peers participating in Moshpit Averaging have independent random vectors $\theta_1, \ldots, \theta_N$ with means $\bar{\theta}_1, \ldots, \bar{\theta}_N$ and variances bounded by $\sigma^2$ before the averaging. Let $\theta_1^T, \ldots, \theta_N^T$ be the outputs of Moshpit Averaging after $T$ iterations. Finally, we assume that each peer from the grid can be dropped out for the whole averaging process before averaging independently from other peers, i.e., $N \sim Binom(M^d, p)$. Then, for all $i = 1, \ldots, N$ we have*

$$\mathbb{E}\left[\left\|\theta_i^T - \mathbb{E}_\theta\left[\theta_i^T\right]\right\|^2\right] \leq M^{T-1}\sigma^2 m_1(M-1,p)\left(m_2(M-1,p)\right)^{T-1}, \tag{25}$$

*where functions $m_1(M,p)$ and $m_2(M,p)$ are defined in (20) and (21) respectively, and $\mathbb{E}_\theta\left[\cdot\right]$ denotes the expectation w.r.t. the randomness from $\theta_1, \ldots, \theta_N$. Moreover, if $p \geq \frac{2}{3}$ and $M \geq 11$, then $m_1(M-1,p) \leq \frac{2}{M}$, $m_2(M-1,p) \leq \frac{3}{M^2}$ and*

$$\mathbb{E}\left[\left\|\theta_i^T - \mathbb{E}_\theta\left[\theta_i^T\right]\right\|^2\right] \leq \frac{2\sigma^2}{M(M/3)^{T-1}}. \tag{26}$$

*Proof.* First of all, we recall an equivalent formulation of Moshpit Averaging. Consider a hypercube $\{1, \ldots, M\}^d$. One can consider the elements of this hypercube as hyperindices and assign a unique hyperindex to each peer so that peers can be viewed as vertices in the hypercube. Then, during the $k$-th iteration of Moshpit All-Reduce, each worker computes the average among those peers that have hyperindices with the same values except the $k$-th index; in other words, peers compute averages along the $k$-th dimension of the hypercube. Next, if $N = 0$, we assume that $\theta_i^T = \mathbb{E}_\theta\left[\theta_i^T\right]$ and (25) holds for free. Therefore, to derive (25), we assume that $N > 0$.

More formally, we use the following notation: $\theta_{C_i} = \theta_i$ for all $i = 1, \ldots, N$, where $C_i = (c_1^i, c_2^i, \ldots, c_d^i)$, $c_j^i \in \{1, \ldots, M\}$ for all $j = 1, \ldots, M$, and $C_i \neq C_k$ for $i \neq k$. Let $\mathcal{C}$ be the set of hyperindices corresponding to all peers. Next, we use $\theta_{C_i}^t$ to define the vector stored on $i$-th peer after $t$ iterations of Moshpit Averaging. Then, for all $i = 1, \ldots, N$ we have $\theta_{C_i}^0 = \theta_{C_i}$ and for all $t = 1, \ldots, d$

$$\theta_{C_i}^t = \frac{1}{b_{i,t}} \sum_{k \in J_{i,t}} \theta_{C_k}^{t-1},$$

where $J_{i,t} = \{k \in N \mid C_k = (c_1^k, \ldots, c_d^k) \in \mathcal{C} \text{ and } c_j^k = c_j^i \ \forall j \neq t\}$ and $b_{i,t} = |J_{i,t}|$. Using this, we derive the following formula for $\theta_{C_i}^t$:

$$\theta_i^T \equiv \theta_{C_i}^T = \frac{1}{b_{i,T}} \sum_{i_1 \in J_{i,T}} \frac{1}{b_{i_1,T-1}} \sum_{i_2 \in J_{i_1,T-1}} \frac{1}{b_{i_2,T-2}} \sum_{i_3 \in J_{i_2,T-1}} \cdots \frac{1}{b_{i_{T-1},1}} \sum_{i_T \in J_{i_{T-1},1}} \theta_{i_T}.$$

Taking the expectation w.r.t. $\theta_1, \ldots, \theta_N$, we get

$$\mathbb{E}_\theta\left[\theta_i^T\right] = \frac{1}{b_{i,T}} \sum_{i_1 \in J_{i,T}} \frac{1}{b_{i_1,T-1}} \sum_{i_2 \in J_{i_1,T-1}} \frac{1}{b_{i_2,T-2}} \sum_{i_3 \in J_{i_2,T-1}} \cdots \frac{1}{b_{i_{T-1},1}} \sum_{i_T \in J_{i_{T-1},1}} \overline{\theta}_{i_T}.$$

Using the independence of $\theta_1, \ldots, \theta_N$, we derive

$$
\begin{aligned}
\mathbb{E}_\theta\left[\left\|\theta_i^T - \mathbb{E}_\theta\left[\theta_i^T\right]\right\|^2\right] &= \mathbb{E}_\theta\left[\left\|\sum_{i_1 \in J_{i,T}} \sum_{i_2 \in J_{i_1,T-1}} \cdots \sum_{i_T \in J_{i_{T-1},1}} \frac{\theta_{i_T} - \overline{\theta}_{i_T}}{b_{i,T} b_{i_1,T-1} \ldots b_{i_{T-1},1}}\right\|^2\right] \\
&= \sum_{i_1 \in J_{i,T}} \sum_{i_2 \in J_{i_1,T-1}} \cdots \sum_{i_T \in J_{i_{T-1},1}} \frac{\mathbb{E}_\theta\left[\left\|\theta_{i_T} - \overline{\theta}_{i_T}\right\|^2\right]}{b_{i,T}^2 b_{i_1,T-1}^2 \ldots b_{i_{T-1},1}^2} \\
&\leq \sum_{i_1 \in J_{i,T}} \sum_{i_2 \in J_{i_1,T-1}} \cdots \sum_{i_T \in J_{i_{T-1},1}} \frac{\sigma^2}{b_{i,T}^2 b_{i_1,T-1}^2 \ldots b_{i_{T-1},1}^2} \\
&= \sum_{i_1 \in J_{i,T}} \sum_{i_2 \in J_{i_1,T-1}} \cdots \sum_{i_{T-1} \in J_{i_{T-2},2}} \frac{\sigma^2}{b_{i,T}^2 b_{i_1,T-1}^2 \ldots b_{i_{T-2},2}^2 b_{i_{T-1},1}}.
\end{aligned}
$$

Next, taking the full expectation from the both sides of the previous inequality and using the tower property, we obtain

$$\mathbb{E}\left[\left\|\theta_i^T - \mathbb{E}_\theta\left[\theta_i^T\right]\right\|^2\right] \leq \mathbb{E}\left[\sum_{i_1 \in J_{i,T}} \sum_{i_2 \in J_{i_1,T-1}} \cdots \sum_{i_{T-1} \in J_{i_{T-2},2}} \frac{\sigma^2}{b_{i,T}^2 b_{i_1,T-1}^2 \ldots b_{i_{T-2},2}^2 b_{i_{T-1},1}}\right]. \quad (27)$$

Notice that $J_{i_k,T-k} \cap J_{i_{k+1},T-k-1} = \{i_{k+1}\}$ for all $k = 0, \ldots, T-1$, where $i_0 = i$. Moreover, for $k_1, k_2 \in \{0, 1, \ldots, T\}$, $k_1 < k_2$ either $J_{i_{k_1},T-k_1} \cap J_{i_{k_2},T-k_2} = \{k_2\}$ or $J_{i_{k_1},T-k_1} \cap J_{i_{k_2},T-k_2} = \varnothing$. The first situation is possible iff $i_{k_1} = i_{k_1+1} = \ldots i_{k_2-1}$.

Taking these observations about sets $J_{i_k,T-k}$ into account, we consider the sets $J'_{i_k,T-k} = J_{i_k,T-k} \setminus \{i_k\}$ for $k = 0, 1, \ldots, T-1$. These sets are pairwise disjoint and their cardinalities $b'_{i_k,T-k} = |J'_{i_k,T-k}|$ satisfy the following relations: $b_{i_k,T-k} = 1 + b'_{i_k,T-k} \geq \max\{1, b'_{i_k,T-k}\} =: \hat{b}_{i_k,T-k}$ for $k = 1, 2, \ldots, T-1$. Moreover, $b'_{i,T}, b'_{i_1,T-1}, \ldots, b'_{i_{T-1},1}$ are independent random variables from the binomial distribution $\text{Binom}(M-1, p)$. Finally, we notice that the number of terms in (27) is upper-bounded by $M^{T-1}$, since $|J_{i,t}| \leq M$ for all $i = 1, \ldots, N$ and $t = 0, \ldots, T$.

Putting all together, we obtain

$$
\begin{aligned}
\mathbb{E}\left[\left\|\theta_i^T - \mathbb{E}_\theta\left[\theta_i^T\right]\right\|^2\right] &\leq \mathbb{E}\left[\sum_{i_1 \in J_{i,T}} \sum_{i_2 \in J_{i_1,T-1}} \cdots \sum_{i_{T-1} \in J_{i_{T-2},2}} \frac{\sigma^2}{\hat{b}_{i,T}^2 \hat{b}_{i_1,T-1}^2 \cdots \hat{b}_{i_{T-2},2}^2 \hat{b}_{i_{T-1},1}}\right] \\
&\leq M^{T-1}\sigma^2 \mathbb{E}\left[\frac{1}{\hat{\xi}_1^2 \hat{\xi}_2^2 \cdots \hat{\xi}_{T-1}^2 \hat{\xi}_T}\right] \\
&= M^{T-1}\sigma^2 \mathbb{E}\left[\frac{1}{\hat{\xi}_1^2}\right] \mathbb{E}\left[\frac{1}{\hat{\xi}_2^2}\right] \cdots \mathbb{E}\left[\frac{1}{\hat{\xi}_{T-1}^2}\right] \mathbb{E}\left[\frac{1}{\hat{\xi}_T}\right],
\end{aligned}
$$

where $\hat{\xi}_k^2 = \max\{1, \xi_k^2\}$ for $k = 1, \ldots, T$ and $\xi_1, \ldots, \xi_T$ are i.i.d. random variables having the binomial distribution $\mathrm{Binom}(M-1, p)$. Then one can simplify the inequality above using Lemma C.1 and get

$$
\mathbb{E}\left[\left\|\theta_i^T - \mathbb{E}_\theta\left[\theta_i^T\right]\right\|^2\right] \leq M^{T-1}\sigma^2 m_1(M-1, p)\left(m_2(M-1, p)\right)^{T-1},
$$

where functions $m_1(M, p)$ and $m_2(M, p)$ are defined in (20) and (21) respectively.

Next, we simplify the obtained upper bound under the assumption that $M$ and $p$ are not too small; specifically, $M \geq 11$ and $p \geq 2/3$. From (20), we have

$$
\begin{aligned}
m_1(M-1, p) &= (1-p)^{M-1} + \sum_{i=1}^{M-1} \frac{1}{i}\left((1-p)^{M-1-i} - (1-p)^{M-1}\right) \\
&\leq (1-p)^{M-1} \sum_{i=1}^{M-1} \frac{1}{i(1-p)^i}.
\end{aligned}
$$

Since

$$
\frac{1}{(k+1)(1-p)^{k+1}} \cdot \frac{k(1-p)^k}{1} = \frac{k}{(k+1)(1-p)} \xrightarrow{k\to\infty} \frac{1}{1-p} \geq 3,
$$

we have

$$
(1-p)^{M-1} \sum_{i=1}^{M-1} \frac{1}{i(1-p)^i} = \Theta\left((1-p)^M \cdot \frac{1}{M(1-p)^M}\right) = \Theta\left(\frac{1}{M}\right).
$$

Using simple algebra, one can prove that for $M \geq 11$ and $p \geq 2/3$ the following inequality holds:

$$
m_1(M-1, p) \leq (1-p)^{M-1} \sum_{i=1}^{M-1} \frac{1}{i(1-p)^i} \leq \frac{2}{M}.
$$

Similarly, we analyze $m_2(M-1, p)$:

$$
\begin{aligned}
m_2(M-1, p) &= (1-p)^{M-1} + \sum_{i=1}^{M-1} \frac{1}{i}\left((1-p)^{M-1-i} - (1-p)^{M-1}\right) \sum_{j=i}^{M-1} \frac{1}{j} \\
&\leq (1-p)^{M-1} \sum_{i=1}^{M-1} \frac{1}{i(1-p)^i} \sum_{j=i}^{M-1} \frac{1}{j}.
\end{aligned}
$$

Since

$$
\begin{aligned}
\frac{\frac{1}{k(1-p)^k} \sum_{j=k}^{M-1} \frac{1}{j}}{\frac{1}{(k-1)(1-p)^{k-1}} \sum_{j=k-1}^{M-1} \frac{1}{j}} &= \frac{(k-1) \sum_{j=k}^{M-1} \frac{1}{j}}{k(1-p)\left(\frac{1}{k-1} + \sum_{j=k}^{M-1} \frac{1}{j}\right)} \geq \frac{3(k-1) \cdot \frac{1}{k}}{k\left(\frac{1}{k-1} + \frac{1}{k}\right)} \\
&= \frac{3(k-1)^2}{k(2k-1)} \xrightarrow{k\to\infty} \frac{3}{2},
\end{aligned}
$$

we have

$$(1-p)^{M-1} \sum_{i=1}^{M-1} \frac{1}{i(1-p)^i} \sum_{j=i}^{M-1} \frac{1}{j} = \Theta\left((1-p)^M \cdot \frac{1}{M^2(1-p)^M}\right) = \Theta\left(\frac{1}{M^2}\right).$$

Next, one can prove with simple algebra that for $M \geq 11$ and $p \geq 2/3$ the following inequality holds:

$$m_2(M-1,p) \leq (1-p)^{M-1} \sum_{i=1}^{M-1} \frac{1}{i(1-p)^i} \sum_{j=i}^{M-1} \frac{1}{j} \leq \frac{3}{M^2}.$$

Plugging the obtained upper bounds for $m_1(M-1,p)$ and $m_2(M-1,p)$ in (25), we obtain (26). $\quad\square$

# D  Convergence Proofs of Moshpit SGD

In this section, we provide the complete statements of the theorems establishing the convergence of Moshpit SGD together with the full proofs. First, we introduce all necessary definitions, basic inequalities and auxiliary lemmas; then we prove the convergence in strongly convex and convex cases; lastly, we provide the proofs for the non-convex case.

## D.1  Definitions, Basic Facts and Auxiliary Results

Below we provide several classical definitions and results which are used in our proofs.

### D.1.1  Standard Definitions from Optimization Theory

**Definition D.1** ($L$-smoothness). *A function $f : \mathbb{R}^n \to \mathbb{R}$ is called $L$-smooth if for all $x, y \in \mathbb{R}^n$, the following inequality holds:*

$$\|\nabla f(x) - \nabla f(y)\| \leq L\|x - y\|. \tag{28}$$

If the function $f$ is $L$-smooth, then for all $x, y \in \mathbb{R}^n$

$$f(y) \leq f(x) + \langle \nabla f(x), y - x \rangle + \frac{L}{2}\|y - x\|^2. \tag{29}$$

Next, if $f$ is additionally convex and $x^*$ is its minimizer, then for all $x \in \mathbb{R}^d$

$$\|\nabla f(x)\|^2 \leq 2L\left(f(x) - f(x^*)\right). \tag{30}$$

**Definition D.2** ($\mu$-strong convexity). *A differentiable function $f : \mathbb{R}^n \to \mathbb{R}$ is called $\mu$-strongly convex if there exists a constant $\mu \geq 0$ such that for all $x, y \in \mathbb{R}^n$*

$$f(y) \geq f(x) + \langle \nabla f(x), y - x \rangle + \frac{\mu}{2}\|y - x\|^2. \tag{31}$$

### D.1.2  Basic Facts

For all $a, b, \theta_1, \ldots, \theta_N \in \mathbb{R}^n$ and $\alpha > 0$, the following inequalities hold:

$$\|a + b\|^2 \quad \leq \quad 2\|a\|^2 + 2\|b\|^2, \tag{32}$$

$$\left\|\frac{1}{N}\sum_{i=1}^{N} \theta_i\right\|^2 \quad \leq \quad \frac{1}{N}\sum_{i=1}^{N} \|\theta_i\|^2, \tag{33}$$

$$\langle a, b \rangle \quad \leq \quad \frac{\|a\|^2}{2\alpha} + \frac{\alpha\|b\|^2}{2}. \tag{34}$$

### D.1.3  Properties of Expectation

**Variance decomposition.** For a random vector $\eta \in \mathbb{R}^d$ and any deterministic vector $x \in \mathbb{R}^d$, the variance satisfies

$$\mathbb{E}\left[\|\eta - \mathbb{E}\eta\|^2\right] = \mathbb{E}\left[\|\eta - x\|^2\right] - \|\mathbb{E}\eta - x\|^2 \tag{35}$$

**Tower property of expectation.** For any random variables $\xi, \eta \in \mathbb{R}^d$ we have

$$\mathbb{E}[\xi] = \mathbb{E}[\mathbb{E}[\xi \mid \eta]] \tag{36}$$

under the assumption that $\mathbb{E}[\xi]$ and $\mathbb{E}[\mathbb{E}[\xi \mid \eta]]$ are well-defined.

### D.1.4 Auxiliary Results

For the readers' convenience, we list all auxiliary results that we use in our proofs below. The first result is classical and establishes that the gradient descent step is a contractive operator.

**Lemma D.1** (Lemma 6 from [59]). *For any $L$-smooth and $\mu$-strongly convex function $f : \mathbb{R}^n \to \mathbb{R}$, points $x, y \in \mathbb{R}^n$, and stepsize $\gamma \in (0, 1/L]$, the following inequality holds:*

$$\|x - \gamma \nabla f(x) - y + \gamma \nabla f(y)\|^2 \le (1 - \gamma\mu)\|x - y\|^2. \tag{37}$$

The next two lemmas are useful for estimating typical recurrences appearing in the analysis.

**Lemma D.2** (Lemma I.2 from [60]). *Let $\{r_k\}_{k \ge 0}$ satisfy*

$$r_K \le \frac{a}{\gamma W_K} + c_1 \gamma + c_2 \gamma^2$$

*for all $K \ge 0$ with some constants $a, c_2 \ge 0$, $c_1 \ge 0$, where $w_k = (1 - \gamma\mu(1 - \delta_{pv,1}))^{-(k+1)}$, $W_K = \sum_{k=0}^{K} w_k$, $\mu > 0$, $\delta_{pv,1} \in [0, 1)$ and $\gamma \le \gamma_0$ for some $\gamma_0 > 0$, $\gamma_0 \le 1/\mu(1-\delta_{pv,1})$. Then, for all $K$ such that*

$$either \; \frac{\ln\left(\max\left\{2, \min\left\{a\mu^2(1-\delta_{pv,1})^2 K^2/c_1, \; a\mu^3(1-\delta_{pv,1})^3 K^3/c_2\right\}\right\}\right)}{K} \le 1$$

$$or \; \gamma_0 \le \frac{\ln\left(\max\left\{2, \min\left\{a\mu^2(1-\delta_{pv,1})^2 K^2/c_1, \; a\mu^3(1-\delta_{pv,1})^3 K^3/c_2\right\}\right\}\right)}{(1 - \delta_{pv,1})\mu K}$$

*and*

$$\gamma = \min\left\{\gamma_0, \; \frac{\ln\left(\max\left\{2, \min\left\{a\mu^2(1-\delta_{pv,1})^2 K^2/c_1, \; a\mu^3(1-\delta_{pv,1})^3 K^3/c_2\right\}\right\}\right)}{(1 - \delta_{pv,1})\mu K}\right\}$$

*we have that*

$$r_K = \widetilde{\mathcal{O}}\left(\frac{a}{\gamma_0}\exp\left(-\gamma_0\mu(1 - \delta_{pv,1})K\right) + \frac{c_1}{(1 - \delta_{pv,1})\mu K} + \frac{c_2}{(1 - \delta_{pv,1})^2\mu^2 K^2}\right).$$

**Lemma D.3** (Lemma I.3 from [60]). *Let $\{r_k\}_{k \ge 0}$ satisfy*

$$r_K \le \frac{a}{\gamma K} + c_1 \gamma + c_2 \gamma^2$$

*for all $K \ge 0$ with some constants $a, c_2 \ge 0$, $c_1 \ge 0$ where $\gamma \le \gamma_0$ for some $\gamma_0 > 0$. Then for all $K$ and*

$$\gamma = \min\left\{\gamma_0, \; \sqrt{\frac{a}{c_1 K}}, \; \sqrt[3]{\frac{a}{c_2 K}}\right\}$$

*we have that*

$$r_K = \mathcal{O}\left(\frac{a}{\gamma_0 K} + \sqrt{\frac{ac_1}{K}} + \frac{\sqrt[3]{a^2 c_2}}{K^{2/3}}\right).$$

Finally, the lemma below is useful for our convergence analysis in the non-convex case.

**Lemma D.4** (Lemma I.1 from [60]). *For any $\tau$ random vectors $\xi_1, \ldots, \xi_\tau \in \mathbb{R}^d$ such that $\forall t = 2, \ldots, \tau$ the random vector $\xi_t$ depends on $\xi_1, \ldots, \xi_{t-1}$ and does not depend on $\xi_{t+1}, \ldots, \xi_\tau$ the following inequality holds*

$$\mathbb{E}\left[\left\|\sum_{t=1}^{\tau} \xi_t\right\|^2\right] \le e\tau \sum_{t=1}^{\tau} \mathbb{E}\left[\|\mathbb{E}_t[\xi_t]\|^2\right] + e\sum_{t=1}^{\tau} \mathbb{E}\left[\|\xi_t - \mathbb{E}_t[\xi_t]\|^2\right], \tag{38}$$

*where $\mathbb{E}_t[\cdot]$ denotes the conditional expectation $\mathbb{E}[\cdot \mid \xi_{t-1}, \ldots, \xi_1]$.*

## D.2 Convex Case

In this section, we give the full proof of Theorem 3.3 about the convergence of Moshpit SGD for convex and strongly convex problems. The scheme of the proof follows the similar steps as in the state-of-the-art analysis of Local-SGD [61, 62, 60]. We start with the following lemma:

**Lemma D.5.** *Let $f_1 = \ldots = f_N = f$, function $f$ be $\mu$-strongly convex (Def. D.2) and $L$-smooth (see Def. D.1), and Assumptions 3.1 and 3.2 hold with $\Delta_{pv}^k = \delta_{pv,1}\gamma\mu\mathbb{E}[\|\theta^k - \theta^*\|^2] + \gamma^2\delta_{pv,2}^2$ and $\widetilde{\theta} = \theta^*$, where $\theta^* \in \operatorname{argmin}_{\theta \in \mathbb{R}^n} f(\theta)$ and $\delta_{pv,1} \in [0,1)$, $\delta_{pv,2} \geq 0$. Then, for any $k \geq 0$ the iterates produced by Moshpit SGD with $\gamma \leq 1/4L$ satisfy*

$$
\begin{aligned}
\gamma\mathbb{E}\left[f(\theta^k) - f(\theta^*)\right] &\leq (1 - \gamma\mu(1-\delta_{pv,1}))\mathbb{E}\left[\|\theta^k - \theta^*\|^2\right] - \mathbb{E}\left[\|\theta^{k+1} - \theta^*\|^2\right] \\
&\quad + \frac{3L\gamma}{2}\mathbb{E}[V_k] + \gamma^2\left(\frac{\sigma^2}{N_{\min}} + \delta_{pv,2}^2\right),
\end{aligned}
\tag{39}
$$

*where $V_k = \frac{1}{N_k}\sum_{i \in P_k}\|\theta_i^k - \theta^k\|^2$ and $\theta^k = \frac{1}{N_k}\sum_{i \in P_k}\theta_i^k$.*

*Proof.* Recall that Assumption 3.2 with $\Delta_{pv}^k = \delta_{pv,1}\gamma\mu\mathbb{E}[\|\theta^k - \theta^*\|^2] + \gamma^2\delta_{pv,2}^2$ and $\widetilde{\theta} = \theta^*$ states

$$
\mathbb{E}\left[\langle\theta^{k+1} - \widehat{\theta}^{k+1}, \theta^{k+1} + \widehat{\theta}^{k+1} - 2\theta^*\rangle\right] \leq \delta_{pv,1}\gamma\mu\mathbb{E}[\|\theta^k - \theta^*\|^2] + \gamma^2\delta_{pv,2}^2,
\tag{40}
$$

where $\widehat{\theta}^{k+1} = \frac{1}{N_k}\sum_{i \in P_k}(\theta_i^k - \gamma g_i^k)$. Next, the definition of $\widehat{\theta}^{k+1}$ implies

$$
\widehat{\theta}^{k+1} = \frac{1}{N_k}\sum_{i \in P_k}\theta_i^k - \frac{\gamma}{N_k}\sum_{i \in P_k}g_i^k = \theta^k - \gamma g^k,
$$

where $g^k = \frac{1}{N_k}\sum_{i \in P_k}g_i^k$. Using this, we derive

$$
\begin{aligned}
\|\theta^{k+1} - \theta^*\|^2 &= \|\widehat{\theta}^{k+1} - \theta^*\|^2 + 2\langle\theta^{k+1} - \widehat{\theta}^{k+1}, \widehat{\theta}^{k+1} - \theta^*\rangle + \|\theta^{k+1} - \widehat{\theta}^{k+1}\|^2 \\
&= \|\theta^k - \theta^* - \gamma g^k\|^2 + \langle\theta^{k+1} - \widehat{\theta}^{k+1}, \theta^{k+1} + \widehat{\theta}^{k+1} - 2\theta^*\rangle \\
&= \|\theta^k - \theta^*\|^2 - 2\gamma\langle\theta^k - \theta^*, g^k\rangle + \gamma^2\|g^k\|^2 \\
&\quad + \langle\theta^{k+1} - \widehat{\theta}^{k+1}, \theta^{k+1} + \widehat{\theta}^{k+1} - 2\theta^*\rangle.
\end{aligned}
$$

Taking the conditional expectation $\mathbb{E}\left[\cdot \mid \theta^k\right] := \mathbb{E}\left[\cdot \mid P_k, \theta_i^k, i \in P_k\right]$ from the both sides of the previous equation and using Assumption 3.1, we obtain

$$
\begin{aligned}
\mathbb{E}\left[\|\theta^{k+1} - \theta^*\|^2 \mid \theta^k\right] &= \|\theta^k - \theta^*\|^2 - 2\gamma\left\langle\theta^k - \theta^*, \frac{1}{N_k}\sum_{i \in P_k}\nabla f(\theta_i^k)\right\rangle \\
&\quad + \gamma^2\mathbb{E}\left[\left\|\frac{1}{N_k}\sum_{i \in P_k}g_i^k\right\|^2 \mid \theta^k\right] \\
&\quad + \mathbb{E}\left[\langle\theta^{k+1} - \widehat{\theta}^{k+1}, \theta^{k+1} + \widehat{\theta}^{k+1} - 2\theta^*\rangle \mid \theta^k\right].
\end{aligned}
\tag{41}
$$

Next, we estimate the second and the third terms in the right-hand side of (41). First,

$$
\begin{aligned}
-2\gamma\left\langle\theta^k - \theta^*, \frac{1}{N_k}\sum_{i \in P_k}\nabla f(\theta_i^k)\right\rangle &= \frac{2\gamma}{N_k}\sum_{i \in P_k}\left(\langle\theta^* - \theta_i^k, \nabla f(\theta_i^k)\rangle + \langle\theta_i^k - \theta^k, \nabla f(\theta_i^k)\rangle\right) \\
&\overset{(31),(29)}{\leq} \frac{2\gamma}{N_k}\sum_{i \in P_k}\left(f(\theta^*) - f(\theta_i^k) - \frac{\mu}{2}\|\theta_i^k - \theta^*\|^2\right) \\
&\quad + \frac{2\gamma}{N_k}\sum_{i \in P_k}\left(f(\theta_i^k) - f(\theta^k) + \frac{L}{2}\|\theta_i^k - \theta^k\|^2\right) \\
&\overset{(33)}{\leq} 2\gamma\left(f(\theta^*) - f(\theta^k)\right) - \gamma\mu\|\theta^k - \theta^*\|^2 + L\gamma V_k,
\end{aligned}
\tag{42}
$$

where $V_k = \frac{1}{N_k} \sum_{i \in P_k} \|\theta_i^k - \theta^k\|^2$. Secondly, since stochastic gradients $\{g_i^k\}_{i \in P_k}$ are computed independently, we get

$$\gamma^2 \mathbb{E}\left[\left\|\frac{1}{N_k} \sum_{i \in P_k} g_i^k\right\|^2 \mid \theta^k\right] \overset{(35)}{=} \gamma^2 \left\|\frac{1}{N_k} \sum_{i \in P_k} \nabla f(\theta_i^k)\right\|^2$$

$$+ \gamma^2 \mathbb{E}\left[\left\|\frac{1}{N_k} \sum_{i \in P_k} (g_i^k - \nabla f(\theta_i^k))\right\|^2 \mid \theta^k\right]$$

$$\overset{(33)}{\leq} 2\gamma^2 \left\|\frac{1}{N_k} \sum_{i \in P_k} (\nabla f(\theta_i^k) - \nabla f(\theta^k))\right\|^2 + 2\gamma^2 \|\nabla f(\theta^k)\|^2$$

$$+ \frac{\gamma^2}{N_k^2} \sum_{i \in P_k} \mathbb{E}\left[\|g_i^k - \nabla f(\theta_i^k)\|^2 \mid \theta^k\right]$$

$$\overset{(33),(30),(7)}{\leq} \frac{2\gamma^2}{N_k} \sum_{i \in P_k} \|\nabla f(\theta_i^k) - \nabla f(\theta^k)\|^2$$

$$+ 4L\gamma^2 \left(f(\theta^k) - f(\theta^*)\right) + \frac{\gamma^2 \sigma^2}{N_k}$$

$$\overset{(28)}{\leq} \underbrace{\frac{2L^2\gamma^2}{N_k} \sum_{i \in P_k} \|\theta_i^k - \theta^k\|^2}_{2L^2\gamma^2 V_k}$$

$$+ 4L\gamma^2 \left(f(\theta^k) - f(\theta^*)\right) + \frac{\gamma^2 \sigma^2}{N_{\min}}. \qquad (43)$$

Plugging (42) and (43) in (41), we obtain

$$\mathbb{E}\left[\|\theta^{k+1} - \theta^*\|^2 \mid \theta^k\right] \leq (1 - \gamma\mu)\|\theta^k - \theta^*\|^2 - 2\gamma(1 - 2L\gamma)\left(f(\theta^k) - f(\theta^*)\right)$$

$$+ L\gamma(1 + 2L\gamma) V_k + \frac{\gamma^2 \sigma^2}{N_{\min}}$$

$$+ \mathbb{E}\left[\langle \theta^{k+1} - \widehat{\theta}^{k+1}, \theta^{k+1} + \widehat{\theta}^{k+1} - 2\theta^* \rangle \mid \theta^k\right],$$

and

$$\mathbb{E}\left[\|\theta^{k+1} - \theta^*\|^2\right] \overset{(40)}{\leq} (1 - \gamma\mu(1 - \delta_{pv,1}))\mathbb{E}\left[\|\theta^k - \theta^*\|^2\right] - 2\gamma(1 - 2L\gamma)\mathbb{E}\left[f(\theta^k) - f(\theta^*)\right]$$

$$+ L\gamma(1 + 2L\gamma)\mathbb{E}[V_k] + \gamma^2\left(\frac{\sigma^2}{N_{\min}} + \delta_{pv,2}^2\right)$$

$$\leq (1 - \gamma\mu(1 - \delta_{pv,1}))\mathbb{E}\left[\|\theta^k - \theta^*\|^2\right] - \gamma\mathbb{E}\left[f(\theta^k) - f(\theta^*)\right]$$

$$+ \frac{3L\gamma}{2}\mathbb{E}[V_k] + \gamma^2\left(\frac{\sigma^2}{N_{\min}} + \delta_{pv,2}^2\right),$$

where in the last inequality we use $\gamma \leq 1/4L$. $\qquad \square$

Next, we estimate the term $\mathbb{E}[V_k]$ measuring the expected dissimilarity between local iterates and their global average at iteration $k$.

**Lemma D.6.** *Let $f_1 = \ldots = f_N = f$, function $f$ be $\mu$-strongly convex (Def. D.2) and $L$-smooth (see Def. D.1), and Assumptions 3.1 and 3.2 hold with $\Delta_{pv}^k = \delta_{pv,1}\gamma\mu\mathbb{E}[\|\theta^k - \theta^*\|^2] + \gamma^2\delta_{pv,2}^2$ and $\widetilde{\theta} = \theta^*$, where $\theta^* \in \operatorname{argmin}_{\theta \in \mathbb{R}^n} f(\theta)$ and $\delta_{pv,1} \in [0,1)$, $\delta_{pv,2} \geq 0$. Then, for any $k \geq 0$ the iterates produced by Moshpit SGD with $\gamma \leq 1/4L$ satisfy*

$$\mathbb{E}[V_k] \leq 2\gamma^2\left(4\delta_{aq}^2 + (\tau - 1)\sigma^2\right), \qquad (44)$$

*where $V_k = \frac{1}{N_k} \sum_{i \in P_k} \|\theta_i^k - \theta^k\|^2$ and $\theta^k = \frac{1}{N_k} \sum_{i \in P_k} \theta_i^k$.*

*Proof.* First of all, if $k = a\tau$ for some integer $a \geq 0$, then (44) follows from Assumption 3.2 (eq. (10)). Therefore, we consider such $k$ that $k = a\tau + t'$ for some $t' \in (0, \tau)$. Then, for any $i, j \in P_k, i \neq j$

$$
\begin{aligned}
\mathbb{E}\left[\|\theta_i^k - \theta_j^k\|^2 \mid \theta^{k-1}\right] &= \mathbb{E}\left[\|\theta_i^{k-1} - \gamma g_i^{k-1} - \theta_j^{k-1} + \gamma g_j^{k-1}\|^2 \mid \theta^{k-1}\right] \\
&\stackrel{(35)}{=} \|\theta_i^{k-1} - \gamma \nabla f(\theta_i^{k-1}) - \theta_j^{k-1} + \gamma \nabla f(\theta_j^{k-1})\|^2 \\
&\quad + \gamma^2 \mathbb{E}\left[\|g_i^{k-1} - \nabla f(\theta_i^{k-1}) + g_j^{k-1} - \nabla f(\theta_j^{k-1})\|^2 \mid \theta^{k-1}\right].
\end{aligned}
$$

Using Lemma D.1 and independence of $g_i^{k-1}$ and $g_j^{k-1}$ for given $\theta_i^{k-1}, \theta_j^{k-1}, i \neq j$ we derive

$$
\begin{aligned}
\mathbb{E}\left[\|\theta_i^k - \theta_j^k\|^2 \mid \theta^{k-1}\right] &\stackrel{(37)}{\leq} (1 - \gamma\mu)\|\theta_i^{k-1} - \theta_j^{k-1}\|^2 + \gamma^2 \mathbb{E}\left[\|g_i^{k-1} - \nabla f(\theta_i^{k-1})\|^2 \mid \theta^{k-1}\right] \\
&\quad + \gamma^2 \mathbb{E}\left[\|g_j^{k-1} - \nabla f(\theta_j^{k-1})\|^2 \mid \theta^{k-1}\right] \\
&\stackrel{(7)}{\leq} (1 - \gamma\mu)\|\theta_i^{k-1} - \theta_j^{k-1}\|^2 + 2\gamma^2\sigma^2,
\end{aligned}
$$

from which we get the following:

$$
\mathbb{E}_g\left[\|\theta_i^k - \theta_j^k\|^2\right] \leq (1 - \gamma\mu)\mathbb{E}_g\left[\|\theta_i^{k-1} - \theta_j^{k-1}\|^2\right] + 2\gamma^2\sigma^2 \leq \mathbb{E}_g\left[\|\theta_i^{k-1} - \theta_j^{k-1}\|^2\right] + 2\gamma^2\sigma^2.
$$

Here, $\mathbb{E}_g[\cdot]$ denotes the expectation conditioned on $\{P_k\}_{k=a\tau}^{(a+1)\tau-1}$. Unrolling the recurrence, we get

$$
\begin{aligned}
\mathbb{E}_g\left[\|\theta_i^k - \theta_j^k\|^2\right] &\leq \mathbb{E}_g\left[\|\theta_i^{a\tau} - \theta_j^{a\tau}\|^2\right] + 2(k - a\tau)\gamma^2\sigma^2 \\
&\leq \mathbb{E}_g\left[\|\theta_i^{a\tau} - \theta_j^{a\tau}\|^2\right] + 2(\tau - 1)\gamma^2\sigma^2. \tag{45}
\end{aligned}
$$

Using this, we estimate $\mathbb{E}_g[V_k]$:

$$
\begin{aligned}
\mathbb{E}_g[V_k] &= \frac{1}{N_k} \sum_{i \in P_k} \mathbb{E}_g\left[\left\|\theta_i^k - \frac{1}{N_k} \sum_{j \in P_k} \theta_j^k\right\|^2\right] \stackrel{(33)}{\leq} \frac{1}{N_k^2} \sum_{i,j \in P_k} \mathbb{E}_g\left[\|\theta_i^k - \theta_j^k\|^2\right] \\
&\stackrel{(45)}{\leq} \frac{1}{N_k^2} \sum_{i,j \in P_k} \mathbb{E}_g\left[\|\theta_i^{a\tau} - \theta_j^{a\tau}\|^2\right] + 2(\tau - 1)\gamma^2\sigma^2 \\
&\stackrel{(32)}{\leq} \frac{2}{N_k^2} \sum_{i,j \in P_k} \left(\mathbb{E}_g\left[\|\theta_i^{a\tau} - \theta^{a\tau}\|^2\right] + \mathbb{E}_g\left[\|\theta_j^{a\tau} - \theta^{a\tau}\|^2\right]\right) + 2(\tau - 1)\gamma^2\sigma^2 \\
&= \frac{4}{N_k} \sum_{i \in P_k} \mathbb{E}_g\left[\|\theta_i^{a\tau} - \theta^{a\tau}\|^2\right] + 2(\tau - 1)\gamma^2\sigma^2 \\
&\leq \frac{4}{N_{a\tau}} \cdot \frac{N_{a\tau}}{N_k} \sum_{i \in P_{a\tau}} \mathbb{E}_g\left[\|\theta_i^{a\tau} - \theta^{a\tau}\|^2\right] + 2(\tau - 1)\gamma^2\sigma^2 \\
&\leq \mathbb{E}_g\left[\frac{8}{N_{a\tau}} \sum_{i \in P_{a\tau}} \|\theta_i^{a\tau} - \theta^{a\tau}\|^2\right] + 2(\tau - 1)\gamma^2\sigma^2,
\end{aligned}
$$

where in the last inequality we use $2N_{(a+1)\tau} = 2|P_{(a+1)\tau}| \geq |P_{a\tau}| = N_{a\tau}$ and $|N_k| \leq |N_{k-1}|$ following from Assumption 3.2. Finally, we take the full expectation from the previous inequality:

$$
\mathbb{E}[V_k] \stackrel{(36)}{\leq} 8\mathbb{E}\left[\frac{1}{N_{a\tau}} \sum_{i \in P_{a\tau}} \|\theta_i^{a\tau} - \theta^{a\tau}\|^2\right] + 2(\tau - 1)\gamma^2\sigma^2 \stackrel{(10)}{\leq} 2\gamma^2\left(4\delta_{aq}^2 + (\tau - 1)\sigma^2\right).
$$

This finishes the proof. $\qquad\square$

Combining Lemmas D.5 and D.6, we get the following result:

**Theorem D.1** (Theorem 3.3, convergence in the convex case). *Let $f_1 = \ldots = f_N = f$ be $\mu$-strongly convex (Def. D.2) and $L$-smooth (see Def. D.1), and Assumptions 3.1 and 3.2 hold with*

$\Delta_{pv}^k = \delta_{pv,1}\gamma\mu\mathbb{E}[\|\theta^k - \theta^*\|^2] + \gamma^2\delta_{pv,2}^2$ *and* $\widetilde{\theta} = \theta^*$*, where* $\theta^* \in \arg\min_{\theta \in \mathbb{R}^n} f(\theta)$ *and* $\delta_{pv,1} \in [0, 1)$*,* $\delta_{pv,2} \geq 0$*. Then, for any* $K \geq 0$*, the iterates produced by Moshpit SGD with* $\gamma \leq 1/4L$ *satisfy*

$$
\begin{aligned}
\mathbb{E}\left[f(\overline{\theta}^K) - f(\theta^*)\right] &\leq (1 - \gamma\mu(1 - \delta_{pv,1}))^K \frac{R_0^2}{\gamma} \\
&\quad + \gamma\left(\frac{\sigma^2}{N_{\min}} + \delta_{pv,2}^2 + 3L\gamma\left(4\delta_{aq}^2 + (\tau - 1)\sigma^2\right)\right),
\end{aligned} \tag{46}
$$

*when* $\mu > 0$*, and*

$$
\mathbb{E}\left[f(\overline{\theta}^K) - f(\theta^*)\right] \leq \frac{R_0^2}{\gamma K} + \gamma\left(\frac{\sigma^2}{N_{\min}} + \delta_{pv,2}^2 + 3L\gamma\left(4\delta_{aq}^2 + (\tau - 1)\sigma^2\right)\right), \tag{47}
$$

*when* $\mu = 0$*, where* $R_0 = \|\theta^0 - \theta^*\|$*,* $\overline{\theta}^K = \frac{1}{W_K}\sum_{k=0}^K w_k\theta^k = \frac{1}{W_K}\sum_{k=0}^K \frac{w_k}{N_k}\sum_{i \in P_k}\theta_i^k$*,* $w_k = (1 - \gamma\mu(1 - \delta_{pv,1}))^{-(k+1)}$*, and* $W_K = \sum_{k=0}^K w_k$*. That is, Moshpit SGD achieves* $\mathbb{E}[f(\overline{\theta}^K) - f(\theta^*)] \leq \varepsilon$ *after*

$$
K = \widetilde{\mathcal{O}}\left(\frac{L}{(1 - \delta_{pv,1})\mu} + \frac{\sigma^2}{N_{\min}(1 - \delta_{pv,1})\mu\varepsilon} + \frac{\delta_{pv,2}^2}{(1 - \delta_{pv,1})\mu\varepsilon} + \sqrt{\frac{L((\tau - 1)\sigma^2 + \delta_{aq}^2)}{(1 - \delta_{pv,1})^2\mu^2\varepsilon}}\right) \tag{48}
$$

*iterations with*

$$
\gamma = \min\left\{\frac{1}{4L}, \frac{\ln\left(\max\left\{2, \min\left\{\frac{R_0^2\mu^2(1-\delta_{pv,1})^2 K^2}{(\delta_{pv,2}^2 + \sigma^2/N_{\min})}, \frac{R_0^2\mu^3(1-\delta_{pv,1})^3 K^3}{3L(4\delta_{aq}^2 + (\tau-1)\sigma^2)}\right\}\right\}\right)}{(1 - \delta_{pv,1})\mu K}\right\}
$$

*when* $\mu > 0$*, and after*

$$
K = \mathcal{O}\left(\frac{LR_0^2}{\varepsilon} + \frac{R_0^2\sigma^2}{N_{\min}\varepsilon^2} + \frac{R_0^2\delta_{pv,2}^2}{\varepsilon^2} + \frac{R_0^2\sqrt{L((\tau - 1)\sigma^2 + \delta_{aq}^2)}}{\varepsilon^{3/2}}\right) \tag{49}
$$

*iterations with*

$$
\gamma = \min\left\{\frac{1}{4L}\sqrt{\frac{R_0}{(\delta_{pv,2}^2 + \sigma^2/N_{\min})K}}, \sqrt[3]{\frac{R_0^2}{3L(4\delta_{aq}^2 + (\tau - 1)\sigma^2)K}}\right\}
$$

*when* $\mu = 0$*.*

*Proof.* Plugging the result of Lemma D.6 in inequality (39) from Lemma D.5, we obtain

$$
\begin{aligned}
\gamma\mathbb{E}\left[f(\theta^k) - f(\theta^*)\right] &\leq (1 - \gamma\mu(1 - \delta_{pv,1}))\mathbb{E}\left[\|\theta^k - \theta^*\|^2\right] - \mathbb{E}\left[\|\theta^{k+1} - \theta^*\|^2\right] \\
&\quad + 3L\gamma^3\left(4\delta_{aq}^2 + (\tau - 1)\sigma^2\right) + \gamma^2\left(\frac{\sigma^2}{N_{\min}} + \delta_{pv,2}^2\right).
\end{aligned}
$$

Next, we sum up these inequalities for $k = 0, \ldots, K$ with weights $w_k = (1 - \gamma\mu(1 - \delta_{pv,1}))^{-(k+1)}$ and divide both sides by $\gamma W_K$, where $W_K = \sum_{k=0}^{K} w_k$:

$$
\frac{1}{W_K} \sum_{k=0}^{K} w_k \mathbb{E}\left[f(\theta^k) - f(\theta^*)\right] \leq \frac{1}{\gamma W_K} \sum_{k=0}^{K} (1 - \gamma\mu(1 - \delta_{pv,1})) w_k \mathbb{E}\left[\|\theta^k - \theta^*\|^2\right]
$$

$$
- \frac{1}{\gamma W_K} \sum_{k=0}^{K} w_k \mathbb{E}\left[\|\theta^{k+1} - \theta^*\|^2\right]
$$

$$
+ \gamma\left(\frac{\sigma^2}{N_{\min}} + \delta_{pv,2}^2 + 3L\gamma\left(4\delta_{aq}^2 + (\tau - 1)\sigma^2\right)\right)
$$

$$
= \frac{1}{\gamma W_K} \sum_{k=0}^{K} \left(w_{k-1} \mathbb{E}\left[\|\theta^k - \theta^*\|^2\right] - w_k \mathbb{E}\left[\|\theta^{k+1} - \theta^*\|^2\right]\right)
$$

$$
+ \gamma\left(\frac{\sigma^2}{N_{\min}} + \delta_{pv,2}^2 + 3L\gamma\left(4\delta_{aq}^2 + (\tau - 1)\sigma^2\right)\right)
$$

$$
= \frac{w_{-1}\|\theta^0 - \theta^*\|^2 - w_K \mathbb{E}\left[\|\theta^{K+1} - \theta^*\|^2\right]}{\gamma W_K}
$$

$$
+ \gamma\left(\frac{\sigma^2}{N_{\min}} + \delta_{pv,2}^2 + 3L\gamma\left(4\delta_{aq}^2 + (\tau - 1)\sigma^2\right)\right)
$$

$$
\leq \frac{\|\theta^0 - \theta^*\|^2}{\gamma W_K}
$$

$$
+ \gamma\left(\frac{\sigma^2}{N_{\min}} + \delta_{pv,2}^2 + 3L\gamma\left(4\delta_{aq}^2 + (\tau - 1)\sigma^2\right)\right).
$$

Since $f$ is convex, we apply the Jensen's inquality

$$
f\left(\frac{1}{W_K} \sum_{k=0}^{K} w_k \theta^k\right) \leq \frac{1}{W_K} \sum_{k=0}^{K} w_k f(\theta^k)
$$

to the previous result and get

$$
\mathbb{E}\left[f(\overline{\theta}^K) - f(\theta^*)\right] \leq \frac{R_0^2}{\gamma W_K} + \gamma\left(\frac{\sigma^2}{N_{\min}} + \delta_{pv,2}^2 + 3L\gamma\left(4\delta_{aq}^2 + (\tau - 1)\sigma^2\right)\right),
$$

where $R_0 = \|\theta^0 - \theta^*\|$ and $\overline{\theta}^K = \frac{1}{W_K} \sum_{k=0}^{K} w_k \theta^k = \frac{1}{W_K} \sum_{k=0}^{K} \frac{w_k}{N_k} \sum_{i \in P_k} \theta_i^k$. If $\mu > 0$, then $W_K \geq w_K \geq (1 - \gamma\mu(1 - \delta_{pv,1}))^{-K}$, implying (46). Next, $w_k = 1$ and $W_K = K$ when $\mu = 0$ gives (47). It remains to estimate the total number of iterations $K$ required by Moshpit SGD to find an $\varepsilon$-solution, i.e., to achieve $\mathbb{E}[f(\overline{\theta}^K) - f(\theta^*)] \leq \varepsilon$. Applying Lemma D.2 to (46), we get the following result: if $\mu > 0$ and

$$
\gamma = \min\left\{\frac{1}{4L}, \frac{\ln\left(\max\left\{2, \min\left\{\frac{R_0^2\mu^2(1-\delta_{pv,1})^2K^2}{\delta_{pv,2}^2 + \sigma^2/N_{\min}}, \frac{R_0^2\mu^3(1-\delta_{pv,1})^3K^3}{3L\left(4\delta_{aq}^2 + (\tau-1)\sigma^2\right)}\right\}\right\}\right)}{(1 - \delta_{pv,1})\mu K}\right\},
$$

then $\mathbb{E}\left[f(\overline{\theta}^K) - f(\theta^*)\right]$ equals

$$
\widetilde{\mathcal{O}}\left(LR_0^2 \exp\left(-\frac{\mu}{L}(1 - \delta_{pv,1})K\right) + \frac{\delta_{pv,2}^2 + \sigma^2/N_{\min}}{(1 - \delta_{pv,1})\mu K} + \frac{L\left(\delta_{aq}^2 + (\tau - 1)\sigma^2\right)}{(1 - \delta_{pv,1})^2\mu^2 K^2}\right),
$$

implying (48). Similarly, we apply Lemma D.3 to (47) and get that for $\mu = 0$ and

$$
\gamma = \min\left\{\frac{1}{4L}\sqrt{\frac{R_0}{(\delta_{pv,2}^2 + \sigma^2/N_{\min})K}}, \sqrt[3]{\frac{R_0^2}{3L\left(4\delta_{aq}^2 + (\tau - 1)\sigma^2\right)K}}\right\},
$$

$$\mathbb{E}\left[f(\overline{\theta}^K) - f(\theta^*)\right] = \mathcal{O}\left(\frac{LR_0^2}{K} + \sqrt{\frac{R_0^2(\delta_{pv,2}^2 + \sigma^2/N_{\min})}{K}} + \frac{\sqrt[3]{R_0^4 L\left(\delta_{aq}^2 + (\tau-1)\sigma^2\right)}}{K^{2/3}}\right),$$

implying (49). □

### D.3 Non-Convex Case

In this section, we give the full proof of Theorem 3.4 about convergence of Moshpit SGD for general non-convex problems. The proof follows the similar steps as in the state-of-the-art analysis of Local-SGD in non-convex case [64, 63]. We start with the following lemma:

**Lemma D.7.** *Let $f_1 = \ldots = f_N = f$, function $f$ be $L$-smooth and bounded from below by $f_*$, and Assumptions 3.1 and 3.2 hold with $\Delta_{pv}^k = \delta_{pv,1}\gamma\mathbb{E}[\|\nabla f(\theta^k)\|^2] + L\gamma^2\delta_{pv,2}^2$, $\delta_{pv,1} \in [0, 1/2)$, $\delta_{pv,2} \geq 0$. Then, for any $K \geq 0$ the iterates produced by Moshpit SGD with $\gamma \leq (1-2\delta_{pv,1})/8L$ satisfy*

$$\frac{(1-2\delta_{pv,1})\gamma}{4}\sum_{k=0}^{K-1}\mathbb{E}\left[\|\nabla f(\theta^k)\|^2\right] \leq f(\theta^0) - f_* + \gamma L^2\sum_{k=0}^{K-1}\mathbb{E}[V_k]$$

$$+KL\gamma^2\left(\frac{\sigma^2}{N_{\min}} + \delta_{pv,2}^2\right), \qquad (50)$$

*where $V_k = \frac{1}{N_k}\sum_{i\in P_k}\|\theta_i^k - \theta^k\|^2$ and $\theta^k = \frac{1}{N_k}\sum_{i\in P_k}\theta_i^k$.*

*Proof.* Recall that Assumption 3.2 with $\Delta_{pv}^k = \delta_{pv,1}\gamma\mathbb{E}[\|\nabla f(\theta^k)\|^2] + L\gamma^2\delta_{pv,2}^2$ states

$$\mathbb{E}\left[\langle\nabla f(\theta^k), \theta^{k+1} - \widehat{\theta}^{k+1}\rangle + L\|\widehat{\theta}^{k+1} - \theta^{k+1}\|^2\right] \leq \delta_{pv,1}\gamma\mathbb{E}[\|\nabla f(\theta^k)\|^2] + L\gamma^2\delta_{pv,2}^2, \qquad (51)$$

where $\widehat{\theta}^{k+1} = \frac{1}{N_k}\sum_{i\in P_k}(\theta_i^k - \gamma g_i^k)$. As for the convex case, the definition of $\widehat{\theta}^{k+1}$ implies

$$\widehat{\theta}^{k+1} = \frac{1}{N_k}\sum_{i\in P_k}\theta_i^k - \frac{\gamma}{N_k}\sum_{i\in P_k}g_i^k = \theta^k - \gamma g^k,$$

where $g^k = \frac{1}{N_k}\sum_{i\in P_k}g_i^k$. Using this and $L$-smoothness of $f$, we derive

$$f(\theta^{k+1}) - f(\theta^k) \overset{(29)}{\leq} \langle\nabla f(\theta^k), \theta^{k+1} - \theta^k\rangle + \frac{L}{2}\|\theta^{k+1} - \theta^k\|^2$$

$$\overset{(32)}{\leq} \langle\nabla f(\theta^k), \widehat{\theta}^{k+1} - \theta^k\rangle + \langle\nabla f(\theta^k), \theta^{k+1} - \widehat{\theta}^{k+1}\rangle$$

$$+L\|\widehat{\theta}^{k+1} - \theta^k\|^2 + L\|\theta^{k+1} - \widehat{\theta}^{k+1}\|^2$$

$$= -\gamma\langle\nabla f(\theta^k), g^k\rangle + L\gamma^2\|g^k\|^2 + \langle\nabla f(\theta^k), \theta^{k+1} - \widehat{\theta}^{k+1}\rangle$$

$$+L\|\theta^{k+1} - \widehat{\theta}^{k+1}\|^2,$$

from which it follows that

$$\mathbb{E}\left[f(\theta^{k+1}) - f(\theta^k) \mid \theta^k\right] \leq -\gamma\left\langle\nabla f(\theta^k), \frac{1}{N_k}\sum_{i\in P_k}\nabla f(\theta_i^k)\right\rangle$$

$$+\mathbb{E}\left[\langle\nabla f(\theta^k), \theta^{k+1} - \widehat{\theta}^{k+1}\rangle \mid \theta^k\right]$$

$$+\mathbb{E}\left[L\|\theta^{k+1} - \widehat{\theta}^{k+1}\|^2 \mid \theta^k\right]$$

$$+L\gamma^2\mathbb{E}\left[\left\|\frac{1}{N_k}\sum_{i\in P_k}g_i^k\right\|^2 \mid \theta^k\right], \qquad (52)$$

where $\mathbb{E}\left[\,\cdot\mid\theta^k\right] := \mathbb{E}\left[\,\cdot\mid P_k, \theta_i^k, i \in P_k\right]$. Next, we estimate the last three terms in the right-hand side of (52). First of all,

$$
\begin{aligned}
-\gamma\left\langle \nabla f(\theta^k), \frac{1}{N_k}\sum_{i\in P_k}\nabla f(\theta_i^k)\right\rangle &= -\gamma\|\nabla f(\theta^k)\|^2 \\
&\quad -\gamma\left\langle \nabla f(\theta^k), \frac{1}{N_k}\sum_{i\in P_k}\nabla f(\theta_i^k) - \nabla f(\theta^k)\right\rangle \\
&\overset{(34)}{\leq} -\gamma\|\nabla f(\theta^k)\|^2 + \frac{\gamma}{2}\|\nabla f(\theta^k)\|^2 \\
&\quad + \frac{\gamma}{2}\left\|\frac{1}{N_k}\sum_{i\in P_k}(\nabla f(\theta_i^k) - \nabla f(\theta^k))\right\|^2 \\
&\overset{(33)}{\leq} -\frac{\gamma}{2}\|\nabla f(\theta^k)\|^2 + \frac{\gamma}{2N_k}\sum_{i\in P_k}\|\nabla f(\theta_i^k) - \nabla f(\theta^k)\|^2 \\
&\overset{(28)}{\leq} -\frac{\gamma}{2}\|\nabla f(\theta^k)\|^2 + \frac{\gamma L^2}{2}V_k,
\end{aligned}
\tag{53}
$$

where $V_k = \frac{1}{N_k}\sum_{i\in P_k}\|\theta_i^k - \theta^k\|^2$. Secondly, since the stochastic gradients $\{g_i^k\}_{i\in P_k}$ are computed independently, we derive

$$
\begin{aligned}
L\gamma^2\mathbb{E}\left[\left\|\frac{1}{N_k}\sum_{i\in P_k}g_i^k\right\|^2 \mid \theta^k\right] &\overset{(35)}{=} L\gamma^2\left\|\frac{1}{N_k}\sum_{i\in P_k}\nabla f(\theta_i^k)\right\|^2 \\
&\quad + L\gamma^2\mathbb{E}\left[\left\|\frac{1}{N_k}\sum_{i\in P_k}(g_i^k - \nabla f(\theta_i^k))\right\|^2 \mid \theta^k\right] \\
&\overset{(33)}{\leq} 2L\gamma^2\left\|\frac{1}{N_k}\sum_{i\in P_k}(\nabla f(\theta_i^k) - \nabla f(\theta^k))\right\|^2 \\
&\quad + 2L\gamma^2\|\nabla f(\theta^k)\|^2 \\
&\quad + \frac{\gamma^2 L}{N_k^2}\sum_{i\in P_k}\mathbb{E}\left[\|g_i^k - \nabla f(\theta_i^k)\|^2 \mid \theta^k\right] \\
&\overset{(33),(7)}{\leq} \frac{2\gamma^2 L}{N_k}\sum_{i\in P_k}\|\nabla f(\theta_i^k) - \nabla f(\theta^k)\|^2 \\
&\quad + 2L\gamma^2\|\nabla f(\theta^k)\|^2 + \frac{\gamma^2 L\sigma^2}{N_k} \\
&\overset{(28)}{\leq} \underbrace{\frac{2L^3\gamma^2}{N_k}\sum_{i\in P_k}\|\theta_i^k - \theta^k\|^2}_{2L^3\gamma^2 V_k} + 2L\gamma^2\|\nabla f(\theta^k)\|^2 \\
&\quad + \frac{\gamma^2 L\sigma^2}{N_{\min}}.
\end{aligned}
\tag{54}
$$

Plugging (53) and (54) in (52), we obtain

$$
\begin{aligned}
\mathbb{E}\left[f(\theta^{k+1}) - f(\theta^k) \mid \theta^k\right] &\leq -\frac{\gamma}{2}(1 - 4L\gamma)\|\nabla f(\theta^k)\|^2 + \frac{\gamma L^2}{2}(1 + 4L\gamma)V_k + \frac{L\gamma^2\sigma^2}{N_{\min}} \\
&\quad + \mathbb{E}\left[\langle\nabla f(\theta^k), \theta^{k+1} - \widehat{\theta}^{k+1}\rangle + L\|\theta^{k+1} - \widehat{\theta}^{k+1}\|^2 \mid \theta^k\right].
\end{aligned}
$$

Next, we take the full expectation from the both sides of the above inequality, apply the tower property (36) and take into account that $\gamma \le (1-2\delta_{pv,1})/8L$:

$$
\begin{aligned}
\mathbb{E}\left[f(\theta^{k+1}) - f(\theta^k)\right] &\le -\frac{\gamma}{2}\left(1 - 4L\gamma\right)\mathbb{E}\left[\|\nabla f(\theta^k)\|^2\right] + \frac{\gamma L^2}{2}\left(1 + 4L\gamma\right)\mathbb{E}[V_k] + \frac{L\gamma^2\sigma^2}{N_{\min}} \\
&\quad + \mathbb{E}\left[\langle\nabla f(\theta^k), \theta^{k+1} - \widehat{\theta}^{k+1}\rangle + L\|\theta^{k+1} - \widehat{\theta}^{k+1}\|^2\right] \\
&\overset{(51)}{\le} -\frac{\gamma}{2}\left(1 - 2\delta_{pv,1} - 4L\gamma\right)\mathbb{E}\left[\|\nabla f(\theta^k)\|^2\right] + \frac{\gamma L^2}{2}\left(1 + 4L\gamma\right)\mathbb{E}[V_k] \\
&\quad + L\gamma^2\left(\frac{\sigma^2}{N_{\min}} + \delta_{pv,2}^2\right) \\
&\le -\frac{(1 - 2\delta_{pv,1})\gamma}{4}\mathbb{E}\left[\|\nabla f(\theta^k)\|^2\right] + \gamma L^2\mathbb{E}[V_k] \\
&\quad + L\gamma^2\left(\frac{\sigma^2}{N_{\min}} + \delta_{pv,2}^2\right).
\end{aligned}
$$

Summing up the obtained inequalities for $k = 0, \ldots, K-1$ and rearranging the terms, we derive

$$
\begin{aligned}
\frac{(1 - 2\delta_{pv,1})\gamma}{4}\sum_{k=0}^{K-1}\mathbb{E}\left[\|\nabla f(\theta^k)\|^2\right] &\le \sum_{k=0}^{K-1}\mathbb{E}\left[f(\theta^k) - f(\theta^{k+1})\right] + \gamma L^2\sum_{k=0}^{K-1}\mathbb{E}[V_k] \\
&\quad + KL\gamma^2\left(\frac{\sigma^2}{N_{\min}} + \delta_{pv,2}^2\right) \\
&= f(\theta^0) - \mathbb{E}[f(\theta^K)] + \gamma L^2\sum_{k=0}^{K-1}\mathbb{E}[V_k] \\
&\quad + KL\gamma^2\left(\frac{\sigma^2}{N_{\min}} + \delta_{pv,2}^2\right) \\
&\le f(\theta^0) - f_* + \gamma L^2\sum_{k=0}^{K-1}\mathbb{E}[V_k] \\
&\quad + KL\gamma^2\left(\frac{\sigma^2}{N_{\min}} + \delta_{pv,2}^2\right),
\end{aligned}
$$

where $f_*$ is a uniform lower bound for $f$. $\qquad\square$

The next step towards completing the proof of Theorem 3.4 gives the upper bound for $\sum_{k=0}^{K-1}\mathbb{E}[V_k]$ that appeared in (50).

**Lemma D.8.** *Let $f_1 = \ldots = f_N = f$ be $L$-smooth and bounded from below by $f_*$, and Assumptions 3.1 and 3.2 hold with $\Delta_{pv}^k = \delta_{pv,1}\gamma\mathbb{E}[\|\nabla f(\theta^k)\|^2] + L\gamma^2\delta_{pv,2}^2$, $\delta_{pv,1} \in [0, 1/2)$, $\delta_{pv,2} \ge 0$. Then, for any $K \ge 0$ the iterates produced by Moshpit SGD with $\gamma \le 1/(4\sqrt{e}L(\tau-1))$ satisfy*

$$
\sum_{k=0}^{K-1}\mathbb{E}[V_k] \le 8e\gamma^2(\tau-1)^2\sum_{k=0}^{K-1}\mathbb{E}[\|\nabla f(\theta^k)\|^2] + 4\gamma^2 K\left(2\delta_{aq}^2 + e(\tau-1)\sigma^2\right), \quad (55)
$$

*where $V_k = \frac{1}{N_k}\sum_{i\in P_k}\|\theta_i^k - \theta^k\|^2$ and $\theta^k = \frac{1}{N_k}\sum_{i\in P_k}\theta_i^k$.*

*Proof.* First of all, consider $k$ such that $k = a\tau + t'$ for some $t' \in [0, \tau)$. Let $\mathbb{E}_g[\cdot]$ denote the expectation conditioned on $\{P_t\}_{t=a\tau}^{(a+1)\tau-1}$. Then

$$
\begin{aligned}
\mathbb{E}_g[V_k] \;&=\; \frac{1}{N_k} \sum_{i \in P_k} \mathbb{E}_g \left[ \|\theta_i^k - \theta^k\|^2 \right] \overset{(35)}{\leq} \frac{1}{N_k} \sum_{i \in P_k} \mathbb{E}_g \left[ \|\theta_i^k - \theta^{a\tau}\|^2 \right] \\
&=\; \frac{1}{N_k} \sum_{i \in P_k} \mathbb{E}_g \left[ \left\| \theta_i^{a\tau} - \theta^{a\tau} - \gamma \sum_{t=a\tau}^{k-1} g_i^t \right\|^2 \right] \\
&\overset{(32)}{\leq}\; \frac{2}{N_k} \sum_{i \in P_k} \mathbb{E}_g \left[ \|\theta_i^{a\tau} - \theta^{a\tau}\|^2 \right] + \frac{2\gamma^2}{N_k} \sum_{i \in P_k} \mathbb{E}_g \left[ \left\| \sum_{t=a\tau}^{k-1} g_i^t \right\|^2 \right]. \quad (56)
\end{aligned}
$$

Next, we estimate the second term in the right-hand side of (56) using Lemma D.4:

$$
\begin{aligned}
\frac{2\gamma^2}{N_k} \sum_{i \in P_k} \mathbb{E}_g \left[ \left\| \sum_{t=a\tau}^{k-1} g_i^t \right\|^2 \right] \;&\overset{(38)}{\leq}\; \frac{2e\gamma^2(k-a\tau)}{N_k} \sum_{i \in P_k} \sum_{t=a\tau}^{k-1} \mathbb{E}_g[\|\nabla f(\theta_i^t)\|^2] \\
&\qquad + \frac{2e\gamma^2}{N_k} \sum_{i \in P_k} \sum_{t=a\tau}^{k-1} \mathbb{E}_g[\|g_i^t - \nabla f(\theta_i^t)\|^2] \\
&\overset{(32),(7)}{\leq}\; 4e\gamma^2(\tau-1) \sum_{t=a\tau}^{k-1} \mathbb{E}_g[\|\nabla f(\theta^t)\|^2] \\
&\qquad + 4e\gamma^2(\tau-1) \sum_{t=a\tau}^{k-1} \frac{1}{N_k} \sum_{i \in P_k} \mathbb{E}_g[\|\nabla f(\theta_i^t) - \nabla f(\theta^t)\|^2] \\
&\qquad + 2e\gamma^2(k-a\tau)\sigma^2 \\
&\overset{(28)}{\leq}\; 4e\gamma^2(\tau-1) \sum_{t=a\tau}^{k-1} \mathbb{E}_g[\|\nabla f(\theta^t)\|^2] \\
&\qquad + 4e\gamma^2 L^2(\tau-1) \sum_{t=a\tau}^{k-1} \frac{N_t}{N_k} \cdot \frac{1}{N_t} \sum_{i \in P_t} \mathbb{E}_g[\|\theta_i^t - \theta^t\|^2] \\
&\qquad + 2e\gamma^2(\tau-1)\sigma^2 \\
&\leq\; 4e\gamma^2(\tau-1) \sum_{t=a\tau}^{k-1} \mathbb{E}_g[\|\nabla f(\theta^t)\|^2] \\
&\qquad + 8e\gamma^2 L^2(\tau-1) \sum_{t=a\tau}^{k-1} \mathbb{E}_g[V_t] + 2e\gamma^2(\tau-1)\sigma^2,
\end{aligned}
$$

where in the last two inequalities we use $N_k = |P_k| \leq |P_{k-1}| = N_{k-1}$ for all $k \geq 1$ and $N_{a\tau} \leq 2N_{(a+1)\tau}$ for all integer $a \geq 0$. Plugging this inequality in (56) and taking the full expectation

from the result, we get

$$
\begin{aligned}
\mathbb{E}[V_k] \quad \leq \quad & 2\mathbb{E}\left[\frac{1}{N_k}\sum_{i\in P_k}\|\theta_i^{a\tau}-\theta^{a\tau}\|^2\right] + 4e\gamma^2(\tau-1)\sum_{t=a\tau}^{k-1}\mathbb{E}[\|\nabla f(\theta^t)\|^2] \\
& +8e\gamma^2L^2(\tau-1)\sum_{t=a\tau}^{k-1}\mathbb{E}[V_t] + 2e\gamma^2(\tau-1)\sigma^2 \\
\leq \quad & 4\mathbb{E}\left[\frac{1}{N_{a\tau}}\sum_{i\in P_{a\tau}}\|\theta_i^{a\tau}-\theta^{a\tau}\|^2\right] + 4e\gamma^2(\tau-1)\sum_{t=a\tau}^{k-1}\mathbb{E}[\|\nabla f(\theta^t)\|^2] \\
& +8e\gamma^2L^2(\tau-1)\sum_{t=a\tau}^{k-1}\mathbb{E}[V_t] + 2e\gamma^2(\tau-1)\sigma^2 \\
\overset{(10)}{\leq} \quad & 4e\gamma^2(\tau-1)\sum_{t=a\tau}^{k-1}\mathbb{E}[\|\nabla f(\theta^t)\|^2] + 8e\gamma^2L^2(\tau-1)\sum_{t=a\tau}^{k-1}\mathbb{E}[V_t] \\
& +2\gamma^2\left(2\delta_{aq}^2 + e(\tau-1)\sigma^2\right),
\end{aligned}
$$

where in the second inequality we also use $N_k=|P_k|\leq|P_{k-1}|=N_{k-1}$ for all $k\geq 1$ and $N_{a\tau}\leq 2N_{(a+1)\tau}$ for all integer $a\geq 0$. Summing up the obtained inequalities for $k=a\tau, a\tau+1,\ldots,K'$ for some $K'\in[a\tau,(a+1)\tau-1]$ we derive

$$
\begin{aligned}
\sum_{k=a\tau}^{K'}\mathbb{E}[V_k] \quad \leq \quad & 4e\gamma^2(\tau-1)\sum_{k=a\tau}^{K'}\sum_{t=a\tau}^{k-1}\mathbb{E}[\|\nabla f(\theta^t)\|^2] + 8e\gamma^2L^2(\tau-1)\sum_{k=a\tau}^{K'}\sum_{t=a\tau}^{k-1}\mathbb{E}[V_t] \\
& +2\gamma^2(K'-a\tau+1)\left(2\delta_{aq}^2 + e(\tau-1)\sigma^2\right) \\
\leq \quad & 4e\gamma^2(\tau-1)^2\sum_{k=a\tau}^{K'}\mathbb{E}[\|\nabla f(\theta^k)\|^2] + 8e\gamma^2L^2(\tau-1)^2\sum_{k=a\tau}^{K'}\mathbb{E}[V_k] \\
& +2\gamma^2(K'-a\tau+1)\left(2\delta_{aq}^2 + e(\tau-1)\sigma^2\right) \\
\leq \quad & 4e\gamma^2(\tau-1)^2\sum_{k=a\tau}^{K'}\mathbb{E}[\|\nabla f(\theta^k)\|^2] + \frac{1}{2}\sum_{k=a\tau}^{K'}\mathbb{E}[V_k] \\
& +2\gamma^2(K'-a\tau+1)\left(2\delta_{aq}^2 + e(\tau-1)\sigma^2\right),
\end{aligned}
$$

where in the last inequality we use $\gamma\leq 1/(4\sqrt{e}L(\tau-1))$. Rearranging the terms, we get that for $K'\geq 0$

$$
\sum_{k=a\tau}^{K'}\mathbb{E}[V_k] \quad \leq \quad 8e\gamma^2(\tau-1)^2\sum_{k=a\tau}^{K'}\mathbb{E}[\|\nabla f(\theta^k)\|^2] + 4\gamma^2(K'-a\tau+1)\left(2\delta_{aq}^2 + e(\tau-1)\sigma^2\right),
$$

where $a\geq 0$ is an integer such that $a\tau\leq K'\leq (a+1)\tau-1$. Summing up the obtained inequalities for $K'=\tau-1,2\tau-1,\ldots,\tau\lfloor(K-1)/\tau\rfloor-1, K-1$, we derive (55). $\qquad\square$

Combining Lemmas D.7 and D.8, we get the following result:

**Theorem D.2** (Theorem 3.4). *Let $f_1=\ldots=f_N=f$, function $f$ be $L$-smooth and bounded from below by $f_*$, and Assumptions 3.1 and 3.2 hold with $\Delta_{pv}^k = \delta_{pv,1}\gamma\mathbb{E}[\|\nabla f(\theta^k)\|^2] + L\gamma^2\delta_{pv,2}^2$, $\delta_{pv,1}\in[0,1/2)$, $\delta_{pv,2}\geq 0$. Then, for any $K\geq 0$ the iterates produced by Moshpit SGD with*

$$
\gamma\leq\min\left\{\frac{1-2\delta_{pv,1}}{8L}, \frac{\sqrt{1-2\delta_{pv,1}}}{8\sqrt{e}L(\tau-1)}\right\}
$$

*satisfy*

$$
\begin{aligned}
\mathbb{E}\left[\|\nabla f(\theta_{rand}^K)\|^2\right] \quad \leq \quad & \frac{8\Delta_0}{(1-2\delta_{pv,1})K\gamma} \\
& +\frac{8L\gamma}{1-2\delta_{pv,1}}\left(\frac{\sigma^2}{N_{\min}} + \delta_{pv,2}^2 + 4\gamma L\left(2\delta_{aq}^2 + e(\tau-1)\sigma^2\right)\right), \quad (57)
\end{aligned}
$$

*where* $\Delta_0 = f(\theta^0) - f_*$ *and* $\theta^K_{rand}$ *is chosen uniformly at random from* $\{\theta^0, \theta^1, \ldots, \theta^{K-1}\}$. *That is,*
*Moshpit SGD achieves* $\mathbb{E}\left[\|\nabla f(\theta^K_{rand})\|^2\right] \leq \varepsilon^2$ *after*

$$\mathcal{O}\left(\frac{L\Delta_0}{(1 - 2\delta_{pv,1})^2 \varepsilon^2}\left[1 + (\tau - 1)\sqrt{1 - 2\delta_{pv,1}} + \frac{\delta_{pv,2}^2 + \sigma^2/N_{\min}}{\varepsilon^2}\right.\right.$$

$$\left.\left. + \frac{\sqrt{(1-2\delta_{pv,1})(\delta_{aq}^2 + (\tau-1)\sigma^2)}}{\varepsilon}\right]\right) \qquad (58)$$

*iterations with*

$$\gamma = \min\left\{\frac{1 - 2\delta_{pv,1}}{8L}, \frac{\sqrt{1 - 2\delta_{pv,1}}}{8\sqrt{e}L(\tau - 1)}, \sqrt{\frac{\Delta_0}{LK\left(\delta_{pv,2}^2 + \sigma^2/N_{\min}\right)}}, \sqrt[3]{\frac{\Delta_0}{4L^2\left(2\delta_{aq}^2 + e(\tau - 1)\sigma^2\right)}}\right\}.$$

*Proof of Theorem 3.4.* Plugging the result of Lemma D.8 in the inequality (50) from Lemma D.7, we obtain

$$
\begin{aligned}
\frac{(1 - 2\delta_{pv,1})\gamma}{4}\sum_{k=0}^{K-1}\mathbb{E}\left[\|\nabla f(\theta^k)\|^2\right] &\leq f(\theta^0) - f_* + 8e\gamma^3 L^2\tau(\tau - 1)\sum_{k=0}^{K-1}\mathbb{E}[\|\nabla f(\theta^k)\|^2] \\
&\quad + KL\gamma^2\left(\frac{\sigma^2}{N_{\min}} + \delta_{pv,2}^2\right) \\
&\quad + 4KL^2\gamma^3\left(2\delta_{aq}^2 + e(\tau - 1)\sigma^2\right) \\
&\leq f(\theta^0) - f_* + \frac{(1 - 2\delta_{pv,1})\gamma}{8}\sum_{k=0}^{K-1}\mathbb{E}\left[\|\nabla f(\theta^k)\|^2\right] \\
&\quad + KL\gamma^2\left(\frac{\sigma^2}{N_{\min}} + \delta_{pv,2}^2\right) \\
&\quad + 4KL^2\gamma^3\left(2\delta_{aq}^2 + e(\tau - 1)\sigma^2\right).
\end{aligned}
$$

Next,

$$
\begin{aligned}
\frac{1}{K}\sum_{k=0}^{K}\mathbb{E}\left[\|\nabla f(\theta^k)\|^2\right] &\leq \frac{8\Delta_0}{(1 - 2\delta_{pv,1})K\gamma} \\
&\quad + \frac{8L\gamma}{1 - 2\delta_{pv,1}}\left(\frac{\sigma^2}{N_{\min}} + \delta_{pv,2}^2 + 4\gamma L\left(2\delta_{aq}^2 + e(\tau - 1)\sigma^2\right)\right),
\end{aligned}
$$

where $\Delta_0 = f(\theta^0) - f_*$. Since $\theta^K_{rand}$ is chosen uniformly at random from $\{\theta^0, \theta^1, \ldots, \theta^{K-1}\}$,

$$\mathbb{E}\left[\|\nabla f(\theta^K_{rand})\|^2\right] \stackrel{(36)}{=} \frac{1}{K}\sum_{k=0}^{K}\mathbb{E}\left[\|\nabla f(\theta^k)\|^2\right]$$

and (57) holds. Applying Lemma D.3 to (57), we get the following result: if

$$\gamma = \min\left\{\frac{1 - 2\delta_{pv,1}}{8L}, \frac{\sqrt{1 - 2\delta_{pv,1}}}{8\sqrt{e}L(\tau - 1)}, \sqrt{\frac{\Delta_0}{LK\left(\delta_{pv,2}^2 + \sigma^2/N_{\min}\right)}}, \sqrt[3]{\frac{\Delta_0}{4L^2\left(2\delta_{aq}^2 + e(\tau - 1)\sigma^2\right)}}\right\},$$

then $\mathbb{E}\left[\|\nabla f(\theta^K_{rand})\|^2\right]$ equals

$$\mathcal{O}\left(\frac{L\Delta_0\left(1 + (\tau - 1)\sqrt{1 - 2\delta_{pv,1}}\right)}{(1 - 2\delta_{pv,1})^2 K} + \sqrt{\frac{L\Delta_0\left(\delta_{pv,2}^2 + \sigma^2/N_{\min}\right)}{(1 - 2\delta_{pv,1})^2 K}} + \frac{\sqrt[3]{L^2\Delta_0^2(\delta_{aq}^2 + (\tau - 1)\sigma^2)}}{(1 - 2\delta_{pv,1})K^{2/3}}\right),$$

which implies the desired convergence result from (58). $\qquad\square$

# E  Decentralized matchmaking

In order to run group all-reduce over unreliable devices, Moshpit Averaging must be able to dynamically form groups of active devices that share the same key $C_i$. In theory, this matchmaking can be implemented precisely as described in Algorithm 1: each peer adds itself to a certain DHT key, waits for a said period of time, and then reads the same key to retrieve a list of its groupmates.

However, in practice, this kind of matchmaking would be extremely fragile: if any peer arrives late (for example, due to latency), it may join the group when other peers have already finished matchmaking. As a result, some workers will treat this peer as active, while others will behave as though there is no such peer at all, breaking the consensus and rendering all peers unable to run all-reduce in a stable manner.

To avoid this and other similar inconsistencies, Moshpit All-Reduce employs a more sophisticated matchmaking protocol with the following guarantees

1. Peers that join the same group are guaranteed to have the same list of groupmates;

2. The group will have the maximum possible number of peers, unless some of them fail;

3. If some peers fail, matchmaking will still form the group out of the remaining ones.

To achieve this, each peer first declares itself onto the DHT (as in Algorithm 1). Then, peers attempt to form groups by calling the REQUEST_JOIN_GROUP remote procedure call. Intuitively, if peer A calls this RPC on peer B, then *peer A requests to join peer B's group*, which can be either accepted or rejected by the group "leader" B, which may or may not have other "followers".

If a peer is accepted to a group, it commits to stay active (i.e. to await other peers) for a set period of time and perform all-reduce with the peers supplied by the group "leader". On the other hand, a peer can be rejected if (a) the potential "leader" is already a follower in another group, (b) the group is already running all-reduce, or (c) if the "leader" failed or left during matchmaking.

To ensure that this protocol forms groups of maximum size, each peer generates a unique "priority" based on its local timestamp[9]. Peers prioritize joining the group of neighbors that have the lowest "priority". Under normal circumstances, all workers will join the group of a peer that was first to start matchmaking according to its own local time. However, if this peer has failed or already finished matchmaking, the group will be formed around one of the remaining peers.

Matchmaking for 64 peers can take less than 1 second if all workers are located in the same cloud region and are highly synchronized. However, this can grow to 2.9 seconds for two different cloud regions and up to 9 seconds when training with commodity hardware around the world.

To ensure that this latency does not affect the training performance, Moshpit SGD performs matchmaking asynchronously in the background thread, while the model is accumulating gradients. All peers begin matchmaking 15 seconds before the estimated averaging round, so that in $\geq 95\%$ of averaging iterations, the matchmaking step is already finished by the time peers need to run all-reduce.

# F  Training with a dynamic number of peers

Many practical setups with unreliable devices allow peers to join or leave at any time, which can produce undesirable side-effects. For instance, consider a participant that joins the "swarm" midway through the training process. If this participant starts with the initial model parameters, it can undo some of the progress made by other peers.

To circumvent this issue, we require each new participant to download the latest parameters from a random up-to-date peer discovered through DHT. The same technique is used to synchronize the optimizer statistics and the learning rate schedule. This protocol is also triggered if a peer becomes desynchronized with others, e.g., after a network freeze.

---

[9]More specifically, the priority is a tuple of (timestamp, peer_id), where peer_id is used to break ties.

# G   Load balancing via linear programming

When running Moshpit Averaging on heterogeneous devices, one must regularly perform Butterfly All-Reduce among peers with uneven network bandwidth. In order to speed up the protocol, we can make low-throughput peers receive, average, and send smaller partitions of the averaged vector; conversely, the high-throughput peers can process greater fractions of the input vector. To compute the optimal partitioning, peers must solve an optimization problem that minimizes the total time spent on communication during all-reduce.

Consider a group of $M$ peers with network bandwidths $b_1, ..., b_M$, defined for simplicity as the minimum of the upload and download speed for each peer. Our objective is to find $w_i$ — a fraction of all input vectors to be processed by the $i$-th peer.

In Butterfly All-Reduce, each peer $i$ splits its vector into parts and sends these parts to corresponding peers. Since there is no need to send $w_i$ to itself, $i$-th peer will upload a total of $1 - w_i$ of the vector to its peers. On the receiving side, peer $i$ will average $w_i$ of the vector from all peers in its group. To do so, it must download $M - 1$ vector parts of size $w_i$ from all other peers. After that, peers distribute the averaged parts by running the same procedure in reverse (see Figure 1).

Thus, the communication time for each peer is proportional to $t_i = (1 - w_i + (M - 1)w_i) \cdot \frac{1}{b_i}$ and the total runtime of Butterfly All-Reduce is the maximum communication time over all peers: $T = \max_i t_i = \max_i (1 - w_i + (M - 1)w_i) \cdot \frac{1}{b_i}$. Formally, we minimize $T$ with respect to $w_i$ with two constraints on the fraction weights:

$$\min_w \quad \max_i (1 - w_i + (M - 1)w_i) \cdot \frac{1}{b_i}$$
$$\text{subject to} \quad \sum_{i=1}^{M} w_i = 1$$
$$w_i \geq 0 \qquad \forall i = 1, \ldots, M$$

Because the functions being maximized and the constraints are linear in $w_i$, this problem can be reduced to linear programming [125]. Namely, we can minimize a surrogate variable $\xi$ such that $\forall i, \xi \geq (1 - w_i + (M - 1) \cdot w_i) \cdot \frac{1}{b_i}$. The resulting linear program is formulated as follows:

$$\min_{w, \xi} \quad \xi$$
$$\text{subject to} \quad \sum_{i=1}^{M} w_i = 1$$
$$w_i \geq 0 \qquad \forall i = 1, \ldots, M$$
$$\xi \geq (1 - w_i + (M - 1)w_i) \cdot \frac{1}{b_i} \quad \forall i = 1, \ldots, M$$

We solve this problem using the interior point method [126] implemented as part of the SciPy package (`scipy.optimize.linprog`). Note that depending on the conditions given by participant bandwidth, optimal weights of specific peers might be equal to 0 in some cases. In essence, this allows our method to smoothly interpolate between data parallelism [9], parameter server [18] and sharded parameter server [25] in manner similar to BytePS [26].

# H   Detailed experimental setup

In this section, we provide the detailed hardware configuration of servers used for each of our distributed training experiments.

## H.1   ImageNet training

Both homogeneous and heterogeneous training setups for ImageNet are provisioned in our on-premise infrastructure across multiple data centers and an office space (for the heterogeneous setup only).

**Homogeneous.** For the homogeneous setup, we use 16 identical instances with the following specifications:

- **GPU:** V100-PCIe,
- **CPU:** 6 vCPUs (Xeon E5-2650v4),
- **RAM:** 64GB.

**Heterogeneous.** In turn, the heterogeneous setup contains multiple instance types listed in Table 2:

Table 2: **Heterogeneous** setup for ImageNet training.

| Instances | GPUs | GPU type | Cores | RAM, GB | CPU type |
|---|---|---|---|---|---|
| 4 | 1 | V100-PCIe | 6 | 64 | E5-2650v4 |
| 17 | 2 | GTX 1080Ti | 8 | 64 | E5-2650v4 |
| 7 | 1 | GTX 1080Ti | 4 | 32 | E5-2650v4 |
| 16 | 1 | P40 | 4 | 32 | E5-2667v2 |
| 20 | 1 | M40-24GB | 4 | 32 | E5-2667v2 |

## H.2 ALBERT training

**Homogeneous.** For the homogeneous setup, we use a single virtual machine with the following specifications:

- **GPU:** $8\times$ V100-PCIe,
- **CPU:** 48 vCPUs (Xeon E5-2650v4),
- **RAM:** 488GB.

At the time of writing, the cloud rent cost for this instance is **\$24.48** per hour.

**Heterogeneous.** Our heterogeneous setup is composed of two parts: AWS EC2 Spot instances and crowdsourced machines from the `Vast.ai` marketplace. For spot instances, we picked the smallest suitable instance size available from the cloud provider and further limited their bandwidth to 1Gb/s[10]. As for marketplace instances, we report the hardware specifications for each worker gathered 1 hour after the start of ALBERT training.

Since both cloud and marketplace instances are preemptible, the actual cost of the server fleet will vary based on the current price. For simplicity, we report the maximum hourly price we ended up paying for this instance (enforced via maximum bid). Finally, some marketplace instances have missing specifications, such as unknown CPU type. This is likely caused by non-standard virtualization configured by the device owner. The resulting fleet configuration, shown in Table 3, costs up to \$15.43/hour, depending on the number of active instances.

# I   Additional averaging experiments

In this section, we evaluate the averaging precision with the same methodology as in 4.1, but for multiple different worker configurations.

Table 4 provides the complete results of our experiments that were used to make conclusions in the main experimental section: instead of reporting the mean squared error for different iterations, we provide the number of rounds that was required to achieve the error of $10^{-9}$ and $10^{-4}$.

In Figure 5, plots 1–5 explore several combinations of grid sizes and failure rates, whereas plot 6 (bottom right) demonstrates a setup with the same number of peers ($10^6$) arranged into several different grid sizes and its relation to convergence. Note that $M{=}32$ outperforms the alternatives only for the specific failure rate of 0.001.

---

[10]We use `tc qdisc` Linux utility to artificially limit the network throughput, similarly to [127]

Table 3: **Heterogeneous** setup for ALBERT training.

| GPU | Cores | RAM, GB | CPU type | Download, Mb/s | Upload, Mb/s | Cost, $/hour |
|---|---|---|---|---|---|---|
| | | | Preemptible `g4dn.xlarge` instances (32×) | | | |
| T4 | 4 | 16 | Xeon Platinum 8259CL | 1000 | 1000 | 0.1578 |
| | | | Marketplace instances | | | |
| GTX 1070Ti | 6 | 16 | E5-2640 | 425 | 255 | 0.036 |
| GTX 1070Ti | 6 | 16 | i3-6100T | 121 | 36 | 0.06 |
| GTX 1080Ti | 4 | 20 | i3-6096P | 817 | 308 | 0.101 |
| GTX 1080Ti | 20 | 129 | E5-2630v4 | 660 | 475 | 0.182 |
| GTX 1080Ti | 1 | 16 | i7-7700K | 245 | 210 | 0.302 |
| GTX 1080Ti | 48 | 97 | Xeon Platinum 8124 | 583 | 539 | 0.217 |
| GTX 1080Ti | 10 | 16 | Unknown | n/a | n/a | 0.15 |
| GTX 1080Ti | 4 | 16 | Xeon Gold 6149 | 98 | 100 | 0.2 |
| GTX 1080Ti | 4 | 16 | Xeon Gold 6149 | 99 | 98 | 0.2 |
| GTX 1080Ti | 4 | 16 | Xeon Gold 6149 | 99 | 99 | 0.2 |
| GTX 1080Ti | 4 | 16 | Xeon Gold 6149 | 99 | 99 | 0.2 |
| RTX 2070S | 24 | 32 | E5-2620v2 | 199 | 25 | 0.199 |
| RTX 2070S | 32 | 97 | E5-2650 | 162 | 64 | 0.285 |
| RTX 2080 | 6 | 16 | E5-2620v3 | 271 | 287 | 0.25 |
| RTX 2080 | 24 | 32 | E5-2630v3 | 199 | 25 | 0.302 |
| RTX 2080S | 4 | 32 | E5-2697v4 | 101 | 99 | 0.292 |
| RTX 2080S | 4 | 32 | E5-2697v4 | 93 | 99 | 0.292 |
| RTX 2080S | 4 | 32 | E5-2697v4 | 94 | 98 | 0.292 |
| RTX 2080S | 4 | 32 | E5-2697v4 | 94 | 98 | 0.292 |
| RTX 2080S | 4 | 32 | E5-2697v4 | 100 | 99 | 0.292 |
| RTX 2080Ti | 4 | 16 | Ryzen Threadripper 3960x | 279 | 271 | 0.35 |
| RTX 2080Ti | 8 | 129 | E5-2670v3 | 616 | 672 | 0.201 |
| RTX 2080Ti | 6 | 32 | E5-2620v3 | 217 | 61 | 0.22 |
| RTX 2080Ti | 8 | 16 | E5-2697v2 | 100 | 58 | 0.3 |
| RTX 2080Ti | 8 | 21 | E5-2697v2 | 145 | 49 | 0.243 |
| RTX 2080Ti | 12 | 32 | Unknown | 111 | 92 | 0.326 |
| RTX 2080Ti | 12 | 64 | E5-2690v3 | 205 | 61 | 0.549 |
| RTX 3080 | 16 | 16 | i7-10700K | 69 | 49 | 0.462 |
| RTX 3090 | 14 | 32 | E5-2695v3 | 93 | 37 | 0.498 |
| RTX 3090 | 16 | 32 | Ryzen 9 3950X | 338 | 38 | 0.511 |
| Titan RTX | 4 | 32 | Xeon W-3223 | 321 | 115 | 1 |
| Titan RTX | 4 | 32 | Xeon Gold 6149 | 99 | 100 | 0.702 |
| Titan V | 8 | 32 | i7-7700K | 97 | 50 | 0.282 |
| V100-FHHL | 8 | 60 | Xeon Gold 6148 | 544 | 584 | 0.39 |
| | | | Total hourly cost (as listed): | | | **15.43** |

## J   Additional image classification experiments

Aside from the two evaluation scenarios provided in 4.2, we also measure the performance of Moshpit-SGD in a non-distributed setup, i.e. on a single server with multiple GPUs. We conduct this experiment on the same $8\times$ V100 machine that was used in the **homogeneous** setup for training ALBERT (see Appendix H.2).

As Figure 6 demonstrates, Moshpit SGD is slower than AR-SGD by approximately $25\%$. This result is expected, since our implementation of Moshpit All-Reduce is more general and communicates over a TCP connection, whereas AR-SGD uses direct peer-to-peer GPU communication over PCIe. On average, this incurs a slowdown of $27\%$ in terms of training time.

Table 4: Averaging performance of different algorithms. Values denote the number of iterations required to achieve the error of $10^{-9}$ ($10^{-4}$ in parentheses), the best result is in bold.

| $N$ | $p$ | All-Reduce | Gossip | PushSum | Random groups | Moshpit |
|---|---|---|---|---|---|---|
| 512 | 0 | **1.0 (1.0)** | 50.0 (50.0) | 47.6 (15.6) | 6.1 (3.0) | 8.2 (3.5) |
| 512 | 0.001 | **1.6 (1.6)** | 50.0 (50.0) | 47.6 (15.6) | 6.3 (3.0) | 8.1 (3.7) |
| 512 | 0.005 | 10.9 (10.9) | 50.0 (50.0) | 47.8 (15.6) | **6.3 (3.0)** | 8.7 (3.9) |
| 512 | 0.01 | 41.7 (41.7) | 50.0 (50.0) | 47.8 (15.6) | **6.6 (3.0)** | 9.1 (3.9) |
| 768 | 0 | **1.0 (1.0)** | 50.0 (50.0) | 43.2 (13.8) | 6.2 (3.0) | 6.0 (3.0) |
| 768 | 0.001 | **1.8 (1.8)** | 50.0 (50.0) | 43.2 (13.8) | 6.5 (3.0) | 6.2 (3.0) |
| 768 | 0.005 | 28.7 (28.7) | 50.0 (50.0) | 43.2 (14.1) | **6.6 (3.0)** | **6.6 (3.0)** |
| 768 | 0.01 | 50.0 (50.0) | 50.0 (50.0) | 43.9 (14.2) | 7.0 (3.0) | **6.8 (3.0)** |
| 900 | 0 | **1.0 (1.0)** | 50.0 (50.0) | 45.0 (14.7) | 6.4 (3.0) | 5.0 (2.8) |
| 900 | 0.001 | **1.8 (1.8)** | 50.0 (50.0) | 45.0 (14.7) | 6.3 (3.0) | 5.5 (3.0) |
| 900 | 0.005 | 50.0 (50.0) | 50.0 (50.0) | 45.2 (14.7) | 6.7 (3.0) | **5.9 (3.0)** |
| 900 | 0.01 | 50.0 (50.0) | 50.0 (50.0) | 45.6 (14.9) | 7.0 (3.1) | **6.4 (3.1)** |
| 1024 | 0 | **1.0 (1.0)** | 50.0 (50.0) | 49.0 (16.2) | 6.2 (3.0) | 2.0 (2.0) |
| 1024 | 0.001 | **2.0 (2.0)** | 50.0 (50.0) | 49.0 (16.3) | 6.5 (3.0) | 3.4 (2.2) |
| 1024 | 0.005 | 42.6 (42.6) | 50.0 (50.0) | 49.5 (16.3) | 6.7 (3.0) | **5.4 (2.9)** |
| 1024 | 0.01 | 50.0 (50.0) | 50.0 (50.0) | 49.5 (16.3) | 6.9 (3.1) | **5.9 (3.0)** |

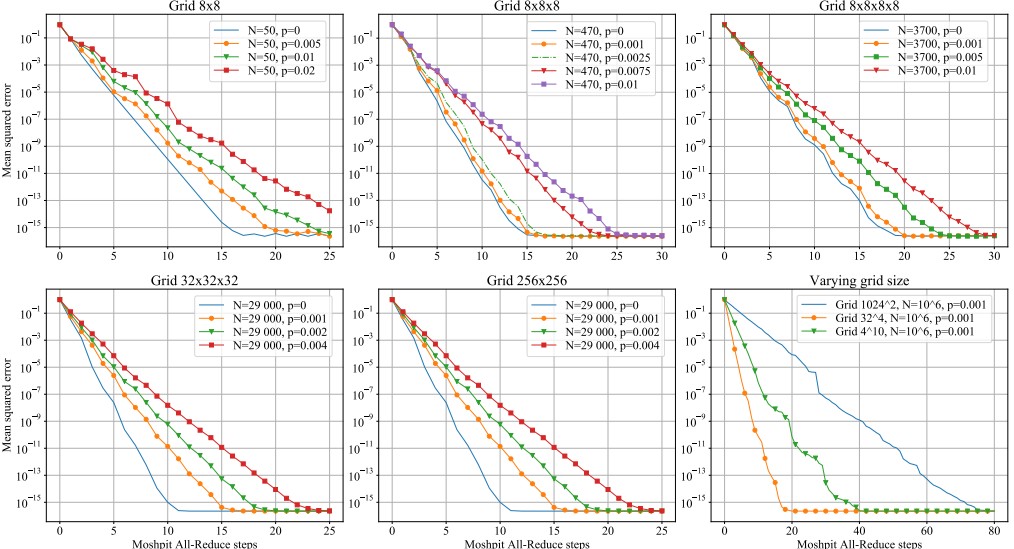

Figure 5: Averaging error of Moshpit All-Reduce as a function of the iteration number for different configurations and failure rates.

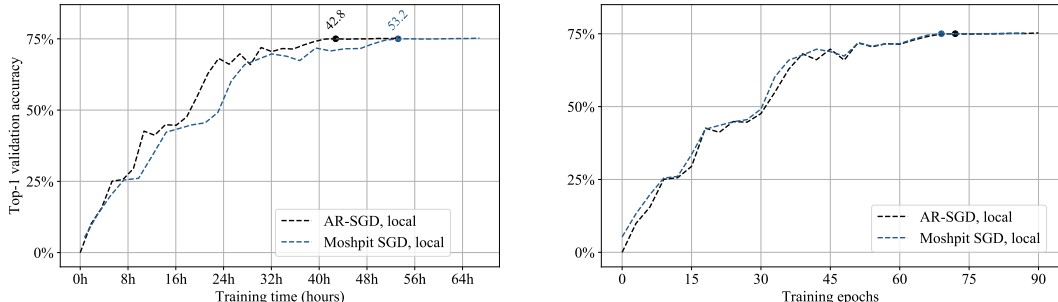

Figure 6: ResNet-50 top-1 validation accuracy on ImageNet when training on a single node with $8\times$ V100-PCIe GPUs. **(Left)** Convergence in terms of training time, **(Right)** Convergence in terms of training epochs