# OpenReview forum: "Moshpit SGD: Communication-Efficient Decentralized Training on Heterogeneous Unreliable Devices"
_NeurIPS.cc/2021/Conference — NeurIPS 2021 Poster_

### Official Review · Reviewer_XkgK · 2021-07-02

**Rating:** 6
**Confidence:** 4

**Summary:**

The paper introduces a decentralized averaging scheme and uses it for distributed learning.
The averaging scheme falls somewhere in between typical gossip averaging and uniform all-reduce averaging.
It inherits some of the robustness to node failures from typical gossip averaging and some of the communication efficiency from all-reduce collectives.

For each communication step, nodes are divided into groups using a decentralized hash table, and perform all-reduce averaging within their group. If a node in one group fails or takes too long to respond, other communication groups can still complete their averaging reliably.
In the light of prior work, this method could be classified as a linear time-varying averaging scheme, with block-uniform averaging matrices.

What I personally consider the most interesting contribution of the paper, is the match-making mechanism used to group the workers. By numbering the nodes in each group at timestep $t$, and using those numbers to group the nodes in the next round, workers are never grouped with the same peers in two consecutive communication rounds. I believe this is what gives the algorithm its name 'MoshPit' (although an explanation of this is missing from the paper).

The paper contains convergence rates for the algorithm with random grouping (not with moshpit match-making), and experiments show that the algorithm can reduce training time and cost, mainly in heterogeneous environments (bandwidth or compute capabilities).

**Ethical Concerns:**

I do not have any ethical concerns.

**Limitations And Societal Impact:**

I do not see any societal impact or limitations that warrant discussion.

**Main Review:**

I have enjoyed reading this paper, and I particularly found the moshpit match-making scheme clever and interesting.
The quality of the writing is good and the exposition is clear.
While I have some hesitations regarding the experiments and focus of the paper (see below), I believe efficient decentralized averaging schemes can have great significance for peer-to-peer and large-scale training.

While I think the match-making scheme described at the start of Section 3.1 is the most interesting contribution of this work (and gives the name to the proposed algorithm), I feel that this particular contribution does not receive the attention it deserves. The theory does not consider moshpit (repelling) grouping, and in the experiments, a many questions are unanswered:

- (1) How much does moshpit matchmaking help over random grouping?
- (2) Reliability to workers leaving and joining is presented as a benefit of this method. Will the group sizes remain balanced across iterations and groups if this happens? Is there a mechanism to achieve this?
- (3) The comment "wait for peers to assemble" in Algorithm 1 hides some non-trivial details. How do the workers synchronize and know their group is 'complete'? Do they use time-outs? How are those time-outs set and what is their impact on the performance of this method?

I also have some hesitations regarding the experimental setup:
- (4) While the experiments convincingly show that there are settings in which this method works well, the setup seems a little artificial (with additional randomized latency). This is fine, but I believe experiments that test individual aspects of the model could be more valuable (suggestions below)
- (5) For a group size of 2, the proposed method recovers something very much like the time-varying (torus) topology in Stochastic Gradient Push [15, 54]. For a group size of n, the method becomes all-reduce. I would like to see the impact of the choice of group size on the resilience to failures and communication efficiency. How should this be set in practice? Is there ever a benefit of using groups of size > 2 or smaller than $n$?
- (6) The benefits in the heterogeneous case seem to come mainly from the load balancing details in Appendix E. Other baselines such as all-reduce would benefit from the same mechanism. I believe it would be better if the authors could separate this contribution from their proposed averaging scheme.

While the theory in this paper is correct and relevant, it does considers a simplified algorithm (with random grouping) that does not include the match-making scheme I find most interesting about this paper.
It also seems that rates for this algorithm follow quite readily from prior work (e.g. Koloskova et al. [63]), by considering this method as time-varying linear gossip averaging with block-uniform matrices. Lines 248 and 249 compare the obtained rate to this paper. Could you please clarify the differences in the analysis?

I would consider increasing my score if the authors can address most of the questions (1 - 6). Below, I include a few minor questions and comments.

- The paper often claims that the method "does not depend on the spectral properties of the graph". I think this is misleading because the method requires a fully connected topology, and competing methods don't have such a dependence either in this case.
- Theorem 3.1 shows that a simplification of Moshpit Averaging (with random groups) converges linearly, even with communication failures. The same seems true for retrying all-reduce. Would it be possible to characterize the gains in reliability?
- For the claim about linear convergence in 131, I think you should list the key assumptions.
- I am missing some training details such as learning rates, schedules, warmup, and whether typical optimizations are used, such as overlapping computation and communication in the backward pass (see PyTorch's DistributedDataParallel).
- The paper mentions that all-to-all implementations of all-reduce are not always practical. While I can imagine this is because many messages are sent, I do not quite understand why this is problematic? Could you please elaborate a bit on this?
- Section 2.4 has some incorrect use of articles. Check [a] PS and "the changes".

__Edit:__ I have read the other reviews and the author's response. The authors have included significant new analysis. If the authors include this analysis in the paper, and commit to giving the proposed matchmaking procedure more attention in the mansucript, I would be fine with acceptance. I'll change my score from 5 to 6.

**Time Spent Reviewing:**

3

---

> ### Author Response · Authors · 2021-08-11
> **Author Response to Reviewer XkgK (part 2)**
>
> > The paper often claims that the method "does not depend on the spectral properties of the graph". I think this is misleading because the method requires a fully connected topology, and competing methods don't have such a dependence either in this case.
>
> Thank you for your suggestion! We will clarify that part. In fact, this phrase means that one should compare averaging procedures with the same budget of data that each worker sends and receives during one step of the procedure (i.e., 1 step of gossip or 1 step of Moshpit All-Reduce). In Moshpit All-Reduce, each worker sends and receives just O(s) amount of data at each iteration, where s is the size of the vector. In contrast, worker $i$ in gossip sends and receives O(N_i * s) amount of data at each iteration. To equalize the data load of the network for each approach, one should consider gossip with such a topology that each worker has O(1) neighbors. In this case, gossip-based methods have inevitable dependence on the spectral properties of the graph/mixing matrix, whereas Moshpit-SGD does not.
>
> > Theorem 3.1 shows that a simplification of Moshpit Averaging (with random groups) converges linearly, even with communication failures. The same seems true for retrying all-reduce. Would it be possible to characterize the gains in reliability?
>
>
> The main difference is that in Moshpit Averaging workers perform averaging in small groups at each iteration that is much cheaper than 1 round of All-Reduce when the number of workers is large.
>
> Regarding the reliability: when one applies restarted version of All-Reduce it is, in general, there are no input-independent guarantees for the 1 step of this procedure. For example, in a Butterfly All-Reduce, if the worker that handles the block of components with largest norms fails, then the averaging guarantee can be arbitrary bad. In the Ring All-Reduce, similar failures can happen. In contrast, even 1 iteration of random group averaging provides input-independent guarantees for averaging in expectation with the respect to the randomness of choosing the groups. This guarantee depends only on the number of resulting groups (Theorem 3.2.). In the same situation when 1 worker fails, we will simply get larger number of groups in the guarantee. For example, when all workers are split into groups of 2 and 1 worker fails it means that the number of groups increases by 1 that only slightly spoils the averaging guarantee from Theorem 3.2. Therefore, Moshpit Averaging is more robust than Restarted All-Reduce.
>
>
> > For the claim about linear convergence in 131, I think you should list the key assumptions.
>
>
> Thank you for spotting this issue! Actually, it is a typo: in general, rates are sublinear. However, when the objective function is strongly convex, Moshpit-SGD does converge with a linear rate to the neighborhood of the solution. The size of the neighborhood depends on the variance of the stochastic gradient and the stepsize. We provide convergence guarantees in regularly (non-strongly) convex and non-convex cases as well. We will clarify this place in the final version of the paper.
>
>
> > I am missing some training details such as learning rates, schedules, warmup, and whether typical optimizations are used, such as overlapping computation and communication in the backward pass (see PyTorch's DistributedDataParallel).
>
>
> For both ResNet and ALBERT experiments, we refer the reader to the original works ([16] and [70] respectively): as stated in L274 and L309-310, we use the exact same setup of the experiments without any modifications to hyperparameters. As a result, we did not list the training details in the text, as it would reduce the amount of space that could be used to describe parts of the setup that are specific to our work. We will, however, add these details of the setup in Appendix G to reduce the possibility of misunderstanding.
>
> Regarding the optimizations, as in AD-PSGD (stated in L279-280) and other baselines, we also overlap matchmaking and communication with computation, although it’s currently implemented in a less optimal way (we do not start averaging parameters for a layer as soon as its backward pass finishes, like in PyTorch DDP). If you are interested in our use of other specific optimizations, we would be happy to clarify on that.
>
>
> > The paper mentions that all-to-all implementations of all-reduce are not always practical. While I can imagine this is because many messages are sent, I do not quite understand why this is problematic? Could you please elaborate a bit on this?
>
>
> We believe you refer to L90 in the submission: in that case, the downsides of all-to-all communication were discussed in [10]. Notably, this work specifically proposes Ring All-Reduce as an alternative to butterfly All-Reduce because of the network contention issue. Thank you for noticing this; we will add the missing reference to that line and mention network contention to provide necessary context  to the reader.
>
>
> > Section 2.4 has some incorrect use of articles. Check [a] PS and "the changes".
>
> Thank you! We will fix that in an updated version of the work.

---

> ### Author Response · Authors · 2021-08-11
> **Author Response to Reviewer XkgK (part 1)**
>
> Thank you for your feedback and insightful suggestions! Please allow us to address your concerns below:
>
>
> > How much does moshpit matchmaking help over random grouping?
>
> Since in Moshpit matchmaking, two peers cannot be in the same group for two consecutive steps of averaging, it often converges better than random grouping due to this property, as demonstrated by our new experiments (see the general response). However, random grouping is easier to analyze, which is why we provide theoretical guarantees based on this simpler method.
>
> > Reliability to workers leaving and joining is presented as a benefit of this method. Will the group sizes remain balanced across iterations and groups if this happens? Is there a mechanism to achieve this?
>
> Although it is likely that the balance of group sizes cannot be strictly guaranteed without an additional procedure (that might incur additional overhead), in practice assigning peer indices by generating these numbers from a uniform distribution yields a uniform distribution across all grid dimensions (and across all groups as a result). If the worker leaves, their place in the grid becomes vacant, and other joining peers can use the corresponding index.
>
> > The comment "wait for peers to assemble" in Algorithm 1 hides some non-trivial details. How do the workers synchronize and know their group is 'complete'? Do they use time-outs? How are those time-outs set and what is their impact on the performance of this method?
>
> Thank you for this question! All workers know the initial group size (set as a parameter of the algorithm) when starting the experiment; later on, it can be changed by reading a corresponding key from the DHT. They synchronize by periodically reading the state of the DHT value associated with $C_i^{t-1}$. All methods related to the procedure are equipped with timeouts, set to a high enough value to offset the latency of communicating over the Internet but as low as possible in general: for example, the “lifetime” for a DHT key corresponding to the peer metadata is 15 seconds and the timeout for a request for averaging is 5 seconds. In practice, this does not result in any significant overhead, because the workers mostly wait for the group to be accumulated and waiting for other peers is executed as an asynchronous procedure in the background.
>
> > While the experiments convincingly show that there are settings in which this method works well, the setup seems a little artificial (with additional randomized latency). This is fine, but I believe experiments that test individual aspects of the model could be more valuable (suggestions below)
>
> We agree and address your suggestions to the best of our ability. However, we would also like to note that the ALBERT experiments (Section 4.3) use a real-world setup with the participating devices spread over multiple continents.
>
> > For a group size of 2, the proposed method recovers something very much like the time-varying (torus) topology in Stochastic Gradient Push [15, 54]. For a group size of n, the method becomes all-reduce. I would like to see the impact of the choice of group size on the resilience to failures and communication efficiency. How should this be set in practice? Is there ever a benefit of using groups of size > 2 or smaller than $n$?
>
>
> Thank you for this question! In general, the optimal group size will depend on the specifics of the hardware used in the experiment, such as the peer failure probability and the number of workers. As a result, one should pick the configuration of the group based on the infrastructure conditions; although it is possible to come up with a dynamic scheme (i.e., increase the group size until there are too many failures), we believe that this subject would benefit from a separate discussion in future work and will add that in the updated revision of the paper.
>
> To give an example of how one might pick the group size in practice, we conducted an additional experiment: for several group sizes and failure probabilities and a fixed number of participants (N=3000), we measured the number of iterations (out of T=200) required to reach the mean squared error of 1e-4 and 1e-9. The horizontal axis corresponds to the failure probability of each peer (same as in Section 4.1), and the vertical axis denotes the corresponding group size (the number of grid dimensions is such that the total size is always 4096). The results of both experiments are given below:
>
> Iterations for error of 1e-9:
>
> | $M$       | 4096  | 64    | 16        | 8         | 4     |
> |-----------|-------|-------|-----------|-----------|-------|
> | $p=0$     | **1** | 5.12  | 9         | 13.75     | 25.36 |
> | $p=0.005$ | 200   | 15    | **12.49** | 15.75     | 26.62 |
> | $p=0.01$  | 200   | 23.63 | **15**    | 17.26     | 28.38 |
> | $p=0.02$  | 200   | 44.53 | **20.5**  | 20.63     | 30.62 |
> | $p=0.05$  | 200   | 200   | 37.99     | **31.38** | 38.25 |
> | $p=0.1$   | 200   | 200   | 88.46     | **52.63** | 53.11 |
>
> Iterations for error of 1e-4:
>
> | $M$       | 4096  | 64     | 16       | 8         | 4     |
> |-----------|-------|--------|----------|-----------|-------|
> | $p=0$     | **1** | 2.74   | 4        | 5.37      | 9     |
> | $p=0.005$ |  200  | 6.51   | **5.12** | 6         | 9.51  |
> | $p=0.01$  |  200  | 10.1   | **6**    | 7         | 10.01 |
> | $p=0.02$  |  200  | 18.23  | 8.12     | **8**     | 10.88 |
> | $p=0.05$  |  200  | 106.06 | 15       | **11.38** | 13.12 |
> | $p=0.1$   |  200  | 200    | 34.22    | **18.38** | 17.5  |
>
> As you can see from the results, there are two important findings:
> 1) Group size of 4096 (which is equivalent to regular All-Reduce among all workers with restarts) is optimal only if the failure probability is negligible.
> 2) Both for high and low precision, the group sizes of 8 and 16 generally allow to achieve a high fraction of successful averaging iterations and sufficiently good convergence, which agrees with our experimental results in Sections 4.2 and 4.3. Thus, one might use these values as a “rule of thumb”, although the exact optimal grid configuration will certainly depend on more factors.
>
>
> > The benefits in the heterogeneous case seem to come mainly from the load balancing details in Appendix E. Other baselines such as all-reduce would benefit from the same mechanism. I believe it would be better if the authors could separate this contribution from their proposed averaging scheme.
>
> While in general the load balancing approach definitely might improve the performance of other methods, there are two caveats:
>
> First, in a homogeneous setting (which frequently arises in practice due to the popularity of cluster/HPC-based distributed training), load balancing will yield no meaningful improvements due to the uniformity and high speed of connections between hosts. As a result, applying it to other general distributed training methods beyond our target scenario might not be a very practical improvement worth extended discussion.
>
> Second, the optimization problem that we designed relies deeply on the algorithmic details of Butterfly All-Reduce (in particular, vector chunking). For example, directly applying it to Ring/Tree All-Reduce (most widely used in today’s data-parallel training due to the reasons described above) might be quite difficult, because each part of the vector eventually needs to be passed to all of the nodes. As a result, an extension of our approach to other methods needs nontrivial modifications of their underlying primitives and thus requires separate investigation.
>
> > It also seems that rates for this algorithm follow quite readily from prior work (e.g. Koloskova et al. [63]), by considering this method as time-varying linear gossip averaging with block-uniform matrices. Lines 248 and 249 compare the obtained rate to this paper. Could you please clarify the differences in the analysis?
>
> Indeed, in the non-convex case, our rates can be alternatively obtained from [63] using the argument of the reviewer. However, to apply the result from [63], one needs to show that Assumption 4 from [63] holds for Moshpit Averaging (or its simplified version with random grouping). Fortunately, it is possible due to our Theorem 3.2, which is of independent interest. Our analysis and the analysis from [63] are indeed similar, because both analyses are inspired by the “perturbed-iterates” analysis of SGD from
>
>     *Mania, Horia, et al. "Perturbed iterate analysis for asynchronous stochastic optimization." SIAM Journal on Optimization 27.4 (2017): 2202-2229.*
>
> that was further extended to Local-SGD, Error Compensated SGD and SGD with delayed updates in
>
>     *Stich, Sebastian U., and Sai Praneeth Karimireddy. "The error-feedback framework: Better rates for SGD with delayed gradients and compressed communication." arXiv preprint arXiv:1909.05350 (2019).*
>
> However, in the convex and strongly convex cases, our rates cannot be obtained directly from [63], since our rates are better: the first terms in our bounds (11) and (12) are $\tau$ times better than corresponding last-terms in the bounds provided in Theorem 2 from [63]. This shows the main difference between the analyses: while [63] is aimed at a highly-general scenario covering a large number of time-varying networks with arbitrarily heterogeneous local loss functions, we consider Moshpit-SGD as centralized Local-SGD with imperfect averaging applied to the problems with homogeneous local loss functions. This helps us derive better convergence guarantees.

---

> ### Author Response · Authors · 2021-08-26
> **Thank you for appreciating our response!**
>
> We thank the reviewer for appreciating our response. We will definitely apply the requested changes to the final version of our paper.

---

### Official Review · Reviewer_5vXz · 2021-07-16

**Rating:** 7
**Confidence:** 3

**Summary:**

Data parallel distributed NN training often depends on homogenous, well-provisioned networks and reliable end hosts.  This paper presents a scheme, Moshpit All-Reduce, that leverages a hybrid approach to communicating parameters (gradients) that combines elements of gossip with all-reduce to better withstand heterogeneous network environments and node failures.  The authors present their technique, and integrate it into a distributed SGD scheme (Moshpit SGD).  Microbenchmarks validate the extensive theoretical exploration of MAR, and the authors compare/contrast the system's effectiveness on both well-connected and distributed/heterogenous infrastructure.  They find that the scheme can tolerate varying network latencies and often more quickly converges by shortening training during individual epochs.

**Limitations And Societal Impact:**

yes

**Main Review:**

++ Robust training on cheap(er) nodes/networks is an important challenge.

++ Paper provides a novel communication algorithm for efficiently performing an all-reduce average.  Extensive analysis provides support for its convergence and robustness properties.

++ Experiments on homogenous and heterogenous networks with varying network latencies illustrate its ability to quickly approximate the true average and support distributed SGD.

++ The writing was clear and the notation consistent.

-- Given the complexity of the theoretical analysis, the empirical validation of Moshpit All-Reduce was notably weak.

-- There was a lack of description of the competing techniques (related work) beyond all-reduce.  (S4.2).

-- While the end-to-end training experiments are compelling, the experiment construction makes comparisons (and lessons learned) difficult.

Overall this appears to be an important problem and the novel communication protocol is well motivated.  Most of my concerns about the work lie within the empirical evaluation.   A key contribution here is MAR, but the simplistic, limited evaluation does the work a disservice.  While theoretical results (assuming the proofs are correct) are necessary, they may not always account for reality.   A key question (and contribution) then, would be to determine under what network conditions (latency, disconnections) and failure rates will Moshpit AR (and by association MSGD) succeed or fail?  Ultimately this feels like solid, important work, but it isn't sufficiently evaluated in the current work.

Before enumerating the kinds of experiments, there were a few places where the technical details were not clear.
* In Algorithm 1, it would be useful to characterize the asynchrony.   For example, there is a wait for peers to assemble.  Essentially, this is view maintenance/consensus.   If a node "joins late" what occurs?   They must be kicked out of the all-reduce.   If a there is agreement on the peer set, and then peers experience failure, what is the impact on the latency of the round (internal all-reduce timeouts), what is the impact of that timeout on convergence?  Is it possible for certain network or failure conditions to allow butterfly all-reduce to end in an inconsistent state?
* It would then seem to make sense to report the number of nodes that fail to participate in a round, as well as the distribution of round length (and variance of the aggregate). Theorem 3.2 maintains that the sum of all the groups is still N.  While the theorem explores an important relaxation of the initial assumptions, node failure/disconnections imply that rounds may proceed that violate that assumption.

To establish these properties the microbenchmarks should a.) plot the theoretical and competing systems (S4.2) results with the empirical results, b.) should change one aspect at a time (as opposed to changing N and p simultaneously), and c.) report performance statistics as a function of increasing node failure and (separately) connection latency.   Again, the point is to establish the claimed properties of MAR: faster (fewer rounds) and more efficient (less communication) convergence during failures and network performance.

Section 4.1 doesn't clearly state M or d (leaving the reader to parse "32x32" 'grid').   I'm assuming "18 steps" is T=18.  When failed peers return, do they return with the last returned parameters?   More importantly, failing for individual rounds seems contrived.   What about failing for some unspecified period of time (modeled with some distribution)?  In reality, peers aren't going to fail on nicely behaved round boundaries.  Also, I don't understand line 268.  Why does this experiment converge in two steps with failures?  Two steps are the minimum to achieve an exact answer without failure.   Were there no failures?    On that same graph, why would N=900 with no failures take so many steps (9) to converge to the true answer?   Clearly, even without failures, the true size of the groups must be << than 32.

A good section of the paper is taken up by showing that Moshpit AR can be used for distributed training (Moshpit SGD).   It isn't clear how many rejected distributed communications protocols have been the cause of SGD failures -- even the establishing of such theoretical bounds doesn't prevent MSGD from failing to converge during training in some situations (Section 4.3).

- Moshpit All-Reduce also takes into consideration network bandwidth.   Was this used in all experiments?  How was the bandwidth measured?   How often / was it stable?  How big an improvement did this provide and would it apply to other systems as well?
- S4.2 says that additional latency is added per communication round.  Is this added per host link or is it added per communicating pair (emulating latency at the end host or in the middle of the network)?  Is this delay left on for the duration of the communication round?
- S4.2: The paper says AR-SGD is "unsuitable for heterogenous hardware."  It would be better to provide data to support this statement, i.e., how bad is it?
- S4.2:  The paper compares against a gossip approach that communicates with 2 neighbors.  This setting appears unmotivated and rather small.   What happens if it is set in a similar fashion to the different group sizes for MSGD (4 and 8 groups)?
- S4.2 mentions "efficiency" but it's unclear what this represents.  Latency of each communication round? Number of messages? Number of bytes on the network?
- While Figure 4 is interesting (though very small), it would also be enlightening to plot summary statistics for accuracy and latency of averaging per epoch, per mechanism.  In some ways, these are the metrics the paper discusses, but we are forced to deduce them from these end products (though the results should be directly available).
* In S4.2, you say Moshpit SGD has "1 round of Moshpit averaging."  How does this relate to Algorithm 2?

* The heterogenous network is unspecified.  How many nodes were at each site?
* During actual model training it wasn't clear if there were node failures.
* What is the distribution of M_k in these heterogenous situations?  How often are there incomplete groups?

Like adjusting partition size based on bandwidth, one could also explore building the groups based on network coordinates (rack, container, data center, etc., locality).

__Edit [comments on author's response] :__  First, thank you for taking the time to help explain your work to the reviewers and provide additional supporting evidence for the work.  If the author's revise their work by adding the missing clarity noted by the reviewers (e.g., implications of group/grid size, how to choose group size), and incorporate the new analysis (as they suggest they will do), it will significantly enhance the paper.   I would also love to see an improved discussion of the impact of incomplete groups and failures based on the additional detail you provided.   In light of the author's commitment to improving clarity in the technical and experimental parts of the work, I will improve my score.




**Time Spent Reviewing:**

6-8

---

> ### Author Response · Authors · 2021-08-10
> **Author Response to Reviewer 5vXz (part 2)**
>
> > Moshpit All-Reduce also takes into consideration network bandwidth. Was this used in all experiments? How was the bandwidth measured? How often / was it stable? How big an improvement did this provide and would it apply to other systems as well?
>
> > S4.2: The paper says AR-SGD is "unsuitable for heterogenous hardware." It would be better to provide data to support this statement, i.e., how bad is it?
>
> While it was technically used in all experiments, only Section 4.2 is meaningfully affected by this optimization. In all other experiments, we use equal and symmetric bandwidth across all peers. In turn, the ALBERT experiment relies on download/upload bandwidths from Table 3 (Appendix G.2) that were obtained using the built-in bandwidth measuring tools from the GPU marketplace provider. To the best of our knowledge, the marketplace measures bandwidth by periodically running an upload/download benchmark(similar to speedtest.net) while the nodes are idle. The stability of network bandwidth varied from one machine to another, but the GPU providers were encouraged to maintain stable bandwidth by the rules of the marketplace.
>
> Applying similar methods to other techniques is generally possible, but requires designing a completely different optimization procedure based on the particular algorithm. For instance, in Ring All-Reduce, each vector chunk must iteratively go through all the nodes, which makes it impossible to “assign” each node to chunks of different size.
>
> As for the effect that it has on the training throughput: Sections 4.1 and 4.2 are unaffected for the reasons we specified above. In turn, Section 4.3 is **significantly** affected in the heterogeneous setup: in fact, not accounting for bandwidth would make it utterly impractical to train, since some nodes have <1/3  the upload speed of their peers and would slow down all-reduce by the same factor. This is also why other elastic training systems are impractical in this setup. To ensure a fair comparison, we do not make direct comparisons between Moshpit and other methods in heterogeneous setup. However, we agree that the paper would benefit from including this discussion and promise to include it in the final revision, using the additional page allowed in the camera ready version.
>
> > S4.2 says that additional latency is added per communication round. Is this added per host link or is it added per communicating pair (emulating latency at the end host or in the middle of the network)? Is this delay left on for the duration of the communication round?
>
> The latency is applied in a way that simulates delays within the network. More specifically, we set `tc qdisc add LINK_NAME root netem delay RANDOM_SAMPLED_DELAY` and leave it on for the duration of the experiment.
>
> > S4.2: The paper compares against a gossip approach that communicates with 2 neighbors. This setting appears unmotivated and rather small. What happens if it is set in a similar fashion to the different group sizes for MSGD (4 and 8 groups)?
>
> We chose this number of neighbors similar to [2]. The reason why gossip-based methods rarely use a large number of neighbors is, because, unlike All-Reduce, their communication load grows linearly with the number of neighbors. If we were to send gossip updates to 8 adjacent peers, each communication round would take up 4 times more time and would ultimately be less efficient than making 4 consecutive rounds with 2 neighbors. However, we consider one alternative graph that is in fact more efficient: the exponential graph described in [1] and used in [3]. This graph has $O(\log_2 n)$ edges and thus has increased communication load; however, the edges are arranged in such a way that allows global propagation in O(\log_2 n) rounds, as opposed to O(n) rounds in traditional AD-PSGD.
>
> > S4.2 mentions "efficiency" but it's unclear what this represents. Latency of each communication round? Number of messages? Number of bytes on the network?
>
> In this case (and in other instances), “efficiency” means the communication efficiency of the method, i.e. the amount of data sent between the workers at each averaging round. Thank you for pointing this out; we will clarify that in the new version of the paper.
>
> > While Figure 4 is interesting (though very small), it would also be enlightening to plot summary statistics for accuracy and latency of averaging per epoch, per mechanism. In some ways, these are the metrics the paper discusses, but we are forced to deduce them from these end products (though the results should be directly available).
>
> Because we are running a single averaging iteration both for Moshpit SGD and for gossip-based methods, it will not necessarily lead to a high averaging accuracy for the described experiments. Furthermore, computing the exact average deviation of the neural network parameters across all peers in our setup would actually be prohibitively expensive, since it would require exact averaging across a slow and heterogeneous network at each step of training.
>
> As a result, it actually might be difficult to compare the averaging performance of the methods in this case, which is why we report the task performance (in terms of accuracy and loss) as a metric that would be most interesting to machine learning practitioners.
>
> > In S4.2, you say Moshpit SGD has "1 round of Moshpit averaging." How does this relate to Algorithm 2?
>
> In this setup, we run Algorithm 1 with T=1, similarly to other decentralized methods.
>
> > The heterogenous network is unspecified. How many nodes were at each site?
>
> We believe you refer to the underspecified node distribution in L287-289: the distribution of all machine types was roughly uniform across both the data centers and the office.
>
> > During actual model training it wasn't clear if there were node failures.
> > What is the distribution of M_k in these heterogenous situations? How often are there incomplete groups?
>
> We agree that this aspect of experiments could be described more clearly and intend to add a more detailed discussion in the next version of the work. Due to the preemptible nature of used instances,  we observed both disconnected nodes and incomplete groups during our training runs. However, replicating the exact conditions of the setup to quantify the effect will require rerunning the experiment on the same platforms; thus, it might be difficult to obtain all required results with a high degree of confidence, but we may provide the ongoing measurements if deemed necessary by the reviewer.
>
> > Like adjusting partition size based on bandwidth, one could also explore building the groups based on network coordinates (rack, container, data center, etc., locality).
>
> Thank you for this suggestion! As this modification requires separate extensive investigation, we will include it in the conclusion as a possible direction for future work.
>
>
> [1] Asynchronous Decentralized Parallel Stochastic Gradient Descent. Xiangru Lian, Wei Zhang, Ce Zhang, Ji Liu. ICML 2018
>
> [2] Can Decentralized Algorithms Outperform Centralized Algorithms? A Case Study for Decentralized Parallel Stochastic Gradient Descent. Xiangru Lian, Ce Zhang, Huan Zhang, Cho-Jui Hsieh, Wei Zhang, Ji Liu. NeurIPS 2017
>
> [3] Stochastic Gradient Push for Distributed Deep Learning. Mahmoud Assran, Nicolas Loizou, Nicolas Ballas, Michael Rabbat. ICML 2019

---

> ### Author Response · Authors · 2021-08-10
> **Author Response to Reviewer 5vXz (part 1)**
>
> Thank you for a detailed review and a thorough evaluation of our paper! Below, we address your concerns to the best of our ability; if you have any further questions or feel that you need additional clarification, we would be happy to answer them during the response period.
>
> > There was a lack of description of the competing techniques (related work) beyond all-reduce. (S4.2).
>
> Although we indeed did not describe other decentralized methods in detail in this part, we described them and provided a list of references in a dedicated part (S2.4) of our “Related work” section. Also, we provide a more elaborate comparison with gossip-based approaches and a more detailed survey in the “Additional related work” section (Appendix B, specifically Section B.1).
>
> However, after having another look at our experimental setup description (L277-283), we noticed that a citation for AD-PSGD[1] was missing. Thus, we thank you for pointing out this issue and will fix the missing reference in an updated version of the work.
>
> > Given the complexity of the theoretical analysis, the empirical validation of Moshpit All-Reduce was notably weak.
> > While the end-to-end training experiments are compelling, the experiment construction makes comparisons (and lessons learned) difficult.
>
>
> We believe that both these points refer to the same issue: namely, that the setup of the averaging experiment might be improved. We agree with your point and provide more thorough comparisons as a part of the general response. Also, we address the specific comments regarding the experimental setup below.
>
> > While theoretical results (assuming the proofs are correct) are necessary, they may not always account for reality
>
> Although this statement is generally true, it does not undermine the importance of our theoretical results. We carefully choose our assumptions to be sufficiently general so as to represent the majority of practical optimization setups. Moreover, we provide non-asymptotic guarantees, i.e., we provide the rates of convergence. That is, we prove much more than the fact that the method converges. As a result, we consider our results to be a significant contribution on their own, not just something that is “necessary” to prove. Quite often, the papers on distributed optimization do not provide any convergence guarantees or provide too weak ones (either under too restrictive assumptions or asymptotic guarantees).
>
> > In Algorithm 1, it would be useful to characterize the asynchrony. For example, there is a wait for peers to assemble. Essentially, this is view maintenance/consensus. If a node "joins late" what occurs?
>
> In practice, if a node joins late, it does not participate in the current round and may download parameters from other peers in the next round of averaging.
>
> > If there is agreement on the peer set, and then peers experience failure, what is the impact on the latency of the round (internal all-reduce timeouts), what is the impact of that timeout on convergence?
>
> If the peers in a given group experience a failure, it does not impact the latency of concurrent averaging in any other groups. Although it is difficult to characterize the impact of timeouts on convergence,
>
> > Is it possible for certain network or failure conditions to allow butterfly all-reduce to end in an inconsistent state?
>
> Although in practice certain errors may indeed lead to algorithmic failures (which is true for any distributed system), in our case, most of the issues are alleviated by the rollback mechanism. That is, if at any point of averaging the peer detects an inconsistency, it sends the “internal error” message to other peers in the same group, and the averaging round for this group is cancelled.
>
> We note, however, that the specifics of exception handling are an implementation detail of any given distributed algorithm: for example, see lines 173-180 of src/client/averaging/allreduce.py in our submitted implementation as an example of error detection/propagation mechanism.
>
> >Theorem 3.2 maintains that the sum of all the groups is still N. While the theorem explores an important relaxation of the initial assumptions, node failure/disconnections imply that rounds may proceed that violate that assumption.
>
> In fact, this assumption of constant sum of group sizes is not violated by the existence of node failures, because (as stated in L192) each group size can be equal to 1. Thus, a failed node (or a group of failed nodes) that does not participate in averaging at a given round can be assigned a group of size 1: in essence, it averages with itself, i.e. keeps the same value. As we state in L201-202 and Appendix C.3, the main result can be generalized to a case of the number of groups that varies between iterations.
>
> > To establish these properties the microbenchmarks should a.) plot the theoretical and competing systems (S4.2) results with the empirical results, b.) should change one aspect at a time (as opposed to changing N and p simultaneously), and c.) report performance statistics as a function of increasing node failure and (separately) connection latency.
>
> We agree with this comment and provide additional experimental results in the general response: specifically, we compare the performance of different methods for distributed iterative averaging in presence of node failures and study the influence of specific aspects of the setup on the final performance.
>
> However, we argue that simulating the connection latency in a microbenchmark setting will not yield meaningful results, because the runtime estimates will largely depend on the network-aware implementation of the averaging algorithm and the findings may not transfer to a more practical setup. We conducted two experiments in realistic setups (Section 4.2, 4.3) and measured convergence with respect to the wall time precisely to give the reader more insight into the real-world performance of each method.
>
> > Section 4.1 doesn't clearly state M or d (leaving the reader to parse "32x32" 'grid'). I'm assuming "18 steps" is T=18.
>
> Thank you for the comment! We will update the paper and explain the correspondence between variables and their functions more clearly.
>
> > When failed peers return, do they return with the last returned parameters?
>
> Yes; in this experiment, we study the performance of the averaging method itself. To the best of our knowledge, optimal recovery strategy for a previously failed peer is an important question that needs to be addressed separately, and thus we omit it from the controlled evaluation.
>
> > More importantly, failing for individual rounds seems contrived. What about failing for some unspecified period of time (modeled with some distribution)? In reality, peers aren't going to fail on nicely behaved round boundaries.
>
> While it is true that failing for individual rounds is a quite simple setup, we believe that it is important to keep the analyzed setting of Section 4.1 clear and concise, because its goal is to demonstrate and verify the core properties of the proposed algorithm. Modeling the distribution of disconnection/connection time comes with additional assumptions that might be difficult to explain and justify, especially because the following sections contain a more realistic setup anyway.
>
> > Also, I don't understand line 268. Why does this experiment converge in two steps with failures? Two steps are the minimum to achieve an exact answer without failure. Were there no failures?
>
> We thank the reviewer for pointing that out. In this specific scenario, the failure rate was so small that there was a 10-15% chance that all peers would finish the averaging rounds without failures, which they seemingly did. We will provide the average number of iterations for a given accuracy over multiple retries.
>
> >On that same graph, why would N=900 with no failures take so many steps (9) to converge to the true answer? Clearly, even without failures, the true size of the groups must be << than 32.
>
> For the 32x32 grid, there were on average 30 peers in each group, with a few peers “missing” from the cells. The lack of these few peers causes the next averaging round to also be “incomplete”, since the second round is no longer guaranteed to aggregate peers that contain “row-wise” averages. However, we would like to emphasize that the algorithm still converges exponentially and obtains an error of <1e-5in 3 steps.

---

### Official Review · Reviewer_LKRP · 2021-07-18

**Rating:** 7
**Confidence:** 4

**Summary:**

This paper introduces Moshpit All-Reduce, a fully decentralized iterative averaging protocol for large-scale training under unstable communication conditions and/or with unreliable workers. The proposed protocol allows an exponential convergence to the global average. It is shown that Moshpit SGD, built on top of Moshpit All-Reduce, is equivalent to the centralized SGD in terms of iteration complexity.

**Limitations And Societal Impact:**

They addressed the limitations in Appendix. I agree with the authors that it is hard to assess potential negative societal impact of this work/application.

**Main Review:**

* Originality and Significance: This paper considers distributed learning over unreliable worker and communication channel. The topic is of significance in that the proposed method resolves the issues in the existing gossip based algorithms lifting the fundamental restriction and tradeoff between the communication required and convergence rate.

* Technical quality: The technical contribution/quality of this paper overall looks good to me (didn't fully check the proof though). My only concern is that some part in the experiment section is not very clear to me. Please refer to the questions at the end of the comment (Miscellaneous).

* Clarity in organization/writing:

-- The organization and writing are mostly clear to me. I especially like the organization of the introduction and related work sections. I believe that It would be helpful for readers who just jumped into this area as it is easy to read and contains a fairly good list of references.

-- Miscellaneous
1) line 49: Can we quantitatively measure the additional burden for dynamically organizing workers? Do we need intervention of a moderator or any kind of centralized operation to do this?
2) line 253: It is not clear what "all information is replicated" means. Please elaborate more.
3) line 264: It seems that the authors use the gaussian noise to model errors added to the values of the original vectors. Please consider modifying this sentence to contain the full detail.
4) Figure 3: Can we think of any tradeoffs from this figure? It was surprising that there is a fairly large gap in the behavior between the red and brown lines given the same p and a small difference in N. It would be better if the readers could get some more intuition behind the results shown in this figure.


**Time Spent Reviewing:**

3

---

> ### Author Response · Authors · 2021-08-10
> **Author Response to Reviewer LKRP**
>
> Thank you for the valuable feedback! Below, we address the concerns you raised in the review:
>
> > line 49: Can we quantitatively measure the additional burden for dynamically organizing workers? Do we need intervention of a moderator or any kind of centralized operation to do this?
>
> This is indeed an important question. As we state in L47-51, L140-141 and later in L175-176, Moshpit All-Reduce uses a fully decentralized matchmaking protocol on top of a Kademlia DHT: as such, a moderator is not required. The additional computation and network costs incurred by this protocol are negligible (<1% compute, <3% data transfer), but it does take up to 3 seconds due to network latency (in the setup from S4.3). To address this issue, Moshpit SGD performs matchmaking asynchronously, i.e. forms groups asynchronously in parallel with the main training code. This allows us to completely bypass the time overhead from matchmaking.
>
> > line 253: It is not clear what "all information is replicated" means. Please elaborate more.
>
> We specifically refer to the way Kademlia DHT stores all entries: each key-value pair is replicated on $k$ peers chosen based on the hash function of the key. Thus, if some of these peers leave abruptly or experience a hardware fault, others will still be able to access the data. Since we already discuss this idea in Appendix B.5 (L888-890), we will add the reference to that appendix in L253.
>
> > line 264: It seems that the authors use the gaussian noise to model errors added to the values of the original vectors. Please consider modifying this sentence to contain the full detail.
>
> In fact, our experiments are conducted precisely in the described setting: we average the values of a vector generated from a Gaussian distribution. This allows us to simplify the setup and to avoid relying on the structure of the weight space for a neural network.
>
> > Figure 3: Can we think of any tradeoffs from this figure? It was surprising that there is a fairly large gap in the behavior between the red and brown lines given the same p and a small difference in N. It would be better if the readers could get some more intuition behind the results shown in this figure.
>
> The reason behind this tradeoff is that we use a 32x32 grid for all experiments. As such, the red line corresponds to a best case scenario of 1024 nodes with a full 32x32 grid. In turn, the brown line with 900 nodes corresponds to a 32x32 grid where approximately 1 in 10 “cells” are missing, which leads to slower averaging convergence. In contrast, if we were to use a smaller 30x30 grid, 900 peers would be averaged significantly faster. We agree that this behavior is not trivial and will discuss it in the paper, as well as in the appendix H on additional averaging experiments.

---

### Official Review · Reviewer_S2Ah · 2021-07-24

**Rating:** 8
**Confidence:** 3

**Summary:**

This paper presents a new collective algorithm and associated implementation of SGD training that is optimized for distributed systems where some ranks will not participate in each training step.  The observation is that parameter servers are not scalable, and that most collective based (all-reduce) distributed algorithms are either not fault tolerant, or have significant performance impacts when there is load imbalance or system noise.  The proposed Moshpit all-reduce and SGD strive to overcome these challenges by breaking down the collective communication domains into many more "sub-domains" that can tolerate delays or absences of its participants.  However, it does so in a structured manner that is more communication efficient than a standard gossip, random, algorithm.  They show that despite stale gradients for each participant, the algorithm still converges and does so faster than SOTA algorithms.

**Main Review:**

Overall, the paper presents a clear and compelling description of the proposed algorithm.  Both the pseudo-code and block diagrams did a good job of conveying the algorithms details.  Evaluations were performed on both a standard visual network as well as a transformer based language model.  The paper also provides a detailed description of the convergence properties of the algorithm, which are empirically supported by the results shown in Figure 4.

**Time Spent Reviewing:**

2

---

> ### Author Response · Authors · 2021-08-10
> **Author Response to Reviewer S2Ah**
>
> Thank you for your review, along with the summary of our contributions and the kind words regarding their presentation! If any questions arise during the discussion phase, we would be happy to answer them.

---

### Author Response · Authors · 2021-08-12
**General Response to Reviewers**

We thank the reviewers for their feedback and efforts they put into reading and evaluating our work. In particular, reviewers appreciated the following aspects of our work:

* **Originality and significance** (**Reviewer LKRP**: “the proposed method resolves the issues in the existing gossip based algorithms lifting the fundamental restriction and tradeoff”, **Reviewer 5vXz**: “Robust training on cheap(er) nodes/networks is an important challenge”, “Paper provides a novel communication algorithm for efficiently performing an all-reduce average”, **Reviewer XkgK**: “I have enjoyed reading this paper, and I particularly found the moshpit match-making scheme clever and interesting”);

* **Strength of theoretical and empirical results** (**Reviewer S2Ah**: “The paper also provides a detailed description of the convergence properties of the algorithm, which are empirically supported by the results”, **Reviewer LKRP**: “The technical contribution/quality of this paper overall looks good to me”, **Reviewer 5vXz**: “Extensive analysis provides support for its [Moshpit-SGD] convergence and robustness properties”, “Experiments on homogenous and heterogenous networks with varying network latencies illustrate its ability to quickly approximate the true average and support distributed SGD”);

* **Clarity of organization and writing** (**Reviewer S2Ah**: “the paper presents a clear and compelling description of the proposed algorithm. Both the pseudo-code and block diagrams did a good job of conveying the algorithm details”, **Reviewer LKRP**: “I especially like the organization of the introduction and related work sections. I believe that It would be helpful for readers who just jumped into this area as it is easy to read and contains a fairly good list of references”, **Reviewer 5vXz**: “The writing was clear and the notation consistent”, **Reviewer XkgK**: “The quality of the writing is good and the exposition is clear”).

We also appreciate all the comments and questions raised by the reviewers, and address them in our responses. In particular, two common concerns raised by the reviewers were related to the details of the matchmaking procedure and the setup of the averaging experiment. For the first concern, we answered the reviewers’ questions about the details of that part of the algorithm and will add a better explanation in the text.

For the second question, we conducted an additional sequence of averaging experiments comparing Moshpit Averaging with all baselines: All-Reduce (with restarts in case of failures), Gossip, PushSum (as in SGP), and group averaging with random groups. Note, however, that the last baseline was implemented so that the group sizes are roughly the same across all iterations, which might be difficult to accomplish in a decentralized setting. We consider the number of averaging nodes from 512 to 1024 and peer failure probabilities from 0 to 0.01 to separate the factors of variation and study the dependency of method convergence on these conditions, as per the suggestions of Reviewer 5vXz.

The results for the group size of 32 and 2 grid dimensions are given below. The values in each cell denote the number of iterations required to achieve the mean squared error of 1e-9 (1e-4 in parentheses), averaged across 100 runs from different random initializations.

| Nodes | $p$ of failure | All-Reduce  | Gossip  	| SGP     	| Random groups | Moshpit   |
|-------|----------------|-------------|-------------|-------------|---------------|-----------|
| 512   | 0          	| **1.0 (1.0)**   | 50.0 (50.0) | 47.6 (15.6) | 6.1 (3.0) 	| 8.5 (3.5) |
| 512   | 0.001      	| **1.6 (1.6)**   | 50.0 (50.0) | 47.6 (15.6) | 6.3 (3.0) 	| 8.7 (3.5) |
| 512   | 0.005      	| 10.9 (10.9) | 50.0 (50.0) | 47.8 (15.6) | **6.3 (3.0)** 	| 8.8 (3.8) |
| 512   | 0.01       	| 41.7 (41.7) | 50.0 (50.0) | 47.8 (15.6) | **6.6 (3.0)** 	| 8.9 (3.9) |
| 768   | 0          	| **1.0 (1.0)**   | 50.0 (50.0) | 43.2 (13.8) | 6.2 (3.0) 	| 6.0 (3.0) |
| 768   | 0.001      	| **1.8 (1.8)**   | 50.0 (50.0) | 43.2 (13.8) | 6.5 (3.0) 	| 6.1 (3.0) |
| 768   | 0.005      	| 28.7 (28.7) | 50.0 (50.0) | 43.2 (14.1) | 6.6 (3.0) 	| **6.5 (3.0)** |
| 768   | 0.01       	| 50.0 (50.0) | 50.0 (50.0) | 43.9 (14.2) | **7.0 (3.0)** 	| 7.1 (3.0) |
| 900   | 0          	| **1.0 (1.0)**   | 50.0 (50.0) | 45.0 (14.7) | 6.4 (3.0) 	| 5.0 (2.7) |
| 900   | 0.001      	| **1.8 (1.8)**   | 50.0 (50.0) | 45.0 (14.7) | 6.3 (3.0) 	| 5.0 (3.0) |
| 900   | 0.005      	| 50.0 (50.0) | 50.0 (50.0) | 45.2 (14.7) | 6.7 (3.0) 	| **5.9 (3.0)** |
| 900   | 0.01       	| 50.0 (50.0) | 50.0 (50.0) | 45.6 (14.9) | 7.0 (3.1) 	| **6.1 (3.0)** |
| 1024  | 0          	| **1.0 (1.0)**  | 50.0 (50.0) | 49.0 (16.2) | 6.2 (3.0) 	| 2.0 (2.0) |
| 1024  | 0.001      	| **2.0 (2.0)**   | 50.0 (50.0) | 49.0 (16.3) | 6.5 (3.0) 	| 3.4 (2.3) |
| 1024  | 0.005      	| 42.6 (42.6) | 50.0 (50.0) | 49.5 (16.3) | 6.7 (3.0) 	| **4.9 (3.0)** |
| 1024  | 0.01       	| 50.0 (50.0) | 50.0 (50.0) | 49.5 (16.3) | 6.9 (3.1) 	| **5.7 (3.0)** |

From these results, we can make several key observations:
1) When the failure rate of each peer is zero, All-Reduce predictably obtains the average faster than all other methods. However, as soon as this probability reaches a value of at least 0.005, the number of retries needed for a success becomes prohibitively high.
2) Previous decentralized averaging methods, such as Gossip or PushSum, require significantly more iterations for convergence to the global average than Moshpit All-Reduce due to the construction of their graphs. This is likely the reason why AD-PSGD still needs to output the average of all weights across all workers at the end of training.
3) As discussed in the paper, when the total number of peers is equal to the grid capacity, Moshpit All-Reduce matches the result of regular All-Reduce with the number of steps equal to the number of dimensions.
4) Averaging in random groups performs comparably to Moshpit Averaging when the number of peers takes around a half of the Moshpit grid. As the grid utilization grows, the difference in convergence becomes more noticeable. However, even if we use 512 peers, arranging them in a proper 8x8x8 grid leads to faster convergence:

| Nodes | $p$ of failure | All-Reduce  | Gossip  	| SGP     	| Random groups | Moshpit   |
|-------|----------------|-------------|-------------|-------------|---------------|-----------|
| 512   | 0          	| **1.0 (1.0)**   | 50.0 (50.0) | 47.6 (15.6) | 10.2 (5.0)	| 3.0 (3.0) |
| 512   | 0.001      	| **1.6 (1.6)**   | 50.0 (50.0) | 47.6 (15.6) | 10.3 (5.0)	| 4.7 (3.1) |
| 512   | 0.005      	| 10.9 (10.9) | 50.0 (50.0) | 47.8 (15.6) | 10.7 (5.0)	| **7.5 (3.7)** |
| 512   | 0.01       	| 41.7 (41.7) | 50.0 (50.0) | 47.8 (15.6) | 10.6 (5.0)	| **8.1 (4.0)** |

As stated in our response to reviewers LKRP and 5vXz, we will provide the above evaluation and explain the influence of grid arrangement on the results in the final version of the paper. In addition, we will elaborate on the practical aspects of averaging in random groups.

We will be happy to answer any remaining questions and to actively participate in the discussion with reviewers.

---

### Decision · Program_Chairs · 2021-09-27

**Decision:**

Accept (Poster)

**Comment:**

This paper proposes a novel approach to distributed data-parallel training, MoshpitSGD. The approach targets systems where fault tolerance and robustness important, e.g., when some worker nodes may fail, there is load imbalance, or other sources of heterogeneity in the system. This is a setting relevant to practitioners applying distributed training.

The reviewers agreed that the contribution is interesting and worthy of acceptance. While there were some questions and concerns raised in the initial reviews, the authors’s responses largely addressed these. In particular, the new empirical results nicely illustrate the tradeoff between messaging rounds and resilience to node failures. It is important that the camera-ready revision should include the points mentioned during the rebuttal to address questions raised by the reviewers to improve the clarity of the paper, especially around experimental setup and findings. We also strongly encourage the authors to include the additional experiments in the paper and make the experiments more focused on specific contributions (e.g., the proposed match-making scheme) in addition to highlighting performance of the method as a whole. This will likely involving moving/adding material to the supplementary material.